# Paths and pathways that generate cell-type heterogeneity and developmental progression in hematopoiesis

Juliet R Girard[1†], Lauren M Goins[1†], Dung M Vuu[1†], Mark S Sharpley[1], Carrie M Spratford[1], Shreya R Mantri[1], Utpal Banerjee[1,2,3,4*]

[1]Department of Molecular, Cell and Developmental Biology, University of California, Los Angeles, Los Angeles, United States; [2]Molecular Biology Institute, University of California, Los Angeles, Los Angeles, United States; [3]Department of Biological Chemistry, University of California, Los Angeles, Los Angeles, United States; [4]Eli and Edythe Broad Center of Regenerative Medicine and Stem Cell Research, University of California, Los Angeles, Los Angeles, United States

**Abstract** Mechanistic studies of *Drosophila* lymph gland hematopoiesis are limited by the availability of cell-type-specific markers. Using a combination of bulk RNA-Seq of FACS-sorted cells, single-cell RNA-Seq, and genetic dissection, we identify new blood cell subpopulations along a developmental trajectory with multiple paths to mature cell types. This provides functional insights into key developmental processes and signaling pathways. We highlight metabolism as a driver of development, show that graded Pointed expression allows distinct roles in successive developmental steps, and that mature crystal cells specifically express an alternate isoform of Hypoxia-inducible factor (Hif/Sima). Mechanistically, the Musashi-regulated protein Numb facilitates Sima-dependent non-canonical, and inhibits canonical, Notch signaling. Broadly, we find that prior to making a fate choice, a progenitor selects between alternative, biologically relevant, transitory states allowing smooth transitions reflective of combinatorial expressions rather than stepwise binary decisions. Increasingly, this view is gaining support in mammalian hematopoiesis.

*For correspondence:
banerjee@mbi.ucla.edu

†These authors contributed equally to this work

## Editor's evaluation

This paper will be of interest to scientists who study hematopoiesis. The authors combine single cell RNA-seq with bulk RNA-seq of transcripts from blood cells in the *Drosophila* larval hematopoietic organ. They present extensive analysis of the datasets, and the pseudotime analyses present a model of how hematopoietic progenitors can differentiate along transitory paths. These datasets reveal cell-type specific isoform expression of Notch pathway regulators, and genetic experiments prove the importance of these factors in development of one lineage. These transcriptomic analyses and subsequent genetic experiences provide strong support for the major claims of the paper.

## Introduction

The *Drosophila* lymph gland is the major hematopoietic organ that develops during the larval stages for the purpose of providing blood cells during later pupal/adult periods (reviewed in *Banerjee et al., 2019*). Hematopoietic function for the larva itself is largely provided by a separate set of sessile or circulating blood cells outside of the lymph gland (reviewed in *Letourneau et al., 2016*). The only time the lymph gland provides blood cells to the circulating larval hemolymph is if the larva faces a

stress or immune challenge. This study entirely concentrates on the primary/anterior lobes of the lymph gland, which display the highest hematopoietic activity during normal larval development.

Past work has identified specific functional zones. The PSC (Posterior Signaling Center) is marked by expression of *Antp* (*Mandal et al., 2007*) and *knot/collier* (*kn/col*) (*Crozatier et al., 2004*). The PSC signals progenitors that belong to the medullary zone (MZ) and are marked by *dome^MESO* and *Tep4* (*Jung et al., 2005*; *Irving et al., 2005*). Differentiating cells form the cortical zone (CZ), expressing *Hemolectin* (*Hml*), *Peroxidasin* (*Pxn*), *lozenge* (*lz*), and other differentiating cell markers (*Jung et al., 2005*). A narrow band of cells that are double positive for *dome^MESO* and *Hml^Δ* occupy the edge abutting these two zones in the early third instar (*Sinenko et al., 2009*), and is referred to as the intermediate zone (IZ), which contains intermediate progenitors (IPs) (*Krzemien et al., 2010*).

Invertebrates predate the evolution of the lymphoid system for adaptive immunity. Accordingly, *Drosophila* blood cells are all similar in function to cells of the vertebrate myeloid lineage. The most predominant class of blood cells, the plasmatocytes (PLs; 95% of all hemocytes), share a monophyletic relationship with vertebrate macrophages. PLs function in the engulfment of microbes and apoptotic cells, and they produce extracellular matrix proteins (*Fessler and Fessler, 1989*; *Tepass et al., 1994*; *Franc et al., 1996*). A minor (2–5%), but important class is represented by crystal cells (CCs) named for their crystalline inclusions of the pro-phenoloxidase enzymes, PPO1 and PPO2. CCs are necessary for melanization, blood clot formation, immunity against bacterial infections, and to help mitigate hypoxic stress (*Rämet et al., 2002*; *Galko and Krasnow, 2004*; *Binggeli et al., 2014*; *Dudzic et al., 2015*; *Cho et al., 2018*). The transcription factor Lozenge (Lz) cooperates with Notch signaling to express a number of target genes (such as *hindsight/pebbled*) to specify CCs (*Lebestky et al., 2000*; *Duvic et al., 2002*), whereas the Sima (vertebrate HIF-1α) protein is required for their maintenance (*Mukherjee et al., 2011*). The orthologue of Lz in mammals is RUNX1, with broad hematopoietic function at many developmental stages, and RUNX1 is often dysregulated in acute myeloid leukemias (*de Bruijn and Speck, 2004*; *Ito, 2004*). The third class of blood cells, lamellocytes (<1%), is usually present only during parasitization by wasps (reviewed in *Letourneau et al., 2016*).

In early genetic studies, the MZ appeared to consist of a fairly homogeneous group of cells, although a small number of cells clustered near the heart (dorsal vessel) are identified as pre-progenitors (*Jung et al., 2005*; *Dey et al., 2016*; *Tiwari et al., 2020*). More recent reports have noted considerable heterogeneity and complexity within the progenitor population (reviewed in *Banerjee et al., 2019*). Particularly noteworthy, in this context, is the functional distinction into a Hh-sensitive and a Hh-resistant group of progenitors within the MZ (*Baldeosingh et al., 2018*).

Hematopoiesis requires complex collaborations between direct cell to cell signals (e.g., Serrate/Notch), interzonal communication (e.g., Hedgehog), signals from the neighboring cardiac tube (*Morin-Poulard et al., 2016*; *Destalminil-Letourneau et al., 2021*), and systemic signals (e.g., olfactory and nutritional) (*Lebestky et al., 2003*; *Crozatier et al., 2004*; *Mandal et al., 2007*; *Shim et al., 2012*; *Shim et al., 2013*; *Ferguson and Martinez-Agosto, 2014*). An important type of interzonal signaling mechanism relevant to this paper involves multiple cell types across the zones. In brief, progenitors are maintained not only through PSC-derived signals but also through a signaling relay mediated by the differentiating cells. This backward signal from the differentiating cells to the precursors is named the Equilibrium Signal (*Mondal et al., 2011*; *Mondal et al., 2014*). In this process, Pvf1 (PDGF- and VEGF-related factor 1) produced by the PSC, trans-cytoses through the MZ to bind its receptor Pvr (PDGF/VEGF receptor), which is expressed at high levels in the CZ. This initiates a STAT-dependent but JAK-independent signaling cascade that ultimately leads to the secretion of the extracellular enzyme ADGF-A (adenosine deaminase-related growth factor A). This enzyme breaks down adenosine, preventing its mitogenic signal and proliferation of MZ progenitors. Acting together the niche and the backward signal maintain a balance between progenitor and differentiated cell types. The genetic studies broadly implicated the CZ cells as originators of this backward signal. Finer analysis, afforded by cell-separated bulk and single-cell RNA-Seq in this study, allows us to attribute this role to a smaller and more specific subset of cells.

RNA-Seq has been used recently as a technique to study *Drosophila* blood cells (*Cattenoz et al., 2020*; *Cho et al., 2020*; *Fu et al., 2020*; *Ramond et al., 2020*; *Tattikota et al., 2020*). Four of the cited studies analyze circulating blood cells that have a completely different developmental profile than the lymph gland. *Cho et al., 2020* utilized the lymph gland and validated its zonal structure at the level of gene expression. Additionally, new markers and sub-zones were identified. The broader

picture revealed in our current manuscript is largely consistent with *Cho et al., 2020*, but several important details and interpretations vary. The results and conclusions of the two independent studies are compared and contrasted later in this paper. Importantly, the primary motivation of this current study is to use the combined strategies of several RNA-Seq analyses as a tool to provide data that can be combined seamlessly with the powerful genetics available in *Drosophila*. This functional validation of the two approaches is an advancement over the use of transcriptomics to distinguish cell types by their expressed markers. This is a level of in vivo mechanistic analysis that is not yet available for many mammalian systems, but for which *Drosophila* could serve as a model. While this work also describes subzones and their characteristic markers, the primary emphasis that makes it distinct is the use of a complex strategy that allows us to extend beyond cell type identification and to dissect mechanisms that define alternate paths and pathways that were not solvable by earlier genetic methods alone.

The novel conclusions from this analysis include a clear characterization of the IZ cells (IPs), and a demonstration of the IPs as a distinct cell type; identification of two separate transitional populations that define distinct paths between progenitors and differentiated cells fates; the role of metabolism in a zone-specific developmental program; previously uncharacterized functional aspects of transcriptional regulation by the JNK and RTK pathways; the unique mechanism of CC maturation by a novel and specific isoform of Sima identified in the RNA-Seq analysis and a previously uncharacterized interaction of this Sima isoform with Notch, Numb, and Musashi, which provides a full mechanism for CC formation and maintenance.

This combination of molecular genetics and whole genome approaches makes it clear that hematopoietic cells are far more heterogeneous and diverse than previously realized by genetics alone, and helps shift our view of hematopoiesis from being a series of discrete steps to a more continuous journey of cells with similar, but not identical transcriptomic profiles along multiple paths. The multiplicity in layers of decision points creates new routes, which can each lead to a distinct differentiated endpoint, or, alternatively, follow their parallel trajectories to a single final outcome.

## Results
### Bulk RNA-Seq analysis of zonal patterning within the lymph gland

To better understand the distribution of gene-expression patterns in different lymph gland zones, we utilize a combination of established, directly driven, reporter constructs that mark the MZ (*dome^MESO* enhancer driven nuclear EGFP) as well as the CZ (*Hml^Δ* enhancer driven nuclear DsRed). These markers are not GAL4-driven and therefore allow simultaneous visualization and manipulation of different cell types (*Figure 1A–A'*). Lymph glands from these marked third instar larvae are dissected and the primary lobes are separated from the rest of the lymph gland. Our samples do not include any of the posterior lobes. Following dissociation of the primary lobes and FACS sorting the resulting cells, two single positive cell types for each marker and a distinct cell population that is positive for both markers are found (*Figure 1B*; *Figure 1—figure supplement 1A*). These three represent cells of the MZ, CZ, and IZ, respectively. The transitioning cells of the IZ (*Sinenko et al., 2009*) are referred to as IPs (*Krzemien et al., 2010*). Direct drivers and nuclearly localized fluorescent reporters make this double positive population easy to identify, both in the intact lymph gland (*Figure 1A–A'*) and in dissociated cells (*Figure 1B*). IPs express lower levels of the markers EGFP and DsRed than in MZ and CZ cells, respectively (*Figure 1B*). The three gated populations are used in bulk RNA-Seq experiments. A fourth population that is double negative for both markers is also detected in the FACS sorted populations. We have not characterized these cells in detail, as they comprise a mixed population including the PSC, which is not marked in these tissues, but is explored in the single-cell RNA-Seq (scRNA-Seq) experiments.

Each bulk RNA-Seq sample utilizes cells from 100 lymph glands from mid-third instar larvae (90–96 hr after egg lay [AEL] at 25°C). Three biological replicates are analyzed for each sample and approximately 11,000 genes meet our threshold criteria for transcript expression. Previously established 'hallmark genes,' such as *Tep4*, *dome*, *shg/E-Cad*, and *kn/collier* (*Benmimoun et al., 2015*), (and *EGFP*), are detected in the MZ (*Figure 1C*). Similarly, *vkg*, *Col4a1*, *Hml*, *Pxn*, and *NimC1* (and *DsRed*) transcripts are enriched in the CZ. The transcript expression for known markers is a validation of the Bulk-Seq approach. In addition, we identify several novel genes that are differentially expressed

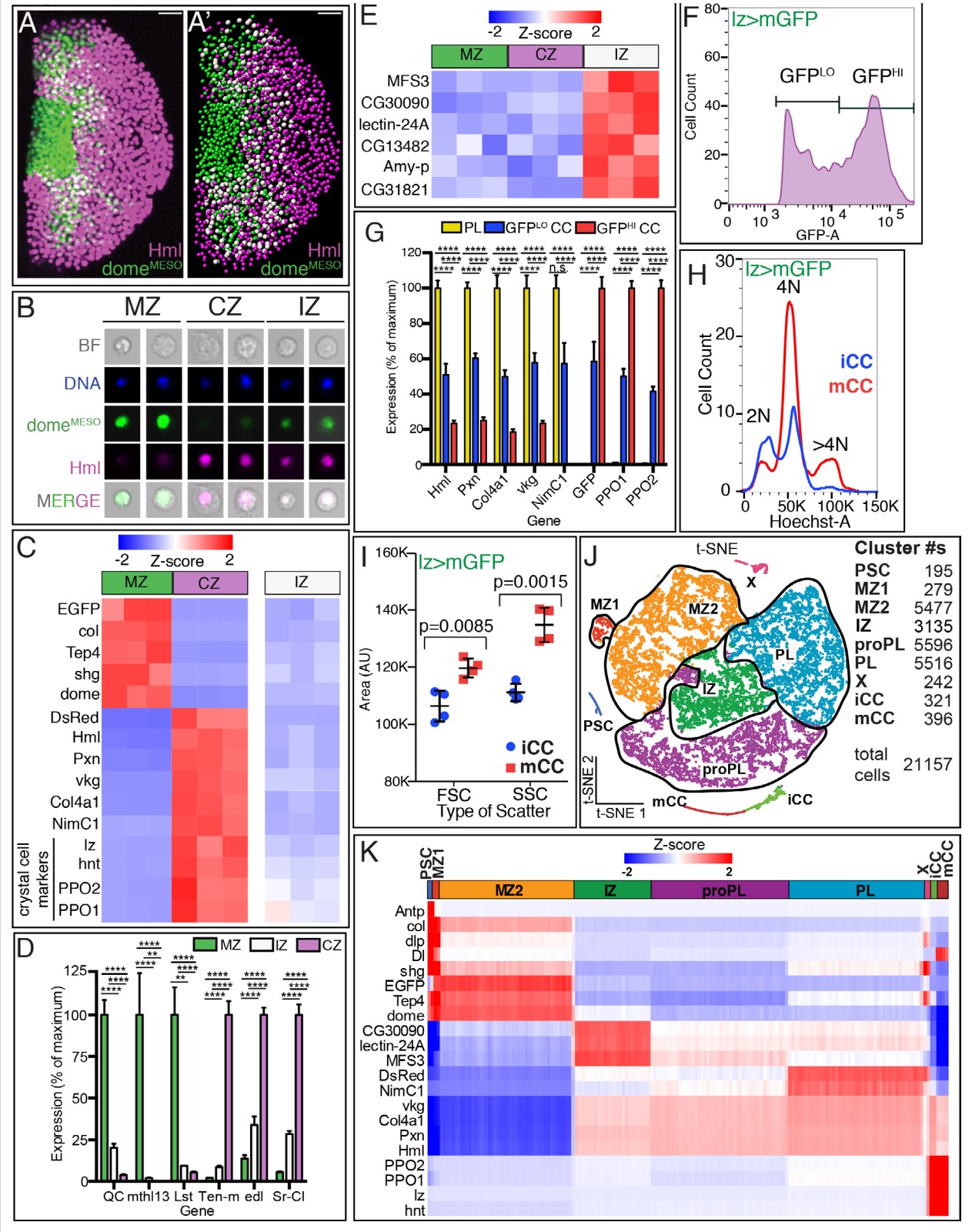

**Figure 1.** Analysis of subzonal patterning of the lymph gland by RNA-Seq. (**A–E**) Genotype of glands used is *dome^MESO-GFP.nls, Hml^Δ-DsRed.nls* GFP (*dome*) is in green and DsRed (*Hml*) is in magenta. (**A**) Confocal image shows the zonal pattern of an early third instar lymph gland. Progenitors of the medullary zone (MZ, green), differentiated cells of the cortical zone (CZ, magenta), and cells in the intermediate zone (IZ, white due to colocalization of green and magenta) are seen as distinct cell types. (**A'**) A digital rendering of the confocal image is shown in (**A**). Nuclei are pseudo-colored based on

*Figure 1 continued on next page*

*Figure 1 continued*

their fluorescence: MZ (green), CZ (magenta), and IZ double positive cells (white). (**A–A'**) Scale bars, 20 μm. (**B**) Images of individual dissociated lymph gland cells. Brightfield (BF), DAPI/DNA (blue), MZ cells (green), CZ cells (magenta), and IZ cells (green [weak] and magenta). (**C–E**) Gene expression profiles from bulk RNA-Seq analysis of dissociated and sorted cells in three biological replicates. (**C**) The sorted MZ and CZ cells express high levels of their corresponding hallmark genes while IZ cells show low to moderate levels of expression of these genes. MZ progenitors are validated by their expression of: *dome^MESO^-EGFP*, *col* (*collier*), *Tep4*, *shg* (*shotgun; E-Cad*), and *dome* (*domeless*); CZ plasmatocytes (PLs) by: *Hml^Δ^-DsRed*, *Hml* (*Hemolectin*), *Pxn* (*Peroxidasin*), *vkg* (*viking*), *Col4a1*, and *NimC1*; and the crystal cells (CC; also part of CZ) by: *lz* (*lozenge*), *hnt* (*hindsight/pebbled*), *PPO2*, and *PPO1*. (**D**) Newly identified zone-enriched genes for MZ include *QC*, *mthl13*, and *Lst*. For CZ, these include *Ten-m*, *edl*, and *Sr-CI*. In general, IZ cells show low to moderate levels of these MZ and CZ-specific markers. (p-values shown are from GSA analysis. Mean with SD shown.) (**E**) Expression of six newly identified IZ-enriched marker genes is not enriched in MZ or CZ. (**F–I**) Genotype: *lz-GAL4, UAS-mGFP; Hml^Δ^-DsRed.nls*. CCs expressing *lz-GAL4* are marked by GFP whereas PLs express DsRed only. Lymph glands are dissociated and the cells are subjected to flow cytometric and/or bulk RNA-Seq analysis. (**F**) Flow cytometry identifies two distinct populations of CCs, expressing either low GFP (GFP^LO^) or high GFP (GFP^HI^). These two CC populations are referred to as iCCs and mCCs, respectively (see text). (**G**) PLs show high expression of *Hml, Pxn, Col4a1, vkg,* and *NimC1* and no expression of *lz> mGFP, PPO1,* and *PPO2*. GFP^LO^ CCs (iCCs) show moderate levels of both PL and CC specific genes. GFP^HI^ CCs (mCCs) show high *PPO1* and *PPO2*, no expression of *NimC1* and low expression of other PL markers. (p-values shown are from GSA analysis. Mean with SD shown.) (**H**) DNA (Hoechst-A) measurement shows that iCCs have 2 N or 4 N DNA content, while mCCs have a significant number of cells with >4 N DNA content indicative of endoreplication. (**I**) Quantitation of data from four individual experiments. mCCs have higher mean FSC-A (cell size) and mean SSC-A (cellular complexity) values than iCCs. (p-values shown are from unpaired t-test. Mean with SD shown.) (**J, K**) Single-cell RNA-Seq analysis of dissociated cells from *dome^MESO^-GFP.nls, Hml^Δ^-DsRed.nls* lymph glands. (**J**) 2D t-SNE visualization of graph-based clustering identifies nine clusters: PSC (dark blue), MZ1 (red), MZ2 (orange), IZ (green), proPL (purple), PL (light blue), X (pink), iCC (light green), and mCC (dark red). The number of cells in each cluster is indicated on the right. (**K**) Expression analysis of hallmark genes shows enrichment in appropriate clusters. PSC (*Antp, col,* and *dlp*), MZ (*shg, EGFP, Tep4,* and *dome*), IZ (*CG30090, lectin-24A,* and *MFS3*), CZ (*DsRed, vkg, Col4a1, Pxn,* and *Hml*), mature PLs (*NimC1*), and CCs (*PPO2, PPO1, lz* and *hnt*). *Delta (Dl), shg, dlp,* and *col* show the highest enrichment in the PSC followed by MZ1. iCC, immature crystal cell; mCC, mature crystal cell.

The online version of this article includes the following figure supplement(s) for figure 1:

**Source data 1.** Source data for *Figure 1D and G*, and *Figure 1—figure supplement 1D*.

**Figure supplement 1.** Resolving heterogeneity in sorted populations.

**Figure supplement 2.** Reproducibility and validity of scRNA-Seq results.

**Figure supplement 3.** Gene/pathway enrichment analysis of scRNA-Seq results.

in the MZ (*QC, mthl13,* and *Lst*) or the CZ (*Ten-m, edl,* and *Sr-CI*) population (*Figure 1D*). Future genetic analysis will determine how these genes function in their specified zones.

*Hml* is considered a hallmark gene for PLs, however, it is also expressed in CC precursors (*Goto et al., 2003*). This low *Hml^Δ^* is lost in CCs expressing very high *Hnt* (*Figure 1—figure supplement 1B-C'*). Therefore, the *Hml^Δ^-DsRed* population contains both PLs and CC expressing *lz, hnt* (*pebbled; peb*), *PPO1,* and *PPO2* (*Figure 1C*).

IPs do not express late differentiation markers such as *NimC1* or *PPO1/2*, which are characteristic of mature PLs and CCs, respectively (*Figure 1C*). Nor do they express significant levels of very early progenitor markers such as *Tep4* and *kn/collier*. IPs represent a transitional population between the MZ and the CZ, but the IPs are also, in themselves, a *bona fide* cell type, uniquely enriched in transcripts such as *MFS3, CG30090, lectin-24A, CG13482, Amy-p,* and *CG31821* as compared with the expression of these transcripts in either MZ or CZ (*Figure 1E*). The collective expression of these bulk RNA-Seq derived novel IZ-enriched transcripts proved crucial in specifying a group of cells as IZ in our subsequent scRNA-Seq analysis.

We next used a genotype, combining *Hml^Δ^-DsRed.nls* and *lz>mGFP* (*lz-GAL4, UAS-mGFP*), such that CCs are separable from PLs. For this second bulk RNA-Seq, we use late wandering third instar larvae (93–117 hr AEL) at which stage CCs are more abundant than in the mid-third instar. All other conditions remain the same. Within the *lz>mGFP* population, two clearly separable groups, expressing high GFP (GFP^HI^) and lower GFP (GFP^LO^) are evident (*Figure 1F*). As expected, a large number of DsRed-positive but GFP-negative cells are sorted into a different gate and these are the PLs (*Figure 1—figure supplement 1C*) that express the hallmark genes: *Hml, Pxn, Col4a1, vkg,* and *NimC1* (*Figure 1G*). They also do not express *PPO1* or *PPO2*. In contrast, both GFP^HI^ and GFP^LO^ cells express *PPO1/2* and therefore they are both CC populations.

PPO1/2 expression in GFP^HI^ cells is much higher than in GFP^LO^ CCs (*Figure 1G*). As PPO1/2 are maturity markers, this allows us to name the two GFP expressing CC classes as mature (mCC) and (relatively speaking) immature (iCC) CCs. PL-related genes are higher in iCCs than in mCCs but the pan-CC marker, *hnt*, is expressed at comparable levels in all CCs (*Figure 1—figure supplement 1D*).

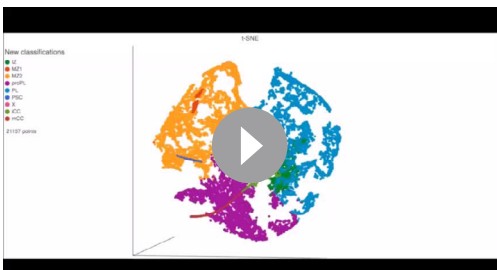

**Video 1.** Movie of three-dimensional (3D) t-SNE visualization of scRNA-Seq data showing the same nine lymph gland populations and color scheme shown in Figure 1J. In 3D, the IZ and proPL clusters are on separate planes that are largely non-adjacent to each other. However, both IZ and proPL each have adjacencies to the MZ2 and PL clusters.

https://elifesciences.org/articles/67516/figures#video1

*lz* RNA is also only marginally different between the two populations, although its surrogate, *lz>mGFP*, is readily distinguishable. As expected, both mCC and iCC contain cells with 2N and 4N DNA content (*Figure 1H*). However a subset of mCCs, but not iCCs, exhibits > 4N DNA content, indicative of endocycling (*Krzemien et al., 2010*; *Terriente-Felix et al., 2013*). These data suggest that endocycling is confined to the more mature, mCC subpopulation. We also find that the average forward scatter (FSC-A), a measure of cell size, and average side scatter (SSC-A), a measure of internal complexity, are higher in mCCs compared to iCCs (*Figure 1I*; *Figure 1—figure supplement 1E, F*). Thus, mCCs are larger, more mature, and more granular than iCCs.

## Single-cell RNA-Seq defines subzonal patterns within the lymph gland

Bulk RNA-Seq is a useful tool for identifying the broad gene expression landscape in a relatively large group of cells with previously established canonical biomarkers. To complement and enhance these data and to characterize subpopulations within each zone, we used single-cell RNA-seq (scRNA-Seq). The same genetic background and developmental timing (90–93 hr AEL at 25°C) are used to facilitate comparison between the two approaches. Each sample utilizes 11 lymph glands to yield a concentrated cell suspension with high (85–90%) cell viability. Three biological replicates are processed in parallel and the transcriptome of about 21,200 individual cells is determined using the 10× Genomics platform and analyzed using Partek Flow software (see Materials and methods). Graph-based clustering analysis and t-distributed stochastic neighbor embedding (t-SNE) visualization in two-dimensions (2D) and three-dimensions (3D) are then performed.

Nine individual cell clusters are predicted for the lymph gland (*Figure 1J*), and each of these populations is present in similar proportions in all three biological replicates (*Figure 1—figure supplement 2A, B*). Known zone-specific markers within the differentially expressed genes assist in the assignment of unique identities to the graph-based clusters (the cluster names are justified in later sections). The PSC and IZ are each represented by single clusters. We identify two clusters (MZ1 and MZ2) with progenitor characteristics. The data suggest that in addition to IZ, a second transitional population, proPL, straddles MZ2 and the PL cluster, PL (*Figure 1J*; more obvious in *Video 1*). As in bulk RNA-Seq analysis, subclustering of CCs splits them into two populations (iCC and mCC). Validating our assignment of cell-type identity, all of the above clusters express their respective previously identified zone-specific hallmark genes (*Figure 1K*; see complete gene list in *Supplementary file 1*). Please note that a subclustering algorithm was not used to generate MZ1 and MZ2 or PL and proPL. These are products of the basic graph-based clustering process. The names MZ1 and MZ2, for example, refer to their similarities to the historical name, MZ attributed to a zone containing progenitors. In contrast, the classification as PH1, PH2, and so on, for groups of cells by *Cho et al., 2020*, result from true subclustering (similar to iCC and mCC identified here as subclusters of CC). In our hands, sub clustering MZ1 leads to some very small groups of cells that are not distinguishable enough to classify as separate populations.

The cluster designated as 'X' on the t-SNE exhibits high levels of mitosis and replication stress-related genes. The PSC, CC, and X clusters are distinct enough from the rest to remain as islands distant from each other and the core group of the other cell populations. The similarities and gradual transitions between the rest of the cells (belonging to MZ1, MZ2, IZ, proPL, and PL) cause their clusters to be closely associated as a core group of neighbors on the t-SNE map (*Figure 1J*). This organization of the t-SNE is seen in all three biological replicates (*Figure 1—figure supplement 2A*). Despite adjacency on the t-SNE, each cluster is distinguished by differences in differential gene expression (*Supplementary file 1*) and gene set/pathway enrichment (*Figure 1—figure supplement 3A*).

Trajectory and pseudotime analysis are used to map the timeline of progression of the identified heterogeneous population of cells through their multiple phases of maturity (*Figure 2A–C*). This analysis allows further groupings within the major clusters. PSC is separate in developmental origin from the rest of the lymph gland (*Mandal et al., 2007*) and cluster X likely represents mitotic states of several distinct cell types and therefore, although these two populations are represented in the t-SNE, they are not included for the purpose of constructing the trajectory. We find that the lymph gland cells form a branched trajectory with a total of three branch points and seven states (*Figure 2C*). Mapping the states back onto the t-SNE (*Figure 2D*) allows visualization of individual paths between related clusters. The relatedness between clusters is often easier to discern on a 3D-tSNE (*Figure 2F*; *Video 1*).

## The PSC cluster

Known canonical PSC markers, such as *Antp*, *col*, and *dlp,* are all co-expressed at high levels in the PSC cluster (*Figure 1K*). Additional highly enriched genes include *Pvf1*, *Dad*, *Dif*, and *EGFR*, each of which has been shown to play a role in lymph gland development (*Mondal et al., 2011*; *Pennetier et al., 2012*; *Sinenko et al., 2011*; *Louradour et al., 2017*). *Delta* is expressed overall at low levels, but is enriched in the PSC with lower levels in MZ1 (*Figure 1K*). This pattern of expression is consistent with *Cho et al., 2020*. Additionally, we detect expression of *Delta* in mCCs.

The nature of the PSC has been extensively investigated prior to this study and GO terms related to many of the biological pathways such as TGF-β, Robo, and Wnt that are deemed important for PSC maintenance and function (*Krzemień et al., 2007*; *Mandal et al., 2007*; *Sinenko et al., 2009*; *Morin-Poulard et al., 2016*; reviewed in *Luo et al., 2020*) are enriched in the PSC cluster (*Figure 1—figure supplement 3A*). This serves as an independent validation of the RNA-Seq assisted assignment of genes for which genetics provides a specific function. Additionally, we focus later on a novel angle of the PSC cells that is related to their unique metabolic profile.

## MZ clusters

Both MZ1 and MZ2 express known hallmark genes as well as a number of genes that are newly identified as MZ-enriched in the bulk RNA-Seq experiments (*Figure 1K*; *Figure 2—figure supplement 1A*). MZ1 cells are found entirely at the beginning of the trajectory at the earliest pseudotime (*Figure 2A–C and G*) and all MZ1 cells are contained within state 1 of the trajectory (thus formally named, MZ1-1; *Figure 2D and G*). MZ1 represents a small number of cells ( 1.3% of total cells; 4.8% of total MZ cells) with higher expression levels of several progenitor markers (e.g., *shg*, *col/kn*, *dome*, and *Tep4*) compared with MZ2 (*Figure 1K*; *Figure 2—figure supplement 1B*). MZ1 does not neighbor CZ clusters (*Figure 1J*) and MZ1 cells are the earliest progenitors at this time point in development. Their similarities to PSC cells (see later), could indicate additional signaling function for this cluster.

MZ2 occupies three separate trajectory states (MZ2-1, MZ2-2, and MZ2-3; *Figure 2D and G*). A majority is in state 1 (MZ2-1, 68% of MZ2). MZ2-2 has very few cells ( 2% of MZ2), whereas MZ2-3 is of significant size ( 29% of MZ2). The levels of the progenitor markers *Tep4* and *dome* are highest in MZ1-1, they decrease in MZ2-1, and further decline in MZ2-2 and MZ2-3 (*Figure 2—figure supplement 1B*). Importantly, however, the *Tep4* and *dome* levels in MZ2-3 are still higher than those seen in the IZ and CZ-related clusters. In contrast, we find that the expression of IZ enriched genes, such as *CG30090* and *MFS3*, show the opposite trend compared to*Tep4* and *dome* (*Figure 2—figure supplement 1B*). Based on gene expression patterns, we propose that MZ2-2 and MZ2-3 are similar and represent a more mature population within MZ2. Together they represent a progenitor group of cells that mature to IZ, proPL, and PL.

MZ1 and MZ2 share 43 of the 241 differentially expressed MZ genes (*Supplementary file 1*). Thus MZ1 and MZ2 are closely related. However their identities are distinct. 144 of the MZ enriched genes are enriched in MZ1 and 54 are enriched in MZ2. More importantly, by multiple criteria, MZ1 and MZ2 show evidence of distinct biological functions. For instance, MZ1, but not MZ2, expresses some genes that are also found in the PSC, such as *dlp* and *kn/col* (*Figure 1K*). MZ1 is distinct from PSC as it lacks established markers such as *Antp* (*Figure 1K*). Surprisingly, *Ubx* expression seems to be a hallmark of MZ1 (*Figure 2—figure supplement 1B*). The expression level is low, and is similar to that seen for early PH2 progenitors in *Cho et al., 2020*. Please note that *Ubx* is also expressed in the tertiary lobe (*Rodrigues et al., 2021*) which was removed from our sample. Several glycolytic genes are expressed

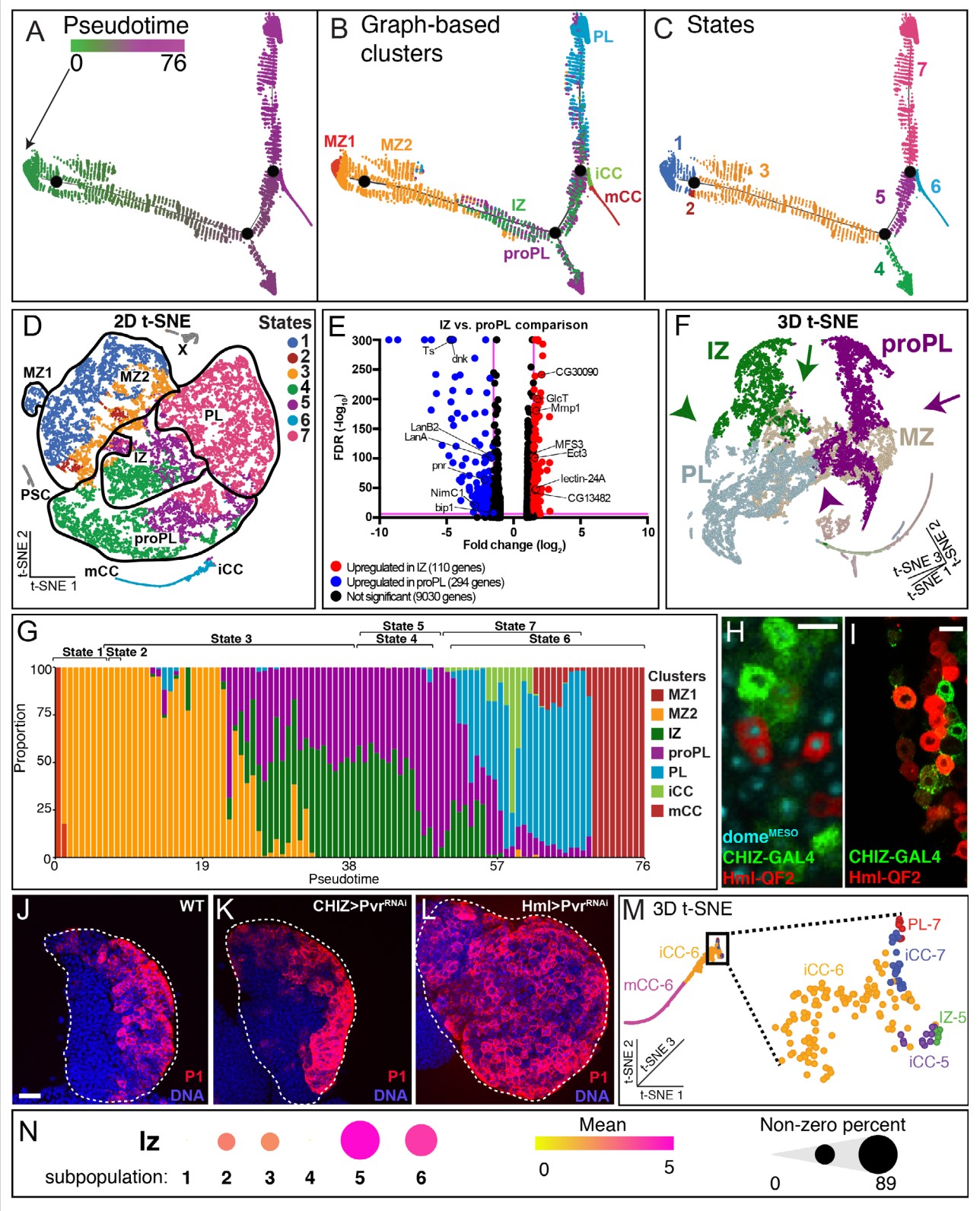

**Figure 2.** Developmental trajectory predicted by single-cell RNA-Seq data. (**A–C**) Trajectory diagram shows the progression of lymph gland cells from the earliest progenitors (left) to the most mature cell types. (**A**) Earlier points in pseudotime (green) progress to later (magenta) developmental stages. (**B**) Superposition of graph-based clusters onto the trajectory shows that MZ1 appears at the beginning of the trajectory, and proceeds through MZ2, IZ/proPL, and ultimately onto the differentiated cell types iCC, mCC, and PL. (**C**) Trajectory diagram showing seven states, each separated by branch points

*Figure 2 continued on next page*

*Figure 2 continued*

of the trajectory. State 1 (MZ1 and early MZ2), state 2 (late MZ2), state 3 (late MZ2, IZ, and proPL), states 4 and 5 (IZ and proPL), state 6 (terminal CCs), and state 7 (terminal PLs). (**D**) Trajectory states overlaid on the 2D t-SNE plot. State 1 (dark blue) contains all MZ1 cells as well as MZ2 cells adjacent to MZ1. State 2 (dark red) is a very minor component of MZ2. State 3 (orange) lies at the border between MZ2 progenitors and the more developmentally advanced clusters (IZ, proPL, and PL). States 4 (green) and 5 (purple) represent the majority of IZ and proPL cells. State 6 (light blue) contains the majority of CCs. State 7 (pink) contains most PL cells as well as IZ and proPL cells adjacent to the PL cluster. A small number of iCC cells are in states 5 and 7, which map to the tip of the CC cluster (see **M**). PSC and X cells are colored gray as they were not used for trajectory analysis. (**E**) Volcano plot depicting the results of an ANOVA comparison between the IZ and proPL clusters. Each gene expressed in the scRNA-Seq data is represented by a dot. The X-axis depicts the magnitude of the difference in expression of each gene in IZ compared to proPL. The Y-axis indicates the statistical significance of each difference in gene expression, the false discovery rate (FDR) value for each comparison, where magenta lines represent the significance thresholds beyond which the difference in gene expression is statistically significant. IZ upregulated genes (red); proPL upregulated genes (blue); Selected statistically significant genes (black labels). (**F**) 3D t-SNE emphasizes that IZ (green) and proPL (purple) are distinct clusters that show little adjacency to each other because of their different 3D-planar locations. IZ and proPL possess separate connections between MZ (tan) (compare green arrow vs. purple arrow) and PL (gray) (compare green arrowhead vs. purple arrowhead). The data strongly suggest that IZ and proPL are two distinct means to connect MZ with PL. For a better view of the 3D-tSNE, see **Video 1**. (**G**) Representation of the fraction of cells from individual clusters at each pseudotime point. MZ1 (red) is found at the earliest pseudotime, while MZ2 (orange) is found in states 1, 2, and 3, slightly later in pseudotime. IZ (green) and proPL (purple) are placed primarily at intermediate pseudotime in states 3, 4, and 5. The placement of the states relative to pseudotime is indicated at the top of the graph and reveal that states 4 and 5 overlap considerably in pseudotime, indicating that spatially distinct clusters can overlap in pseudotime. Similarly, states 6 and 7 overlap and include a number of cell types. (**H**) Image of an early third instar lymph gland expressing *CHIZ-GAL4, UAS-mGFP* (green) to mark IZ cells, and *dome^MESO^-EBFP2* (cyan) with *Hml^Δ^-QF2, QUAS-mCherry* (red) to mark proPL cells. IZ and proPL cells display distinct non-overlapping expression patterns. proPL cells can be distinguished here from PLs (which also express *Hml^Δ^-QF2, QUAS-mCherry*) by their expression of *dome^MESO^-EBFP2*. Scale bar, 10 µm. Full lobe shown in **Figure 2—figure supplement 2C**. (**I**) *CHIZ-GAL4, UAS-mGFP* (green) and *Hml^Δ^-QF2, QUAS-mCherry* (red) are expressed in distinct cells of the lymph gland with little colocalization. Scale bar, 10 µm. Full lobe shown in **Figure 2—figure supplement 2D**. (**J–L**) Loss of equilibrium signal with *Pvr^RNAi^* increases P1 staining when driven with *Hml^Δ^-GAL4* (**L**) but not with *CHIZ-GAL4* (**K**) compared to wild-type (**J**). Scale bar, 20 µm. Quantifications found in **Figure 2—figure supplement 2E**. (**M**) A magnified view of the CC island from the 3D t-SNE. The boxed area is further magnified to show the identity of individual cells. iCC-6 and mCC-6 make up the majority of CCs. However, the base of the CC island (boxed part) includes small populations of iCC-5 and iCC-7 in close proximity to IZ-5 and PL-7, respectively, from which they are derived. (**N**) Dot plot showing the expression pattern of *lz* (*lozenge*) in iCCs and their immediate IZ-5 and PL-7 neighbors compared with IZ-5 and PL-7 cells that are not on the CC island. The 'subpopulations' are as follows: 1. IZ-5 cells not on the CC island. 2. IZ-5 cells on the CC island. 3. iCC-5 cells. 4. PL-7a/b cells not on CC island. 5. PL-7a/b cells on the CC island. And 6. iCC-7 cells. In each population, dot color reflects the mean level of *lz* expression (mean) and the dot size indicates the percentage of cells that express *lz* (non-zero percent). The data show that the IZ and PL cells that map to the CC island are enriched for *lz* unlike the rest of the IZ and PL cells. CZ, cortical zone; iCC, immature crystal cell; IZ, intermediate zone; mCC, mature crystal cell; PL, plasmatocyte; t-SNE, t-distributed stochastic neighbor embedding.

The online version of this article includes the following figure supplement(s) for figure 2:

**Source data 1.** Source data for *Figure 2E* and *Figure 2—figure supplement 2E*.

**Figure supplement 1.** Cluster characteristics for MZ1, MZ2, X, and IZ.

**Figure supplement 2.** Expression profiling of proPL and PL.

**Figure supplement 3.** Crystal cell (CC) genes and comparative enrichment of genes in differentiating cells.

---

at much higher levels in MZ1 than in MZ2 suggesting distinct metabolic requirements (explored later in *Figure 3*).

Another distinct biological difference between MZ1 and MZ2 involves the expression of specific immunity-related genes. For example, MZ2 (but not MZ1) progenitors are enriched for all four Cecropin genes (*CecA1*, *CecA2*, *CecB*, and *CecC*) (*Figure 1—figure supplement 3A*; *Figure 2—figure supplement 1C*), which are involved in antibacterial humoral response downstream of the Toll and Imd pathways. Other Imd-related genes such as *PGRP-SC2*, and Toll pathway genes such as *grass*, are also enriched in MZ2 compared to MZ1 (*Figure 1—figure supplement 3A*; *Figure 2—figure supplement 1C*). Overall, these results indicate that the slightly more mature MZ2 cells might be better poised to respond to immune challenge than MZ1 cells, while MZ1 cells are the least mature and are more likely to respond to different metabolic and local signaling cues.

## Cluster X

X is a very small cluster of cells (~ 1% of total) with a rather unique genomic composition. While most zones, which are much larger than X, are enriched for approximately 100–200 genes, ANOVA analysis suggests that for X this number is over 2000 (*Supplementary file 1*). X represents a mitotic component of the lymph gland and likely includes cells from multiple zones. Cell cycle-related proteins

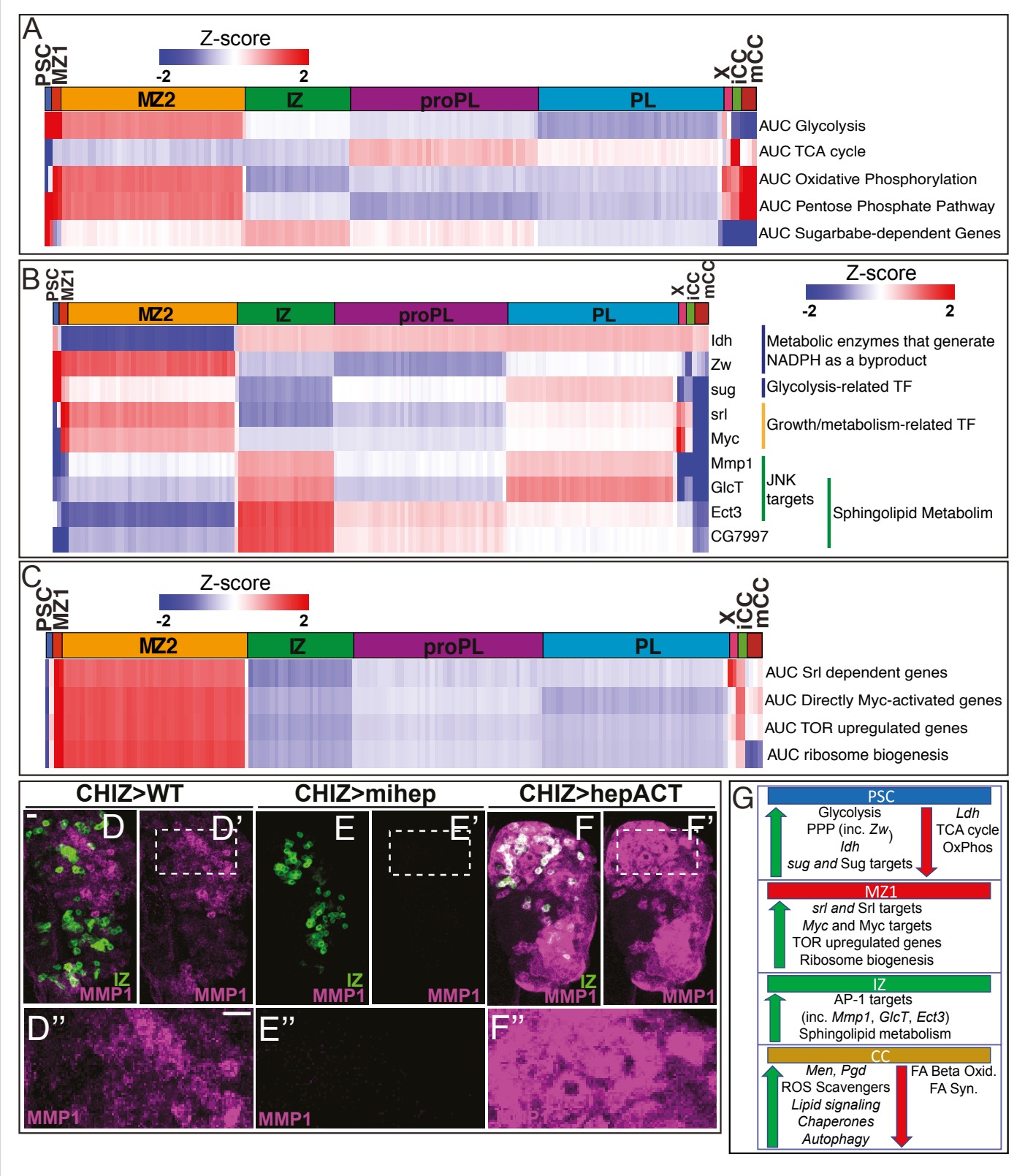

**Figure 3.** Developmental metabolism of the lymph gland by single-cell analysis. Expression analysis of either single genes (**B**) or groups of genes scored as AUCell (**A, C**) displayed in heatmaps with z-scores ranging from –2 (blue) to 2 (red). (**A**) Glycolysis pathway: enriched in PSC (z-score=5.7; corresponds to p<0.0001) and MZ1 (z-score=2.5; p=0.006). Other clusters: not statistically significant (n.s.). TCA cycle enzymes: not enriched in PSC (z-score=–4.1; p<0.0001). Other clusters: n.s. Oxidative phosphorylation: not enriched in PSC (z-score=–2.2; p=0.015), slightly enriched in MZ1 (z-score=2.0; p=0.026).

*Figure 3 continued on next page*

*Figure 3 continued*

Other clusters: n.s. Pentose phosphate pathway (PPP): enriched in PSC (z-score=2.4; p=0.008), slightly enriched in MZ1 (z-score=1.9; p=0.02), and mCC (z-score=2.6; p=0.004). Other clusters: n.s. Sugarbabe-dependent genes: enriched in PSC (z-score=3.4; p=0.0004). Other clusters: n.s. (**B**) Heatmap showing expression of assorted metabolic genes. PSC is enriched for *Zw*, *Idh*, and *sug*. MZ1 is enriched for *srl* and *Myc*. IZ is enriched for *Mmp1*, *GlcT*, *Ect3*, and *CG7997*. (**C**) Heatmap of AUCell analysis of growth-related pathways. MZ1 is slightly enriched in: *srl* dependent genes (z-score=2.06; p=0.020), and enriched for Myc target genes (z-score=2.44; p=0.007), TOR upregulated genes (z-score=2.64; p=0.004), and ribosome biogenesis genes (z-score=2.40; p=0.008). (**D–F″**) Manipulation of the JNK pathway using *CHIZ-GAL4, UAS-mGFP* (green) that marks IZ cells (*Spratford et al., 2020*). Immunolocalization of MMP1 is shown in magenta. Images are maximum intensity projections of the middle third of a confocal z-stack of lymph glands from wandering third instar larvae. The regions boxed in (**D′–F′**) are shown at a higher magnification in (**D″–F″**). Scale bars, 10 µm. (**D–D″**) MMP1 protein expressed in close proximity to IZ cells in wild-type (WT). (**E–E″**) A microRNA-based depletion of *JNKK/hep* in the IZ cells results in loss of MMP1 throughout the lymph gland (including in the region neighboring the IZ). (**F–F″**) Overactivation of JNK by a constitutively active isoform of JNKK/hep causes a large increase in MMP1 staining throughout the lymph gland. (**G**) Summary of the metabolic gene expression signatures found in the PSC, MZ1, IZ, and CC clusters. Green (up) arrows indicate enrichment of genes/pathways while red (down) arrows indicate the absence of enrichment. CC, crystal cell; IZ, intermediate zone; MZ, medullary zone.

The online version of this article includes the following figure supplement(s) for figure 3:

**Figure supplement 1.** Unique metabolic signatures of the PSC and CCs.

**Figure supplement 2.** Metabolic signatures of the MZ and IZ.

are enriched in cluster X (*Figure 1—figure supplement 3A*), and the five most represented genes are regulators of cell cycle (reviewed in *Lee and Orr-Weaver, 2003*; *Berridge, 2014*) in S and G2: *cdc25/stg* (30-fold), *cdt1/dup* (11-fold), *Mcm5* (10-fold), *Claspin* (9-fold), and *dap/p21* (9-fold). Also enriched are genes involved in DNA replication, cell division, spindle checkpoint, mitotic spindle, and kinetochores (*Figure 1—figure supplement 3A*). AUCell analysis, another tool for analyzing the enrichment of specific gene sets (see Materials and methods), further suggests that Cluster X shows high levels of 'mitotic G2/M transition' activity (*Figure 2—figure supplement 1D*). Interestingly, single-cell transcriptomic study of the human HSC/HSPC cells from the bone marrow (*Velten et al., 2017*) also found small high cell cycle activity clusters with characteristics similar to X.

Another characteristic of X is that it is the only cluster to include replication stress-related intra-S DNA damage checkpoint genes (*Figure 2—figure supplement 1E*). This is a characteristic of replication fork formation in transcriptionally active cells (*Lee et al., 2012*; *Blythe and Wieschaus, 2015*; *Iyer and Rhind, 2017*) and 'replication stress' is a means to control the progression of the cell cycle (*Berti and Vindigni, 2016*; *Zou and Nguyen, 2018*). In the lymph gland, the MZ cells are prolonged in their G2 state (*Sharma et al., 2019*) and it is attractive to hypothesize that replication stress-related S-phase events are, at least in part, responsible for the slow-down of the subsequent G2. Importantly, the enrichment of DNA damage-related genes in cluster X is not due to cells of this cluster being generally damaged or dying as quality control metrics such as percentage mitochondrial and ribosomal reads are on par with the other clusters (*Figure 1—figure supplement 2C, D*).

Finally, 11 out of the 12 members of the Myb complex (Myeloblastosis oncoprotein family), including the *Myb* transcription factor itself, are highly enriched in Cluster X (*Figure 1—figure supplement 3A*). Myb regulates DNA damage checkpoints and participates in the DNA repair process in cancer cells (*Yang et al., 2019*) with an established role, as well, in *Drosophila* blood cells (*Davidson et al., 2005*) and is similar to that in mammalian hematopoietic cells (*Greig et al., 2008*).

Cluster X is physically separated from the larger core group of clusters on the t-SNE and although small, the X cluster itself is separated into two islands (*Figure 1J*). One of the X islands (~30%) is physically closer to the MZ, while the larger one (~70%) maps in the direction of the PL cluster. The smaller island is more MZ-like (X$^{MZ}$) and the larger is more CZ-like (X$^{CZ}$) in gene expression (*Figure 2—figure supplement 1F*). Subclustering of X leads to three separate groups, one which corresponds directly to X$^{MZ}$ and the other two represent a split of X$^{CZ}$ into two subclusters, X$^{TR}$ (transitional) and X$^{PL}$ (PL-like) (*Figure 2—figure supplement 1F, G*). These subclusters show different patterns when profiled for hallmark zone-specific genes and markers. The clear variation in the expression of MZ and CZ genes within X provides the basis for the designation X$^{MZ}$, X$^{TR}$, and X$^{PL}$ for the three subclusters. When a trajectory diagram that only includes cells from X is constructed, X$^{MZ}$ is earliest in pseudotime and X$^{PL}$ is the latest (*Figure 2—figure supplement 1H, I*). X$^{TR}$ represents a mid-point transition into the PL state. This variation in the three X subclusters is limited to the expression of the MZ and PL hallmark

genes. In contrast, cell cycle and DNA damage-related genes that are enriched in X relative to the other clusters are equally represented in $X^{MZ}$, $X^{TR}$, and $X^{PL}$ (*Figure 2—figure supplement 1J*).

## IZ cluster

In the bulk RNA-Seq experiment, 10 genes were identified as enriched in the double positive IZ cells. Six of these 10 markers are expressed at levels that are detected in the scRNA-Seq. Of these six, five are enriched in the IZ cluster (*Figure 2—figure supplement 1K*; *Supplementary file 1*). The top IZ-enriched markers continue to be *CG30090, lectin-24A, MFS3,* and *CG13482* (*Figure 1E and K*; *Figure 2—figure supplement 1K*). Single-cell analysis also confirms that the IZ cluster expresses lower levels of canonical CZ markers compared to the PL clusters and does not express mature blood cell markers (*Figure 1K*). On the t-SNE, the IZ cluster lies between MZ2 and PL (*Figure 1J*), consistent with its intermediate nature between progenitors and differentiated cells. Although IZ cells form a single cluster, they are found in multiple trajectory states, IZ-3, IZ-4, IZ-5, and IZ-7 that lie between MZ2 and PL/CC on the trajectory and in pseudotime (*Figure 2A–D and G*). IZ-3 borders and represents a step immediately after MZ2-3 (*Figure 2B and D*). Whereas, IZ-5 cells are either near CC clusters or placed between IZ-4 and IZ-7 (*Figure 2D and M*). IZ-7 is at the border with PL and therefore represents a transition to committed PLs (*Figure 2D*). The most parsimonious model is that IZ-5 cells have the capacity to directly become CCs, or alternatively, via IZ-7, they take on a PL fate. Finally, IZ-independent paths to mature blood cells are also observed that are described later.

## proPL and PL clusters

Two separate clusters (proPL and PL) identified in scRNA-Seq both enriched for hallmark PL genes and neither cluster expresses CC-specific genes (*Figure 1K*). Expression of several PL markers is slightly lower in proPL than in PL (*Figure 1K*), and proPL appears earlier in pseudotime than PL (*Figure 2A–C and G*). In addition to such quantitative temporal differences in the expression of known markers, the distinction between PL and proPL clusters is highlighted by the differential expression of at least 750 genes that are represented differently between the two clusters (*Supplementary file 1*).

## proPL cluster

The proPL population arises in multiple states on the developmental trajectory appearing first in state 3 (proPL-3), followed by proPL-4/5/7 (*Figure 2A–D and G*). The proPL subclasses differ from each other in their placement in pseudotime (*Figure 2G*), and in the expression levels of multiple mitosis- and maturation-related genes (*Figure 2—figure supplement 2A, B, J*).

As described earlier for IZ, placement of a group of cells in multiple trajectory states is a characteristic of a transitional population that can represent multiple distinct developmental paths. The IZ and proPL cells belong to distinct cell clusters with clear transcriptomic differences, and are largely non-adjacent on the t-SNE; this fact is easier to discern in a 3D t-SNE representation (*Figure 2F*; better seen in *Video 1*). This contrasts with the extensive direct adjacencies observed for both proPL and IZ with MZ2 and PL (*Figure 1J*; *Figure 2F*; *Video 1*). Both IZ and proPL cells are found in similar transitional states (3–5) on the trajectory and arise at similar points in pseudotime, although their pseudotime profiles are distinct (*Figure 2G*).

IZ and proPL are both transitional populations with some similarities in the scRNA-Seq, but can be genetically distinguished from each other in vivo by using a combination of the Q and GAL4 systems. In a genetic background that includes $Hml^{\Delta}$-QF/QUAS (*Lin and Potter, 2016*) and CHIZ-GAL4/UAS (*Spratford et al., 2020*), the IZ is marked by CHIZ-GAL4 while $Hml^{\Delta}$-QF marks proPL and PL. CHIZ-GAL4 is a split GAL4 construct that contains a combination of $Hml^{\Delta}$ and $dome^{MESO}$ enhancers and includes the strong p65 activation domain. CHIZ-GAL4 expression overlaps with cells that are double positive for the directly driven markers $dome^{MESO}$-GFP and $Hml^{\Delta}$-DsRed (*Spratford et al., 2020*). The percentage of cells marked by CHIZ-GAL4 (~ 17% at 96 hAEL) (*Spratford et al., 2020*) is consistent with the size of the IZ cluster defined by scRNA-Seq (~14–16% in individual replicates). Early proPL cells can be identified by co-expression of $Hml^{\Delta}$-QF and $dome^{MESO}$-EBFP2 (*Figure 2H*; *Figure 2—figure supplement 2C*). $Hml^{\Delta}$-QF has an identical expression pattern as the $Hml^{\Delta}$-GAL4 from which it is derived (*Lin and Potter, 2016*) but shows little overlap with CHIZ-GAL4 (*Figure 2H–I*; *Figure 2—figure supplement 2C, D*). Why the $Hml^{\Delta}$-QF/$Hml^{\Delta}$-GAL4 constructs exhibit a more restricted expression pattern that does not include IZ cells, while the directly driven $Hml^{\Delta}$-DsRed is expressed at low

levels in the IZ is unclear, but is likely due to differences in timing or level of expression. Nevertheless, the use of this complex genetic background allows simultaneous detection of IZ and proPL cells in the same lymph gland (*Figure 2H–I*; *Figure 2—figure supplement 2C, D*), and these two populations are largely distinct.

Direct comparison between IZ and proPL shows that they differ significantly in their expression of many genes (*Figure 2E*). Another important distinction between IZ and proPL is that proPLs (proPL-4 in particular), but not IZ cells, show high AUCell activity for a diverse set of genes that are collectively identified as participants of the backward or equilibrium signal (*Pvr*, *STAT92E*, *ADGF-A*, *bip1*, *RPS8*, and *Nup98-96*) (*Figure 2—figure supplement 2A, B*). Past genetic analysis had indicated that the CZ initiates the equilibrium signal via ADGF beginning in the second instar (*Mondal et al., 2011*). Consistent with this finding, we find the loss of equilibrium signaling with *Pvr^{RNAi}* in *Hml^Δ-GAL4+* cells results in increased differentiation, while *Pvr^{RNAi}* in *CHIZ-GAL4+* IZ cells does not cause a similar phenotype (*Figure 2J–L*; *Figure 2—figure supplement 2E*). These genetic results, when combined with the data that show that the backward signal genes show higher AUCell activity in proPL relative to IZ and PL, suggest that it is the proPL, and not the IZ or PL, cells that largely participate in the backward signaling to progenitors in vivo. This functional difference, taken together with the trajectory and gene enrichment results, further support the notion that proPL and IZ are two separate transitional populations that follow distinct paths as they mature towards the same set of differentiated cells.

## PL cluster

In contrast to the multiple pseudotime states of the proPL cells, virtually all cells of the PL cluster (>99%) are seen exclusively in state 7 (PL-7) at the terminal arm of the trajectory (*Figure 2A–D*). Accordingly, the highest transcript levels of the mature PL marker, *NimC1*, are seen in PL-7 (*Figure 1K*).

The high *NimC1* expressing PL-7 can be further subclustered into four smaller groups that we named PL-7a, PL-7b, PL-7c, and PL-7d (or PL-7a/b and PL-7c/d for convenience; *Figure 2—figure supplement 2F*). PL-7a/b cells have lower *NimC1* than PL-7c/d. In fact, the *NimC1* levels of PL-7a/b are more similar to that in the transitioning proPL-5/7 cells than in the mature PL-7c/d (*Figure 2—figure supplement 2I*). By far, the highest *NimC1* levels are reserved for PL-7c/d located along the edge of the t-SNE (*Figure 2—figure supplement 2F, I*). Pseudotime analysis shows that PL-7a/b arise earlier in development than PL-7c/d (*Figure 2—figure supplement 2G*). We conclude that PLs follow a maturation path from the proPL-5/7 to PL-7a/b and then to the most mature PL-7c/d cells.

A very small fraction of PL cells arise earlier as PL-3 (*Figure 2G*). They are represented on the t-SNE as a very thin but distinctive border separating MZ2-3 and PL-7 (*Figure 2—figure supplement 2F*). PL-3 (and some adjacent cells) express unexpectedly high amounts of the progenitor marker *E-Cad/shg* (*Figure 2—figure supplement 2H*), as do the PL-7 cells in the vicinity (PL-7a and PL7d). Normally, *E-Cadherin* expression is a characteristic of MZ, with its expression declining in IZ/proPL and PL clusters (*Figure 1K*). The placement of the PL-3 subpopulation on the t-SNE, the trajectory, and its high *E-Cad* level may suggest that PL-3 is derived directly from the MZ progenitors.

## CC clusters

The cells of the CC cluster are identified by the high expression of canonical CC markers and by the complete absence of *NimC1* (*Figure 1K*). Additionally, transcripts encoding factors that respond to stress, heat, and unfolded proteins are enriched in CCs (*Figure 1—figure supplement 3A*). This is consistent with genetic data on CCs as mediators of stress response (*Sorrentino et al., 2002*; *Cho et al., 2018*; *Miller et al., 2017*). Two subclusters of CCs, iCC and mCC, predicted by flow cytometry and bulk RNA-Seq experiments, are also distinguishable in scRNA-Seq (*Figure 1J*) by their differential expression of the maturation markers *PPO1* and *PPO2* (*Figure 2—figure supplement 3A*). Transcripts for *lz* and *hnt* are also higher in mCC than in iCC, while *Hml* shows the opposite trend (*Figure 2—figure supplement 3A*). Also, *lz* and *hnt* correlate positively with *PPO2* in both iCCs and mCCs, whereas *Hml* correlates negatively, especially strongly in mCCs, with *PPO2* (*Figure 2—figure supplement 3B-D*). These results further reinforce that the two subclusters of CCs represent an immature (iCC) and a mature (mCC) population as has been seen in recent transcriptomic studies (*Cho et al., 2020*; *Tattikota et al., 2020*).

All mCCs and the vast majority of iCCs (~95%) belong to terminal state 6 (mCC-6 and iCC-6) (*Figure 2A–D and G*), which is the dedicated CC arm of the trajectory, but a small (~5%) fraction

of iCCs are also found in states 5 (iCC-5) and 7 (iCC-7) (*Figure 2G and M*). iCC-5 and iCC-7 represent distinct developmental paths in the formation of iCCs from their precursors. On the t-SNE, the broader tip of the CC island bifurcates and one arm contains iCC-5 and their adjacent small number of IZ-5 cells, while the other arm includes iCC-7 adjacent to a few PL-7 cells (*Figure 2M*). Interestingly, these IZ-5 and iCC-5 cells share expression of *CG30090* (*Figure 2—figure supplement 3E, F*) and *lz* (*Figure 2N*). Similarly, these PL-7 and iCC-7 cells both express *NimC1* (*Figure 2—figure supplement 3E, G*) and *lz* (*Figure 2N*). These results indicate that these IZ-5 and PL-7 cells are the precursors to iCC-5 and iCC-7, respectively. Both of these iCC populations then transition to iCC-6 and further to mCC-6. The IZ-5/iCC-5/iCC-6/mCC-6 path represents the straight-forward maturation of CCs. The PL-7/iCC-7/iCC-6/mCC-6 path, on the other hand, has a step that is a reversal in pseudotime. This likely represents the phenomena of dedifferentiation or transdifferentiation of CCs from PLs, which are supported by genetic data (*Terriente-Felix et al., 2013*; *Leitão and Sucena, 2015*).

## Comparative gene enrichment in differentiating cells

To better understand major genetic components that control similarities and differences between the cells of IZ, proPL, PL, and CC, we performed gene enrichment analysis (*Figure 1—figure supplement 3A*).

An important role of differentiated hemocytes is the formation and secretion of ECM/BM components (*Tepass et al., 1994*; *Martinek et al., 2008*; reviewed in *Fessler and Fessler, 1989*; *Pastor-Pareja, 2020*). Surprisingly, genes related to ECM/BM such as *vkg*, *Col4a1*, *SPARC*, *Laminins* (*A*, *B1*, and *B2*), and *Tiggrin* are enriched not only in the PL but also in the transitory proPL and IZ populations (*Figure 1—figure supplement 3A*; *Figure 2—figure supplement 3H*). This is also true for genes involved in hydroxyproline production required for Collagen formation, such as *PH4αEFB*, *Pdi*, and *Plod*, which positively correlate with *Col4a1* expression (*Figure 2—figure supplement 3H, I*). However, we found that genes involved in secretion of Collagen and ECM/BM proteins are not enriched in IZ/proPL as they are in PL (*Figure 1—figure supplement 3A*; *Figure 2—figure supplement 3H*). We conclude that while ECM/BM proteins initiate their expression and maturation in the transitory IZ/proPL cells, the secretory mechanism for these proteins likely becomes fully functional only at the PL stage.

As expected, components of common signaling pathways, known for their context-dependent function are not zone-specific, with some more broadly represented than others. The Ras/MAPK pathway, for example, is enriched in multiple zones (IZ, proPL, and PL), and is particularly not enriched in CCs (*Figure 1—figure supplement 3A*; *Figure 2—figure supplement 3H*). The Notch pathway shows the opposite pattern with enrichment in CCs compared to IZ, proPL, and PL (*Figure 2—figure supplement 3H*). It could be argued that while hallmark genes make good markers, the distributed ones may contain more developmental information, for example, the above trends suggest that the Notch and MAPK pathways oppose each other in the choice between CCs and PLs, respectively (see case studies later).

More novel and surprising is the finding that genes that belong to prominent metabolic pathways are enriched in the PSC. This prompted us to investigate if key metabolic pathways play unique zone-specific roles in the lymph gland.

## Gene enrichment of metabolic pathways

Complete functional insight into the role of metabolism in lymph gland development will require metabolomic analysis, which is beyond the scope of this study. However, much can be gleaned from transcriptomic data since multiple components of any single metabolic pathway are often co-regulated by common transcription factors.

## Glycolysis and TCA cycle

Glycolysis-related genes are significantly enriched in the PSC (AUC scores: *Figure 3A*; individual genes: *Figure 3—figure supplement 1A, B*). The gene *sugarbabe* (*sug*) encodes a transcription factor that regulates multiple glycolysis and gluconeogenesis-related genes. *sug* and its known downstream targets are highly enriched in the PSC (*Figure 3A–B*) and they exhibit a strong positive correlation with glycolytic gene expression (*Figure 3—figure supplement 1E*). In contrast, TCA cycle and oxidative phosphorylation-related genes are particularly not enriched in the PSC (*Figure 3A*). However, it is

very unlikely that the bioenergetic requirements of the PSC are maintained through aerobic glycolysis (Warburg effect) as in cancer cells (reviewed in *Liberti and Locasale, 2016*; *Drosophila* example: *Wang et al., 2016*) because the transcript for lactate dehydrogenase (*Ldh*), the enzyme involved in the last step of glycolysis is not expressed in the PSC (*Figure 3—figure supplement 1B*). Combined with the low expression of TCA and Ox-Phos genes, we conclude that the PSC has a very low bioenergetic requirement that is characteristic of quiescent post-mitotic cells. This is further supported by the fact that the percentage of mitochondrial reads is lower in the PSC compared to other clusters (*Figure 1—figure supplement 2C*).

## Pentose phosphate pathway

If not for energy generation, what could be the need for the high expression of glycolytic genes (other than *Ldh*) in the PSC? The evidence points to the importance of pentose phosphate pathway (PPP), the biosynthetic arm of glucose metabolism (*Stincone et al., 2015*). PPP-related genes are enriched in the PSC and MZ1 (*Figure 3A*). The absence of enrichment of oxidative phosphorylation genes in the PSC and their relatively higher levels in the MZ (*Figure 3A*) points to a higher bioenergetic status for the progenitors than that of the PSC. However, while increased mitochondrial activity facilitates ATP generation, it would also potentially raise reactive oxygen species (ROS) levels in the MZ.

## NADPH and ROS

G6PD (Zw), the PPP component enzyme that catalyzes the first reaction in the PPP produces NADPH, a crucial metabolite that maintains glutathione in its reduced form (GSH), which in turn acts as a scavenger of intracellular ROS (*Ying, 2008*; *Fan et al., 2014*; *Lewis et al., 2014*; *Kuehne et al., 2015*). Unlike NADH, NADPH is produced by only a handful of enzymes, the most prominent being isocitrate dehydrogenase (*Idh*) (*Geer et al., 1979a*; *Geer et al., 1979b*). Both *Zw and Idh* are enriched in the PSC (*Figure 3B*; *Supplementary file 1*), and together they would raise NADPH, facilitating GSH formation and maintaining a low ROS level in the PSC. The lower *Idh and Zw* expression in the MZ suggests lesser scavenging of physiological ROS content, which has interesting biological correlates from past genetic studies (see Discussion).

Additional important enzymes involved in NADPH generation are malic enzyme (*Men*) and phosphogluconate dehydrogenase (*Pgd*) (*Geer et al., 1979a*; *Geer et al., 1979b*; reviewed in *Stanton, 2012*). The CCs, by far, show the highest expression for both *Men* and *Pgd*, with a considerable increase from iCCs to mCCs (*Figure 3—figure supplement 1C*). Presumably, ROS is kept particularly low in the mCCs to prevent premature JNK activation, which is known to promote CC bursting and melanization (*Bidla et al., 2007*). ROS may also be kept in check by the expression of antioxidants, such as *Sod1*, *Catalase* (*Cat*), *Jafrac1*, and *Trx-2,* that are higher in mCCs than in iCCs (*Figure 3—figure supplement 1C*). Of these, *Trx-2* mutations have been shown to cause CC defects (*Jin et al., 2008*).

## Lipids, autophagy, and chaperones

The iCCs and mCCs are also different in additional metabolic aspects. For instance, both peroxisomal and mitochondrial fatty acid beta oxidation, as well as fatty acid synthesis genes decrease in relative expression in mCCs compared to iCCs (*Figure 3—figure supplement 1D*), whereas glycerolipid remodeling/lipid signaling genes (e.g., *Bbc*, *Pld*, *Lpin*, *laza*, *Plc21C*, *GK2*, *sws*, and *CG10602*) are highly enriched in mCC (*Figure 3—figure supplement 1D*).

Autophagy related genes (such as *Atg1*, *Atg13*, and *Atg17*) and pathways indirectly related to autophagy (reviewed in *Soto-Avellaneda and Morrison, 2020*; *Carra et al., 2010*; *Kaushik and Cuervo, 2012*; *Uytterhoeven et al., 2015*) are strongly enriched in mCCs relative to iCCs (*Figure 3—figure supplement 1D*). This includes glycerolipid remodeling/lipid signaling and chaperone-mediated protein folding (e.g., Hsc70-4, Hsp67Bc, and Hsp70Bb) (*Figure 3—figure supplement 1D*), both of which correlate strongly (r=0.99 and 0.95, respectively) with autophagy genes (*Figure 3—figure supplement 1F, G*). Future genetic explorations will likely unravel the precise link between lipid signaling, chaperone-mediated autophagy, and the maturation of CCs.

## Transcription factors in metabolic control

Among transcription factors that control metabolism-related genes, Spargel (*srl*; PGC1-α) and its targets are enriched in MZ1 (*Figure 3B–C*). *Srl*, a homolog of mammalian PGC1-α, is a transcriptional target of Myc and both Srl and Myc function downstream of the insulin receptor/TOR signaling pathways to mediate ribosome biogenesis, mitochondrial activity, and cell growth (*Tiefenböck et al., 2010*; *Mukherjee and Duttaroy, 2013*; *Mukherjee et al., 2014*; *Teleman et al., 2008*). *Myc* and its transcriptional targets, as well as TOR upregulated genes, and those related to ribosome biogenesis are all enriched in MZ1 (*Figure 3B–C*). These trends are similar in MZ2 when compared to the other clusters. In a related observation by direct comparison of cell size by FSC, we find that MZ progenitors are on average larger in size than the cells of the CZ (*Figure 3—figure supplement 2A*), which is consistent with the higher growth-promoting pathway activity within the MZ.

## Sphingolipid metabolism

The IZ cells frequently express intermediate levels (between MZ and CZ) of most metabolic pathway genes, with the prominent exception of sphingolipid metabolism that is enriched in the IZ (*Figure 1—figure supplement 3A*). This further reinforces the independent cell-type identity of the IPs. For example, the gene encoding the rate-limiting enzyme for de novo ceramide synthesis pathway (*spt2/lace*; *Kraut, 2011*) is enriched in the IZ (*Figure 1—figure supplement 3A*; *Supplementary file 1*). This enzyme helps convert palmitoyl-CoA and serine to ceramide. AUCell scores for the entire de novo ceramide synthesis pathway are higher in the IZ when compared to other clusters (*Figure 3—figure supplement 2B*). Excess ceramide is toxic and is kept in check by enzymes of the glycosphingolipid pathway (*Kohyama-Koganeya et al., 2004*). Such genes include *GlcT, Ect3/Beta-Gal*, and *CG7997/alpha-Gal* that are also enriched in the IZ (*Figure 1—figure supplement 3A*; *Figure 3B*; *Supplementary file 1*).

## Ceramide and JNK activation

Ceramide production is linked to JNK activation in *Drosophila* and in other organisms (*Adachi-Yamada et al., 1999*; reviewed in *Ruvolo, 2003*; *Kraut, 2011*) and predicted JNK/AP-1 targets are enriched in the IZ (*Figure 1—figure supplement 3A*). AUCell activity for predicted AP-1 target genes is also highest in the IZ relative to the other clusters (*Figure 3—figure supplement 2B*). Moreover, predicted AP-1 targets positively correlate with de novo ceramide synthesis in IZ cells (r=0.94; *Figure 3—figure supplement 2C*). *Mmp1* is prominent amongst the JNK targets (*Uhlirova and Bohmann, 2006*; *Stevens and Page-McCaw, 2012*) in that it is enriched in the IZ and correlates positively (r = 0.9) with de novo ceramide synthesis (*Figure 3B*; *Figure 3—figure supplement 2D*). The gene encoding the rate-limiting enzyme in the glycosphingolipid pathway, *GlcT*, is also a target of the JNK pathway and its expression positively correlates (r = 0.89) with *Mmp1* (*Figure 3—figure supplement 2E*). This suggests an opportunity for feedback inhibition whereby ceramide activates the JNK pathway, including its downstream target GlcT, which limits free ceramide levels. This would prevent uncontrolled JNK activation that can result in cell death (*Kohyama-Koganeya et al., 2004*).

The possibility of a link between ceramide biosynthesis, JNK pathway, and MMP1 within the transitional IZ population is intriguing from a functional standpoint, and we therefore probed this further using molecular-genetic tools. Immunolocalization using an antibody against MMP1 reveals that the expression of the protein is limited to the region of the IZ (*Figure 3D–D''*). MMP1 is a secreted protein, and is detected in cells at the edge of the IZ, likely to act as a metalloprotease in reorganizing the ECM around the newly forming hemocytes. Consistent with the high representation of *Mmp1* transcript in the IZ, inhibition of the JNK pathway (*JNKK/hep*[RNAi]) in the IZ (*CHIZ-GAL4*) alone eliminates all the diffuse MMP1 protein detected in the IZ neighbors (*Figure 3E–E''*), suggesting the IPs are a source of MMP1. Likewise, a huge increase in MMP1 protein is seen when an activated form of JNKK (*hep*[act]) is expressed in the IPs (*Figure 3F–F''*). Interestingly, activation of JNK in this manner does not cause extensive cell death suggesting the possible concurrent presence of a cell death inhibition mechanism operating within the IZ cells (*Uhlirova et al., 2005*). A schematic diagram summarizing the transcriptomic control of metabolic genes in different cell populations is shown in *Figure 3G*.

## Synergistic combinations of genetic and transcriptomic data
### Case study 1. Pointed and plasmatocyte formation

The ETS family transcription factor Pointed (Pnt) functions downstream of RTK/Ras/MAPK pathways and regulates differentiation and proliferation in multiple fly tissues including blood (*Zettervall et al., 2004*; *Dragojlovic-Munther and Martinez-Agosto, 2013*; *Shwartz et al., 2013*; reviewed in *Vivekanand, 2018*). *pnt* transcript is expressed in very few cells in the PSC, rising slightly in MZ1/MZ2, which is particularly noticeable in MZ2-3 and then continuing in its rising trend in IZ/proPL and PL. This suggests the possibility of multiple functions for *pnt* in these different cell types. *Pnt* levels decline significantly in CCs, particularly mCC, suggesting low RTK-related activity in these cells (*Figure 4A*).

Knockdown of *pnt* specifically in the MZ (*dome^MESO^-GAL4, UAS-pnt^RNAi^*) blocks the differentiation of the progenitor population (*Figure 4B–C*). No *Hml^Δ^-DsRed* positive IZ, proPL, or PL cells are detected (*Figure 4D*). There is an increase in *dome^MESO^* positive progenitors, but the complete lack of IZ and CZ results in an overall smaller lymph gland (*Figure 4—figure supplement 1A, B*). Published literature shows that the marker *Tep4* is expressed in a limited number of MZ cells that are the least mature (*Benmimoun et al., 2015*; *Oyallon et al., 2016*; *Blanco-Obregon et al., 2020*). *Dome* expression initiates in the same cells as *Tep4* but extends further within the MZ. In fact, using a combination of cell-marking methods, we clearly detect a population of cells that are *dome*-positive, *Hml^Δ^-DsRed*-negative, and *Tep4*-negative (*Figure 4E*) and these are also distinct from the IZ cells since they do not express an IZ specific-*GAL4* driver (*CHIZ-GAL4*) (*Figure 4F*). In stark contrast to *dome^MESO^-GAL4, UAS-pnt^RNAi^*, the same *pnt^RNAi^* expressed in the high *Tep4* positive early MZ progenitors (*Tep4-GAL4, UAS-pnt^RNAi^*), has no observable effect on either differentiation or lymph gland size (*Figure 4G–I*; *Figure 4—figure supplement 1C, D*). Combined with the fact that *pnt* expression is higher in MZ2-3 than in earlier MZ subpopulations, we propose that Pnt functions in a post-*Tep4* and pre-IZ population of *dome*-expressing cells likely within the sub-state MZ2-3, and promotes their transition into the intermediate IZ and proPL cell types. Interestingly, CCs still form when *pnt* is depleted in the MZ (*Figure 4J–K*; *Figure 4—figure supplement 1E*). This suggests that CC formation does not require Pnt activity and that there is a direct route (perhaps made more prominent under these mutant conditions) for a MZ cell to become a CC without first going through a *Hml^Δ^-DsRed* positive IZ/proPL/PL cell type.

Since the loss of *pnt* in the MZ blocks entry into IZ, the higher *pnt* expression in the IZ suggests yet another different and additional role in this zone. This IZ function is explored in some detail elsewhere (*Spratford et al., 2020*), where we demonstrate that loss of *pnt* in the IZ prevents these cells from exiting their transitional state and prevents PL differentiation. Together with the data presented here, we conclude that Pnt is required for both entry into and exit from the IZ. Once again, CC formation is not affected upon loss of *pnt* in the IZ (*Figure 4L–M*; *Figure 4—figure supplement 1F*), reinforcing the idea of a direct path between MZ and CC without an intervening *Hml^Δ^*-positive cell. This is in addition to the IZ-dependent CC formation in wild-type (WT) described earlier.

Finally, since even higher *pnt* levels are seen in PL (*Figure 4A*), we eliminated Pnt function in *Hml^Δ^-GAL4* expressing cells (*Hml^Δ^-GAL4, UAS-pnt^RNAi^*). This causes a large number of *Hml^Δ^-GAL4+* cells to be converted into CCs (*Figure 4N–P*). Thus, loss of *pnt* in a *Hml^Δ^-GAL4+* precursor alters the PL/CC fate choice. Keeping in mind that Pnt is activated by RTK/MAPK pathways and that the Serrate/Notch pathway is important for CC formation, we conclude that Notch activation directs *Hml^Δ^* positive cells towards CC differentiation, while Pnt functions antagonistically to prevent this process, driving the cell instead to a PL fate (similar antagonistic interactions between these two pathways are seen in many tissues; reviewed in *Sundaram, 2005*). Overall, our combined RNA-Seq and genetic analysis demonstrate that Pnt has several distinct, context-dependent functions during lymph gland development and the genetic and sequencing data mutually validate each other.

Note that the expression of *pnt* changes gradually as one progresses through the lymph gland. There are no quantal jumps between zones, yet the function of Pnt is distinguishable between cell types (summarized in Figure 7C). We believe that most developmentally relevant genes will not be 'hallmark indicators' or markers of zones, but the trend and subtle modulations of their expression could have unique functional consequences for each cell type.

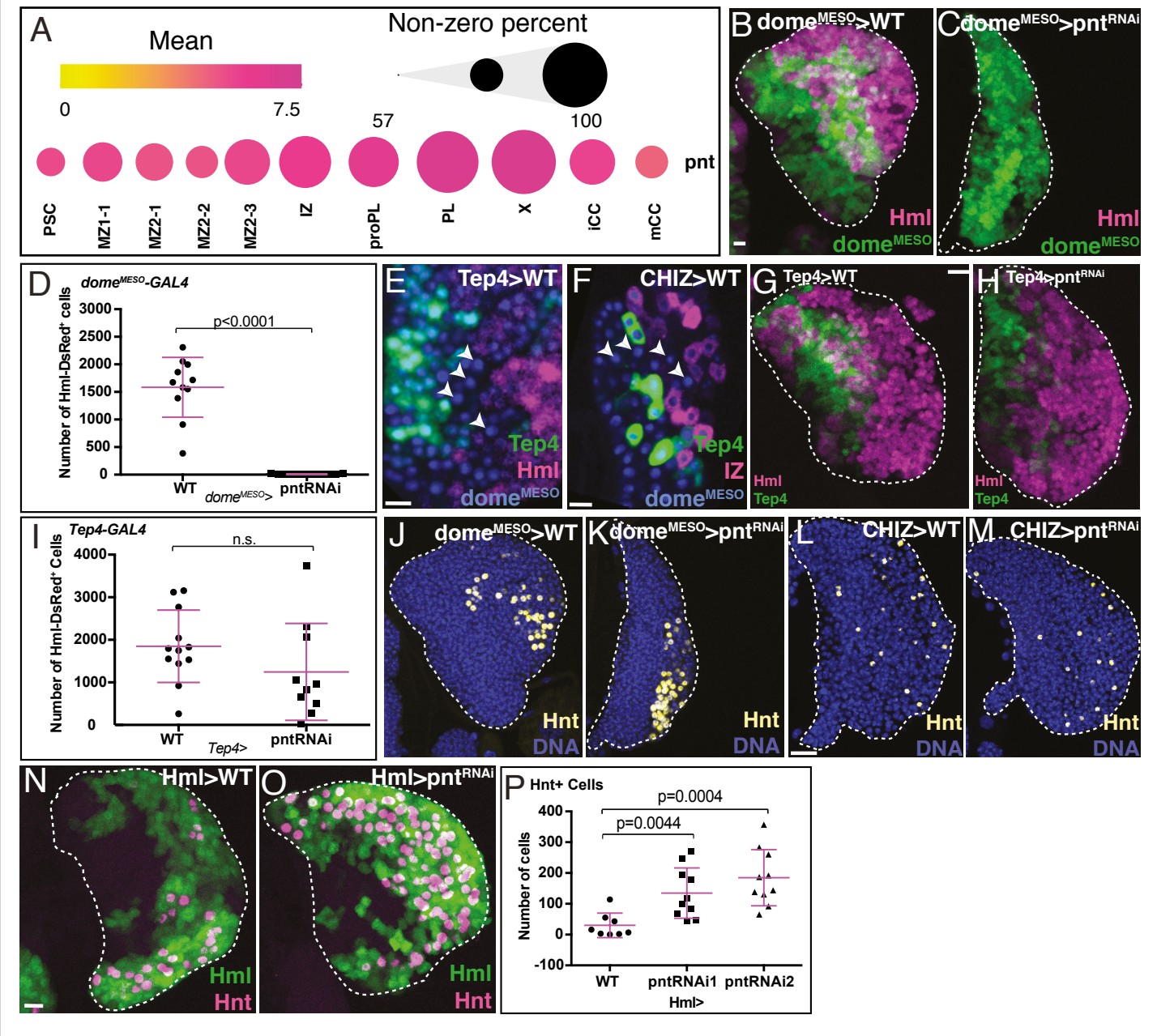

**Figure 4.** Role of Pnt in lymph gland development. (**B, C, G, H**, and **J–O**) are maximum intensity projections of the middle third, and (**E, F**) are single confocal slices. Lymph glands are from wandering (**B, C, G, H, J–M**) or early (**E, F, N, O**) third instar larvae. Scale bars: 10 µm. (**A**) scRNA-Seq analysis shows graded *pnt* expression in different subpopulations (see text for details). The mean level of *pnt* expression is represented by the dot color and the percentage of cells that expresses *pnt* is indicated by the dot size. (**B–D**) Genetic analysis of *dome^MESO^-GAL4, UAS-GFP, Hml^Δ^-DsRed* lymph glands. (**B**) Control lymph gland. *dome^MESO^* marks MZ (reported by GFP, green) and *Hml^Δ^* marks IZ and CZ (reported by DsRed, magenta). (**C**) Expression of *pnt^RNAi^* in the MZ cells that are *dome^MESO^* positive prevents the formation of IZ and CZ cells. (**D**) Quantitation of data shown in (**B, C**) reveals a complete loss of *Hml^Δ^-DsRed+* cells (IZ, proPL, or PL) in *dome^MESO^-GAL4, UAS-pnt^RNAi^*. n=11. (**E**) Genotype, *Tep4-GAL4, UAS-GFP, dome^MESO^-EBFP2, Hml^Δ^-DsRed*. Late progenitors marked by arrowheads are positive for *dome^MESO^* (blue) but negative for *Tep4* (green) and *Hml^Δ^* (red). (**F**) Genotype, *CHIZ-GAL4, UAS-mGFP, dome^MESO^-EBFP2, Tep4-QF2, QUAS-mCherry*. A population of pre-IZ late progenitors marked by arrowheads are positive for *dome^MESO^* (blue), but negative for *Tep4* (green) and *CHIZ* (red). (**G–I**) Genotype, *Tep4-GAL4, UAS-GFP, Hml^Δ^-DsRed*. (**G**) Control lymph gland. *Tep4* (green) is expressed in a subset of MZ progenitors. *Hml* (magenta) marks IZ/CZ. (**H**) *pnt^RNAi^* expressed in *Tep4+* MZ cells has no effect on the formation of IZ/CZ cells. (**I**) Quantitation of the data in (**G, H**) shows no significant difference in the number of *Hml^Δ^-DsRed+* cells when *pnt^RNAi^* is expressed using *Tep4-GAL4* (contrast with **D**). WT: n=12; *pnt^RNAi^*: n=10. (**J, K**) Genotype is the same as in (**B, C**). (**J**) Control shows nuclei (DNA, blue) and crystal cells (CCs; Hnt, yellow). (**K**) Depletion of *pnt* in *dome^MESO^* positive MZ cells does not prevent formation of Hnt+ CCs. Quantitation in *Figure 4—figure supplement*

*Figure 4 continued on next page*

Figure 4 continued

*1E*. (**L, M**) Genotype, *CHIZ-GAL4, UAS-mGFP* (GFP not shown). (**L**) Control shows DNA (blue) and Hnt (yellow). (**M**) Depletion of *pnt* in IZ cells does not prevent formation of Hnt+ CCs. Quantitation in *Figure 4—figure supplement 1F*. (**N–P**) Genotype, *Hml^Δ-GAL4, UAS-2xEGFP. Hml^Δ* (green) and Hnt (magenta). (**N**) Control. (**O**) *pnt^RNAi* expressed in *Hml^Δ-GAL4+* cells increases the number of Hnt+ CCs. (**P**) Quantitation of the data in (**N, O**) shows a significant increase in the number of Hnt+ CCs with *pnt^RNAi* using two independent RNAi lines driven by *Hml^Δ-GAL4*. WT: n=8; *pnt^RNAi* 1 & 2: n=10. CZ, cortical zone; IZ, intermediate zone; MZ, medullary zone; PL, plasmatocyte.

The online version of this article includes the following figure supplement(s) for figure 4:

**Source data 1.** Source data for *Figure 4D1 and P* and *Figure 4—figure supplement 1A-F*.

**Figure supplement 1.** Quantification of *pnt^RNAi* phenotypes.

## Case study 2. Numb/Musashi assisted non-canonical Notch signaling in crystal cells

A canonical, Serrate-dependent Notch signal is required for CC formation from a *Hml+* precursor; whereas a separate, non-canonical, ligand-independent and Sima (Hif)-dependent Notch signal is important for CC maintenance (*Mukherjee et al., 2011*). Mechanistic details of this complex process, which remained elusive for over a decade are described below, and could only be deciphered when genetic data are analyzed in the context of the expression profiles of a number of genes.

Notch target genes have been investigated at length using multiple functional and biochemical criteria (*Krejcí et al., 2009*; *Terriente-Felix et al., 2013*). Based on the RNA-Seq data, such a list of targets can be classified into two groups, which for simplicity, we call type I and type II. Type I targets (such as *E(spl)m3-HLH*, *cv-c*, and *CG3847*) are expressed at a higher level in iCCs than in mCCs (*Figure 5A*), while type II targets (such as *CG32369*, *bnl*, and *IP3K2*), are more highly enriched in mCCs than in iCCs (*Figure 5B*). The type I targets correlate positively with the maturity marker *PPO2* in iCC but negatively in mCCs (*Figure 5C*). In contrast, type II targets positively correlate with *PPO2* in both iCC and mCC populations (*Figure 5D*). As one example of validation, *branchless* (*bnl*), a type II target by its expression, is seen in only a subset of the CCs, expected to be the more mature (*Figure 5— figure supplement 1A-A'*; *Tattikota et al., 2020*). Type II Notch targets, including *bnl* and *CG32369*, have been shown in independent studies to be both Notch pathway and Sima/hypoxia-responsive (*Li et al., 2013*; *Terriente-Felix et al., 2013*; *Kamps-Hughes et al., 2015*; *Du et al., 2017*), and we find that enhancer sequences for these two genes contain combinations of both Su(H) and Sima binding sites (*Figure 5—figure supplement 1B, C*).

Our past work established a role for Sima in CC maintenance (*Mukherjee et al., 2011*). However, these results predate the sub-classification of CCs into iCC and mCC subpopulations, which could, potentially, provide a cellular context for the switch between the two different modes of Notch signaling. Based on the expression patterns of the type II Notch targets, Sima-dependent signaling is expected to be highest in mCC, however, we detect no significant differences in total *sima* transcript levels between PL and mCC populations (*Figure 5E*). This was not entirely surprising because Sima is known to be primarily controlled at the protein level (*Bertolin et al., 2016*). However, the bulk RNA-Seq experiments have sufficient depth of sequencing to identify different isoforms, and this was key to the understanding of the pathway.

Of the four proposed RNA isoforms, *sima-RA, -RB, -RC, and -RD* (*Figure 5E–F*), two, *-RB* and *-RD*, are not expressed in lymph glands. The full length and most widely studied isoform is *sima-RA*, which is significantly higher in its expression in PLs than in CCs (*Figure 5E–F*). Importantly for this study, the smaller isoform *sima-RC* is expressed highest in mCCs (*Figure 5E–F*; *Figure 5—figure supplement 1D*). The iCCs contain low levels of both isoforms, which explains why the total *sima* transcript expression is lower in iCCs than in PLs and mCCs (*Figure 5E*). qRT-PCR using isoform-specific primer sets confirms *sima-RC*, but not *sima-RA*, is expressed at a higher level in mCCs compared to iCCs and PLs (*Figure 5—figure supplement 1E, F*). Interestingly, a past study has suggested that a stabilized Sima protein is involved in the auto-regulation of *sima-RC* (*Kamps-Hughes et al., 2015*) that is consistent with the presence of Sima binding sites within the identified *sima-RC* enhancer element (*Figure 5— figure supplement 1G*).

Importantly, the predicted protein, Sima-PC, encoded by the smaller than full-length *sima-RC* transcript, would be truncated and lack the N-terminal motif that is essential for binding its partner Tango (Hif-1β/ARNT). The Sima/Tango heterodimer is essential for eliciting a hypoxia response (*Gorr*

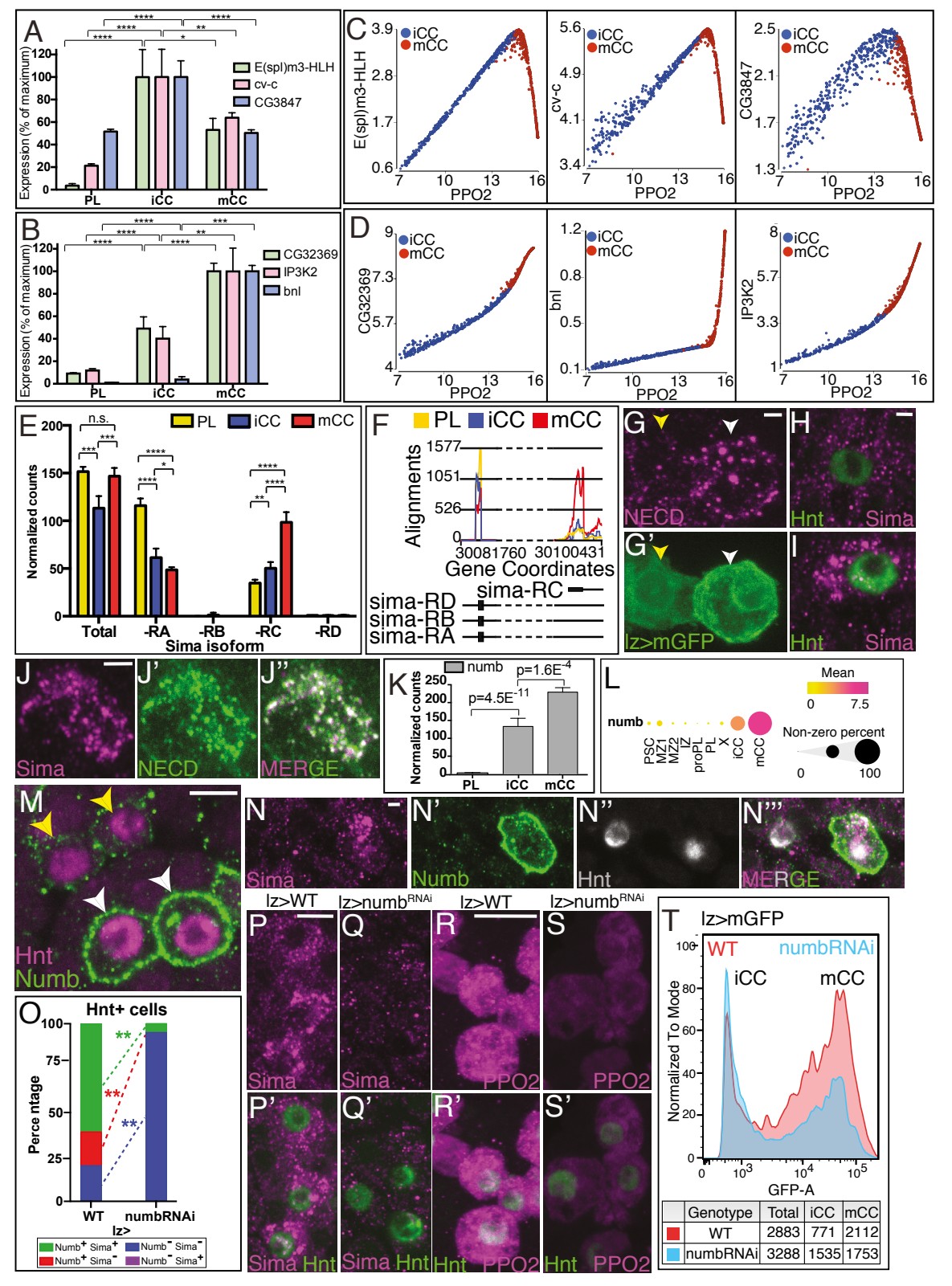

**Figure 5.** Numb promotes non-canonical Notch/Sima signaling. (**A, B, E, F, K**) are from bulk RNA-Seq whereas (**C, D, L**) are from single-cell RNA-Seq. (**A**) 'Type I' Notch targets with highest expression in iCC, lower in mCC, and lower still in PL. (**B**) 'Type II' Notch targets have their lowest expression in PLs, increase in iCCs, and are expressed highest in mCCs. (**C**) Type I Notch targets correlate positively with the CC maturity marker *PPO2* in iCC and negatively in mCC. (**D**) Type II Notch targets correlate positively with *PPO2* in both iCCs and with an even higher slope in mCCs. (**E**) Total *sima* transcript

*Figure 5 continued on next page*

*Figure 5 continued*

levels are similar in PL and mCC. The usually major splice variant, *sima-RA* decreases with CC maturity. The normally minor *sima-RC* isoform increases from PL to iCC and is higher still in mCC. (**F**) Alignment counts for an exon specific to *sima-RC* are highest in mCC (red). The FlyBase coordinates are in *Figure 5—figure supplement 1D*. (**G–G'**) Live internalization assay in *lz-GAL4, UAS-mGFP* lymph glands with an antibody against the extracellular domain of Notch (N$^{ECD}$, magenta) to visualize uptake and stabilization of full-length Notch protein. Large Notch punctae are specifically located in mCC (GFP$^{HI}$; white arrowhead) but not in iCC (GFP$^{LO}$; yellow arrowhead). (**H, I**) Protein staining for Hnt (green) and Sima (magenta) shows numerous large Sima punctae in mCC (**I**, high Hnt) but not in iCC (**H**, low Hnt). (**J–J''**) Full-length endocytosed Notch protein is visualized in a live internalization assay with an antibody against N$^{ECD}$ (green, **J'**) and then fixed and stained for Sima protein (magenta, **J**). Numerous large N$^{ECD}$ and Sima punctae colocalize and therefore appear white in the merged image (**J''**). (**K**) *numb* transcript level is minimal in PL, increases in iCC and further increases in mCC. (**L**) Dot plot showing the mean level of *numb* expression (indicated by dot color) and the percentage of cells that express *numb* in each population (indicated by dot size). Compared to all cells identified by scRNA-Seq, *numb* transcript levels are enriched in iCC and are even higher in mCC. (**M**) Strong Numb protein staining (green) is restricted to mCCs (white arrowheads), with stronger Hnt staining (magenta) and not in low Hnt-expressing iCCs (yellow arrowheads). (**N–N'''**) Large Sima punctae (magenta) are only seen in Hnt (gray) positive crystal cells (CCs) with high Numb staining (green). (**O**) Quantitation of the data in (**P–Q'**) showing the percentage of Hnt+ CCs that are positive or negative for Sima and Numb in wild-type (WT) and upon knockdown of *numb*. No Numb negative Sima+ cells are evident in either genotype. Depletion of *numb* causes loss of nearly all Sima+CCs. n=221 total CCs from WT and n=111 CCs for *numb*$^{RNAi}$. (**P–P'**) WT lymph glands display large Sima punctae (magenta) in Hnt+ (green) CCs. (**Q–Q'**) The large Sima punctae are eliminated when *numb* is depleted in CCs using *lz-GAL4 UAS-numb*$^{RNAi}$. Quantitation in *Figure 5—figure supplement 3F*. (**R–R'**) PPO2 protein (magenta) is high in most Hnt+ (green) CCs in WT. (**S–S'**) PPO2 levels (magenta) are lower in Hnt+ (green) CCs when *numb* is depleted using *lz-GAL4 UAS-numb*$^{RNAi}$. Quantitation in *Figure 5—figure supplement 3I*. (**T**) Flow cytometry shows that when *numb* is knocked down (Genotype: *lz-GAL4 UAS-numb*$^{RNAi}$), a large proportion of the mCCs (GFP$^{HI}$) are lost while the total number of CCs does not change significantly (2883 in WT vs. 3288 in *numb*$^{RNAi}$). Scale bars: 2 µm in (**G, H, J, N, P–Q'**); 5 µm in (**M, R–S'**). See *Figure 5—figure supplement 3* for lower magnification views of lymph glands shown in (**P–S'**). CC, crystal cell; iCC, immature crystal cell; mCC, mature crystal cell; PL, plasmatocyte.

The online version of this article includes the following figure supplement(s) for figure 5:

**Source data 1.** Source data for *Figure 5A–B, E, K and O*, *Figure 5—figure supplement 1E, F, M*, *Figure 5—figure supplement 2D, G* and *Figure 5—figure supplement 3H, I*.

**Figure supplement 1.** Notch/Sima related gene expression in crystal cells (CCs).

**Figure supplement 2.** Numb levels are controlled by Notch signaling.

**Figure supplement 3.** Numb promotes Notch/Sima signaling and crystal cell (CC) maturity.

**Figure supplement 4.** Flow cytometric analysis of CC subclasses from *lz-GAL4* (control) and *lz-GAL4, UAS-numb*$^{RNAi}$ lymph glands.

---

*et al., 2004*; *Romero et al., 2008*). Sima-PC retains the oxygen (and prolyl-hydroxylation)-dependent degradation (ODD) domain (*Figure 5—figure supplement 1H, I*; *Gorr et al., 2004*; *Romero et al., 2008*). We conclude that the Sima-dependent non-canonical Notch signaling (*Mukherjee et al., 2011*) involves the formation of the Tango-independent Sima-PC/Notch heterodimer that does not activate hypoxia-inducible genes. Rather, such a complex will only form in mCCs where the abundance of the Sima-RC transcript is high, and help maintain the mature CCs. Indeed, we find that mCCs, but not iCCs, contain large punctae of endocytosed and stabilized full-length Notch (*Figure 5G–G'*) and large punctae of Sima protein (*Figure 5H–I*) that colocalize in the same punctae (*Figure 5J–J''*). Taken together, the data show that mCCs, but not iCCs, participate in Notch/Sima signaling that is facilitated by the switch to an alternate isoform of Sima in the mature CCs.

## Numb and Musashi in CC determination

The existence of punctae containing stabilized N/Sima proteins suggests that endocytic mechanisms are important for this non-canonical Notch pathway. This motivated us to focus on the gene *numb*, which encodes a component of the endocytic pathway that promotes internalization/trafficking of Notch (*Couturier et al., 2013*; *Yap and Winckler, 2015*; *Johnson et al., 2016*; *Shao et al., 2016*) and intracellular Numb blocks canonical Notch signaling (*Frise et al., 1996*; *Spana and Doe, 1996*). Given that CC induction requires a ligand-dependent canonical Notch signal (*Lebestky et al., 2003*), we found it surprising that *numb* RNA is enriched in CCs (*Cho et al., 2020*; *Figure 5L*). *numb* levels are by far the highest in both iCCs and mCCs compared to all other lymph gland cell types (*Figure 5L*), with a significant increase during the transition from iCC to mCC (*Figure 5K–L*), and correlating strongly (r=0.99) with *PPO2* (*Figure 5—figure supplement 1J*). *numb* is reported to be a Notch target (*Rebeiz et al., 2011*), a conclusion we find is well-supported for CCs as well. Constitutively active Notch (Notch$^{ACT}$) expressed in CCs, raises *numb* levels while knockdown of Notch observed in *Notch*$^{RNAi}$ has the opposite effect (*Figure 5—figure supplement 2A-G*).

By far, the most spectacular control of Numb is post-transcriptional. In spite of the significant quantities of *numb* RNA in iCCs, Numb protein is exclusively detected in mCCs and not in iCCs (*Figure 5M*; also later in *Figure 6A–C'*). In fact, immunohistochemically detected Numb protein is a new and distinctive marker for mCC. The large Sima punctae described above are also exclusively seen in the Numb-expressing mCCs (*Figure 5N–N''''*; and WT quantitation in *Figure 5O*). These full-length Notch/Sima punctae co-localize with Numb punctae in a live endocytosis assay and are also seen in Hrs8-2 positive early endosomes (*Figure 5—figure supplement 3A-C''''*).

When trapped in endosomes, full-length Notch can promote a signal even in the absence of a ligand (*Vaccari et al., 2008*; reviewed in *Fortini and Bilder, 2009*). We propose that the exclusive and specific expression of the Numb protein in mCCs assists such a trapping of the full-length Notch/Sima complex in early endosomes. This enhances rather than inhibits signaling by this unusual and non-canonical pathway. As a direct genetic test of this model, we downregulate *numb* in CCs (*lz-GAL4; UAS-numb^RNAi*; *Figure 5—figure supplement 1K-M*) and find a clear reduction in the large Sima punctae, as well as reduction in PPO2 expression, indicating a decrease in the number of mCCs without affecting the total number of CCs (*Figure 5*; *Figure 5—figure supplement 3D-I*). In summary, loss of Numb has no effect on initial CC specification and iCC induction; instead depletion of Numb inhibits the maturation of iCCs to the mCC state. These results are further confirmed by flow cytometric analysis, which shows that when *numb^RNAi* is expressed in all CCs, the proportion of mCCs decreases with a corresponding increase in iCCs (*Figure 5T*). Flow cytometry also shows that *numb^RNAi* exhibits smaller cell size (FSC) and less cellular complexity (SSC) in mCCs when compared to WT, but does not cause such reductions in iCCs (*Figure 5—figure supplement 4A-D*). As shown earlier, the most mature CCs contain > 4N DNA attributed to endocycling (*Figure 1H*). We find that *numb* knockdown in CCs causes a very clear reduction in the number of cells with > 4N DNA content (*Figure 5—figure supplement 4E*). Thus loss of *numb* reduces mCCs (*Figure 5T*), and the ones that remain show loss of endoreplication and other signs of terminal CC maturity.

To summarize, *numb* mRNA is expressed in all CCs, but its translation to Numb protein is blocked in iCCs and is therefore unable to inhibit the Ser/N specification signal. In this model, it is essential that this translational block be specifically eliminated in mCC, where Numb promotes the Sima/Notch non-canonical signal (summarized in *Figure 7D*). Investigations in mammalian systems have suggested that the RNA-binding protein Musashi (Msi) binds to *numb* mRNA and represses its translation (*Imai et al., 2001*). RNA-Seq results indicate that *msi* mRNA is expressed widely in the lymph gland, with the clear and notable exception of CCs, particularly in mCCs where *msi* is not enriched (*Figure 6D–E*). *msi* RNA shows a very strong negative correlation with *PPO2* (*Figure 6F*). It is therefore reasonable to consider *msi* as a candidate for the *numb* translational repressor in iCCs but not in mCCs. Consistent with this proposal, the Msi protein is also expressed broadly except in the Numb positive mCCs (*Figure 6G–G''*; *Figure 6—figure supplement 1A-B'*). A knockdown of *msi* (*Figure 6—figure supplement 1G*) in all CCs (*lz-GAL4, UAS-msi^RNAi*) causes a strong increase in Numb protein level in individual CCs relative to WT (*Figure 6H–J*; *Figure 6—figure supplement 1C-D'*). Similarly, when *msi^RNAi* is driven in *Hml^Δ* expressing cells (*Hml^Δ-GAL4, UAS-msi^RNAi*) that include the iCCs but not mCCs (*Figure 6C–C'*), an even more dramatic increase in the Numb protein is readily evident (*Figure 6K–M*; *Figure 6—figure supplement 1E-F'*). Combining the established nature of Msi function in mammals as an RNA-binding post-transcriptional repressor of *numb* (*Nakamura et al., 1994*), with the transcriptomic and genetic data presented here, we conclude that Msi controls *numb* RNA post-transcriptionally in iCCs and the absence of Msi in the mCC allows Numb protein to be made in these cells (summarized in *Figure 7D*).

## Discussion

### Heterogeneity of cell types

The cells of the small, hematopoietic lymph gland tissue are far more complex at the genome-wide expression level than could have been anticipated by earlier marker and genetic analyses. This is now confirmed by this work, and by the earlier results of *Cho et al., 2020*. The first step in our analysis was to separate cells by FACS based on the canonical markers that classically define each zone within the lymph gland (reviewed in *Banerjee et al., 2019*). When probed for the presence of known 'hallmark genes,' the separated cells expressing them match up with their corresponding zones, providing early validation of the methods used. This process also allows us to identify zone enriched gene expression

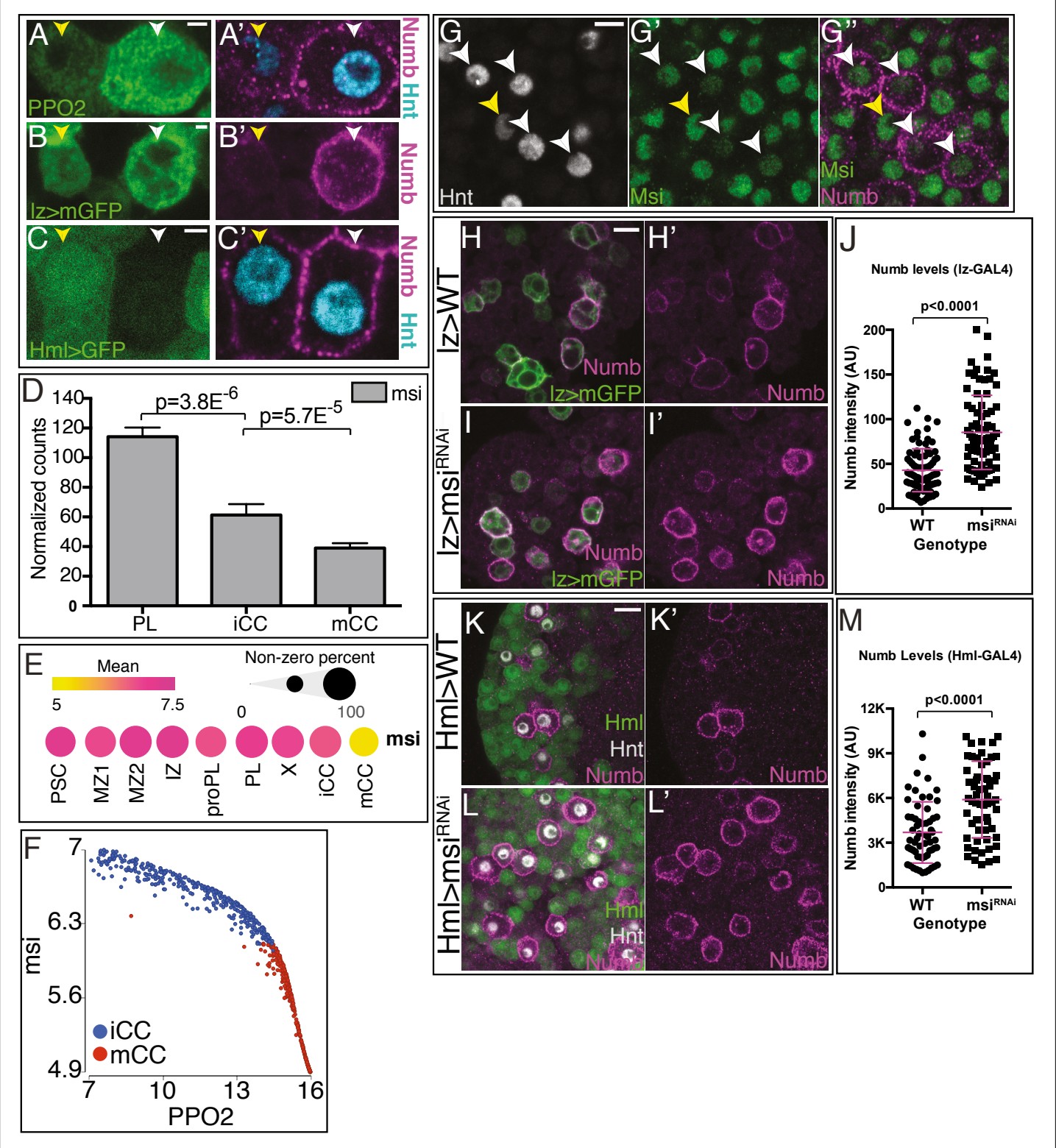

**Figure 6.** Musashi (Msi) regulates Numb protein levels. (**A–C'**) Three independent criteria distinguish iCCs from mCCs and validate Numb expression. A pair of cells is shown in each panel, and for each, the one on the left (yellow arrowhead) is iCC and the one on the right (white arrowhead) is mCC by the described criteria. (**A–A'**) iCC is low in Hnt (cyan) and PPO2 (green), whereas mCC is high for both. Numb (magenta) is only seen in mCCs. (**B–B'**) iCC is low and mCC high in *lz> mGFP* expression. Numb (magenta), is only seen in mCCs. (**C–C'**) iCC expresses and mCC is negative for *Hml^A>GFP* (green) expression. Hnt (cyan) is lower in iCC than in mCC. Numb (magenta) is only detected in mCCs. (**D**) Bulk RNA-Seq shows *msi* transcript levels

*Figure 6 continued on next page*

*Figure 6 continued*

decrease with crystal cell (CC) maturity from PL to iCC to mCC. (**E**) Mean level of *msi* expression (indicated by dot color) and the percentage of cells that express *msi* in each population (indicated by dot size) are represented in a dot plot. scRNA-Seq shows reasonably uniform *msi* transcript level in all lymph gland cells with the exception of mCCs in which *msi* transcript is much lower. (**F**) *msi* expression negatively correlates with *PPO2* levels in iCC and mCC populations. (**G–G″**) CCs (Hnt+; gray) with low levels of the fusion protein Msi-GFP (green; white arrowheads) have high Numb staining (magenta), indicating that they are mCCs. iCC with low Hnt staining (yellow arrowhead) shows higher levels of Msi-GFP and no Numb protein. (**H–J**) Genotype, *lz-GAL4, UAS-mGFP*. Numb (magenta) is expressed at moderate levels in wild-type (WT) *lz>mGFP+* CCs (**H–H′**) and increases significantly with *lz-GAL4, UAS-msi^{RNAi}* (**I–I′**). Quantitation of Numb levels in individual crystals, n=80 (**J**). (**K–M**) CCs are visualized with Hnt (gray) and *Hml^{Δ}-GAL4, UAS-2xEGFP* is used as a driver. In WT (**K–K′**) *Hml^{Δ}>GFP* positive cells (green) do not show high levels of Numb protein (magenta). Whereas upon expression of *UAS-msi^{RNAi}* (**L–L′**), Numb protein greatly increases. Data quantitating Numb levels in individual CCs, n=60 (**M**). All images are single confocal slices. See *Figure 6—figure supplement 1* for lower magnification views of lymph glands shown in (**G–G″, H–I′, K–L′**). Scale bars: (**A–C**), 2 µm; (**G**), 5 µm; (**H I, K, L**), 10 µm. iCC, immature crystal cell; mCC, mature crystal cell; PL, plasmatocyte.

The online version of this article includes the following figure supplement(s) for figure 6:

**Source data 1.** Source data for *Figure 6D, J and M* and *Figure 6—figure supplement 1G*.

**Figure supplement 1.** Numb protein level is controlled by Musashi.

for less well-characterized cell types, including the IZ cells (IPs), as well as immature and mature CC types (iCC and mCC). This bulk RNA-Seq approach was further extended using scRNA-Seq and genetics to identify possible combinations of markers that identify each cell type (*Figure 7A*). However, the primary goal of this work is not to identify more tissue-specific hallmark genes (although several were found), but to utilize RNA-Seq as a tool with other genetic strategies to understand: cell-fate specification, the multiple developmental paths available to a cell, and the mechanistic links between expression trends and developmental function. Many individual examples, and two complete case studies are presented that solve long-standing questions in *Drosophila* hematopoiesis (*Figure 7C–D*).

The transcriptomic data are most useful in determining trends in the collective behavior of a set of related genes. At the core of this assertion is the fact that most developmentally relevant genes function a context-dependent manner, and their individual expression is therefore not exclusively limited to a single cell type, but certain combinations of expressed genes could approximate their identities (*Figure 7A*). Obvious exceptions are genes marking functions of terminal states such as *lz* or *NimC1*, but even in such cases, RNA expression begins in multipotent precursors and continues in the terminal cell types. The case studies presented in this work demonstrate this concept, showing that a graded expression pattern of a transcription factor allows the identification of specific phenotypes for each developmental step. Similarly, expression of an alternate isoform for the protein Sima and the RNA-binding protein Msi explains why Numb inhibits canonical Ser/Notch function but not non-canonical Sima/Notch function in the same cell type. Thus the motivation for this study is to provide multiple examples that take advantage of the ready access to genetic tools that make *Drosophila* a particularly attractive system in which to establish detailed mechanistic aspects of complex pathways. Based on the long history of conservation of basic principles, it is not unreasonable to expect that parallels to such mechanisms will be found in mammalian hematopoiesis.

Employing fairly conservative criteria for cluster separation in scRNA-Seq, we identify eight primary clusters. The CCs were subclustered to yield iCC and mCC giving rise to the following nine groups of cells: a single cluster each for PSC, X (a mitosis and replication stress-related cluster), PL, and CC (subclustered into iCC and mCC). Two clusters each were identified for MZ (MZ1 and MZ2), and one for the two transitional populations (IZ and proPL). The compact arrangement of the majority of clusters implies smooth developmental transitions between them even as, from a gene-enrichment point of view, they represent different cell types. However, from a developmental biology point of view, it is the functional differences between clusters that must be used to define them as distinct cell types. It is virtually impossible to find any transcript that is 100% cell-specific, and therefore our analysis focuses on trends and enrichments in transcriptional patterns. Sometimes, as in the case of *pnt*, the changes in expression along each developmental step can be very small, but the trend defines its multiple functions and only functional data from mutant analysis provides validation for the gene expression patterns.

RNA-Seq is by now a commonly used technique in many fields, although its first use in lymph gland hematopoiesis was relatively recent (*Cho et al., 2020*). That study identified new markers and validated the expression of a representative number of the expressed genes. In *Supplementary file 2*,

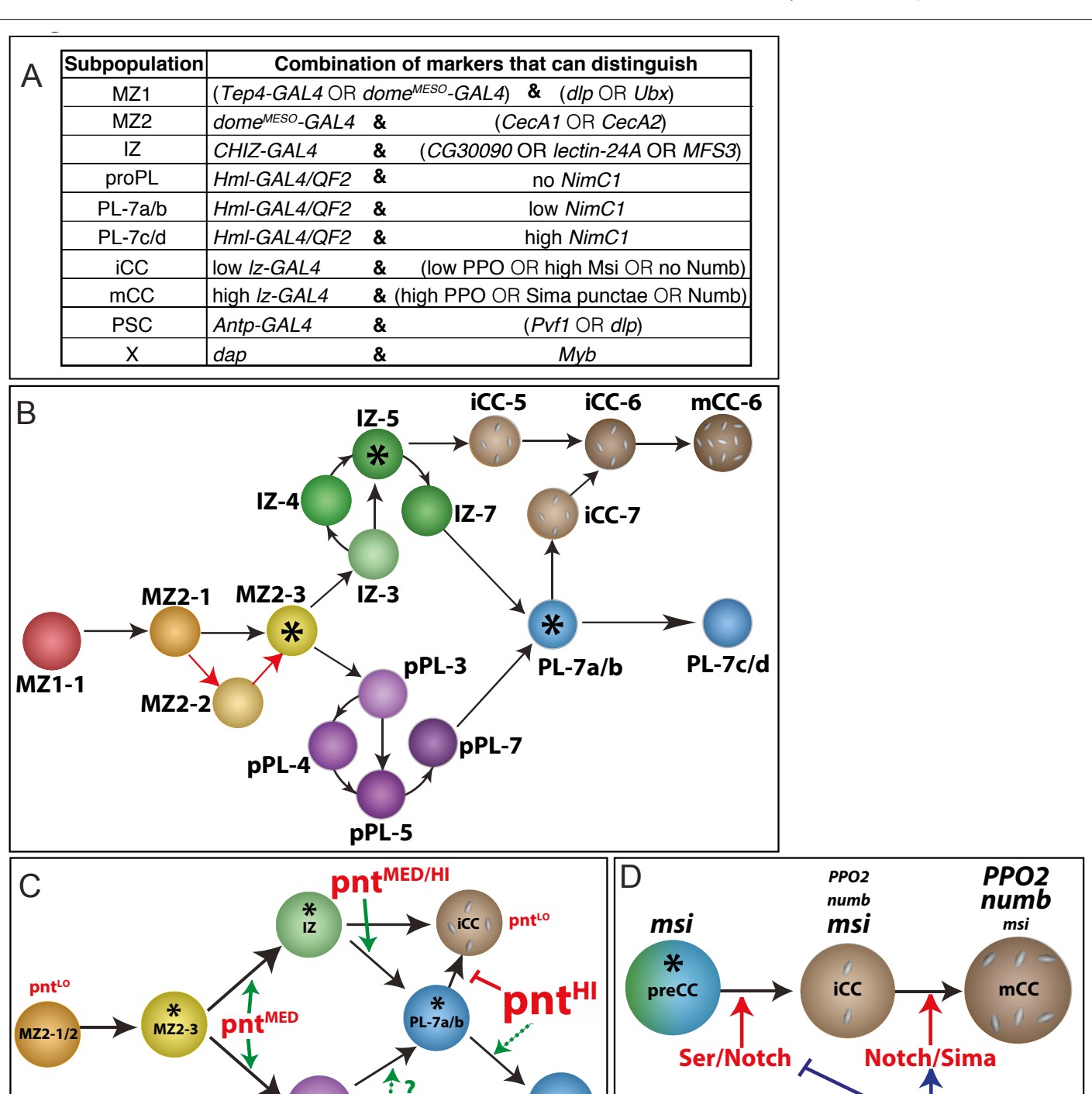

**Figure 7.** Summary of markers, case studies, and a model for the developmental progression of lymph gland cells. (**A**) Table showing representative combinations of markers that can be used to distinguish between the subpopulations of cells identified in this study. Markers specific to any one cell type or cluster are rare. The entire transcriptome contributes to cellular identities. The combinations listed here are useful identifiers of the subpopulations. Alternative marker combinations are separated by 'OR.' (**B–D**) LG developmental models. Nodes at which cells make alternate fate choices (FCs) are marked by asterisks. (**B**) A model demonstrating the complexity and developmental progression of cell types within the *Drosophila* lymph gland (see Discussion for details). PSC cells (not shown) have a developmental origin distinct from the rest of the primary lobe cells. The medullary zone (MZ) cells are found in clusters MZ1 (earliest in pseudotime state 1) and MZ2. The most mature subpopulation within the progenitors is MZ2-3, which is an FC point that leads to two alternate transitional zones IZ and proPL (pPL), each replete with its own set of temporally distinct states.

*Figure 7 continued on next page*

*Figure 7 continued*

IZ-5 is the second FC node that allows either a plasmatocyte (PL) or a CC fate choice. The third FC node is at PL-7a/b, a state where a choice between a CC fate is weighed against maturation to a terminally differentiated PL. Additional minor paths and variations thereof are discussed in the text. The model emphasizes flexibility as its central feature that is characterized by transitional and alternate paths connecting progenitors to differentiated cells. These routes are specified by gradual trends and combinations of gene expression rather than by quantal transitions between cell types maturing towards common or distinct fates. (**C**) Context-specific function of Pnt is determined by its graded expression pattern that increases progressively from low ($pnt^{LO}$) in the early MZ2-1/2, to medium ($pnt^{MED}$) in the later MZ2-3 progenitors. Pnt plays a role, indicated by green arrows, in differentiation of MZ2-3 progenitors into either the IZ or proPL (pPL) cell types. Transcript levels for *pnt* are higher in IZ/proPL (designated MED/HI) than in the MZ. Its function is required for both the entry into and exit from the IZ state. High levels of Pnt ($pnt^{HI}$) promote PL fate and inhibit CC formation. (**D**) A combination of results from bulk and scRNA-Seq, flow cytometry, and in vivo genetics is used here to depict the mechanism by which canonical and non-canonical Notch signaling define CC fate. Precursors of CCs (preCC) receive a Serrate-dependent canonical Notch signal to be specified as iCCs. The RNA-binding protein Musashi (Msi), expressed in all preCCs and iCCs, inhibits Numb translation, allowing canonical Notch signaling. Msi is not expressed in mCCs (we speculate it is repressed by high Lz), and therefore Numb is expressed in these cells. This allows a non-canonical ligand-independent Sima-PC/Notch signal while inhibiting any residual Ser-dependent canonical Notch signaling in the mature CCs. CC, crystal cell; iCC, immature crystal cell; IZ, intermediate zone; mCC, mature crystal cell.

we present a detailed comparison of the transcriptional map comparing the clusters and subclusters of Cho et al., with those generated in our single-cell RNA-Seq. By comparing the sizes of the clusters/subclusters, the overlapping gene lists, and the expression patterns and genetic profiles (***Supplementary file 2***), we find that MZ1 is similar to the PH1 and PH2 subclusters in Cho et al.; MZ2 is similar to PH3 and PH4; IZ to PH5 and PH6; proPL to PM1; PL to PM2, PM 3, and PM4; PSC to PSC; iCC to CC1; mCC to CC2; and X is most similar to the 'GST-rich' cluster of Cho et al. The differences in where boundaries are drawn could arise from many sources, such as the experimental technique (drop Seq by Cho et al. vs. 10×), genetic background (Oregon R vs. w1118), and perhaps most importantly, the computational strategy (manual curation and aggregation of the clusters based on known gene expression by Cho et al. vs. unsupervised graph-based clustering in this study). Both studies provide useful data. The strength of our study is that we use FACS to sort populations defined as MZ, CZ, IZ, CC, and so on, and therefore, we are certain that the two clusters MZ1 and MZ2, for example, belong to the traditionally defined 'MZ' (***Jung et al., 2005***) and the same is true for the others. The second strength is that our strategy requires the use of multiple backgrounds and biological replicates, and the results are very consistent. Finally, given that most expression patterns represent trends rather than specific cells, and often different from the proteins they encode (such as for *numb*), the strongest validation of expression data, we feel is when it is in agreement with genetic strategies based on loss of function in a subset of cells (such as with *pnt or Mmp1*).

## An updated model for lymph gland hematopoiesis

In ***Figure 7B***, we summarize our results to present a model of lymph gland development. Our analysis is based on a single time point in development but the occupancy states in pseudotime allow us to use maturation states as a form of developmental clock. The model is largely based on adjacencies, genetic compositions, and validation by mutant analysis. Transition from pre-progenitors to progenitors, then through transitional IZ or proPL populations, finally on to PLs or CCs is a continuous process traversing gradually through a permissive landscape. It does not appear to be a set of pre-programmed, quantal decisions that a cell makes based on the expression of a single fate-specifying gene. This idea is gaining increased traction in the newer reports on mammalian hematopoiesis (***Velten et al., 2017***; ***Rodriguez-Fraticelli et al., 2018***; ***Weinreb et al., 2020***).

The developmental trajectory for *Drosophila* hematopoiesis is branched, and the subdivision of 9 expression-based clusters into 22 subpopulations is based on both cell type and the trajectory state in which they reside. It is important to point out that in this context, the cluster name (e.g., MZ1 or MZ2) represents cell types distinguishable by their gene-enrichment profile, whereas the 'states' (such as MZ2-1, MZ2-2, and MZ2-3) represent the same cell type (MZ2), but appearing at different pseudo-times (1, 2, or 3). Although the analysis is a snapshot of a particular real-time point in development, many developmental steps of a single cell type are represented as progress in pseudotime. For example, the MZ2-3 state is composed of the most mature cells of the MZ2 cell type. The next transitions to either of the two separate transitional cell types, IZ or proPL, that define alternate developmental paths. The cell states MZ2-3, IZ-5, and PL-7a/b (marked with an asterisk in ***Figure 7B–D***) are nodes of bifurcation based on this model. Some details of the model require further functional

confirmation in vivo that is beyond the scope of the current manuscript. It is anticipated that such details of cell identity will change with future refinements. However, the model in *Figure 7B* provides a blueprint and a rich opportunity to study changes in signaling, cell cycle, or possible modes of cell divisions that promote alternate cell fates.

## Transition zones provide alternative paths for hematopoietic development

An important finding of this study is the demonstration of alternate paths that initiate with the same progenitor types and terminate in the same differentiated fate, but they traverse through distinct transitory cell types. The distinction between transitional states such as IZ and proPL would be less remarkable, if they did not also have additional unique characteristics and functions. For example, together the genetic and RNA-Seq data suggest that proPL is likely a major source of the equilibrium signal, whereas IZ largely contributes to the JNK signal. The two cell types are largely non-overlapping and virtually non-adjacent in a 3D t-SNE representation of the clusters. These alternate routes are reminiscent of the concept of progression through alternate epigenetic landscapes proposed by Waddington (*Waddington and Kacser, 1957*) at the very dawn of Developmental Biology. Finally, in T cell development, there is evidence to suggest that intermediate cells bridge the major singly and doubly marked populations, but even less is known about their possible developmental roles (*Kaech and Cui, 2012*).

Minor paths not involving either of the two major transitional states (IZ or proPL) are consistent with, but not fully established yet by our data. For instance, the earliest PL clusters (PL-3) are sandwiched between MZ2 and PL-7 with no intervening proPL or IZ cells, suggesting a direct MZ to PL path, or perhaps one that involves X as an intermediary. As another example of a minor path, a small number of iCC cells follow the path PL-7/iCC-7/iCC-6/mCC-6. The iCC-7 to iCC-6 transition is a reversal in pseudotime. Although unexpected, this supports the concepts of transdifferentiation (*Leitão and Sucena, 2015*) and dedifferentiation (*Terriente-Felix et al., 2013*) proposed in *Drosophila* hemato-poiesis. It will be interesting to determine in future studies if paths that are minor during homeostasis become more prominent under stress or immune challenge when a rapid and amplified response is prioritized over orderly development.

## Developmental metabolism and the transcriptome

Contrary to a commonly held viewpoint, metabolic pathways are regulated in a cell-specific manner and their participation is not limited to 'housekeeping' roles during development. Indeed, data on both cancer and developmental metabolism show that selective use of such pathways can drive certain critical developmental decisions instead of the other way around (*Pavlova and Thompson, 2016*; *Nagaraj et al., 2017*; *Miyazawa and Aulehla, 2018*; *Chi et al., 2020*; *Li and Simon, 2020*; *Nakamura-Ishizu et al., 2020*; *Tiwari et al., 2020*).

The analysis presented in this paper demonstrates that in *Drosophila* hematopoiesis, cells within individual zones are not only defined by their position within the organ and the markers that they express, but also by their metabolic status that is foreshadowed by the content of their transcriptome. The PSC cells, as a group, for example, are well represented by most upper glycolysis genes that are then used, not for bioenergetic purposes, but to increase the PPP flux of glucose metabolism that aids in maintaining an NADPH/GSH-dependent low ROS status for these cells. This is important as high ROS in the PSC is a trigger for a specific immune response that must be repressed during homeostasis (*Sinenko et al., 2011*; *Louradour et al., 2017*). Interestingly, the immediately adjacent MZ cells are lower in NADPH-forming enzymes, and their genes controlling oxidative phosphorylation are higher than in the PSC. This would lead to higher ROS even during homeostasis. Indeed, the MZ ROS levels are high and this physiological amount is essential for progenitor differentiation (*Owusu-Ansah and Banerjee, 2009*). A very interesting example of metabolic control is in the IZ cluster. Surprisingly, this narrow band of cells is enriched for genes required for both synthesis and clearance of free ceramide from a cell. This is important given the known role of ceramide in the activation of the JNK pathway, and we provide genetic and immunohistochemical evidence of transient activation of JNK and MMP1 in this group of cells.

Unlike cancer metabolism, developmental metabolism is at a surprisingly early phase of research, and *Drosophila* hematopoiesis could be a very attractive system to study this phenomenon during

homeostasis. More broadly, the results point to the continued relevance of the use of *Drosophila* as the singular invertebrate hematopoietic model, which provides a logical framework within which to establish less-studied concepts such as the characterization of parallel transitory populations, the roles of developmental metabolism, mechanisms of unusual signaling paradigms, and genetic dissection of pleiotropy.

# Materials and methods

## Key resources table

| Reagent type (species) or resource | Designation | Source or reference | Identifiers | Additional information |
|---|---|---|---|---|
| Genetic reagent (*Drosophila melanogaster*) | *dome^MESO^>GFP Hml^Δ^-DsRed* | Banerjee Lab | | *dome^MESO^-GAL4, UAS-EGFP, Hml^Δ^-DsRed* |
| Genetic reagent (*D. melanogaster*) | *lz-GAL4* | Gift from John Pollock, **Lebestky et al., 2000** | | |
| Genetic reagent (*D. melanogaster*) | *dome^MESO^-GFP.nls, Hml^Δ^-DsRed.nls* | Banerjee Lab, **Makhijani et al., 2011** | | Recombinant of *dmGFPnls* (Banerjee Lab) and *HmlDsRednls* (Brückner Lab) |
| Genetic reagent (*D. melanogaster*) | *dome^MESO^-EBFP2* | Banerjee Lab, **Evans et al., 2014** | | |
| Genetic reagent (*D. melanogaster*) | *CHIZ-GAL4, UAS-mGFP* | Banerjee Lab, **Spratford et al., 2020** | | IZ-specific split GAL4 |
| Genetic reagent (*D. melanogaster*) | *Tep4>GFP Hml^Δ^-DsRed* | Banerjee Lab | | Recombinant of *Tep4-GAL4, UAS-EGFP, Hml^Δ^-DsRed* |
| Genetic reagent (*D. melanogaster*) | *Tep4-QF2 ^G4H^* | This study | | *Tep4-GAL4* NP7374 line (DGRC Kyoto) was converted to QF2 using the HACK method (**Lin and Potter, 2016**) |
| Genetic reagent (*D. melanogaster*) | *Hml^Δ^>2xEGFP* | Banerjee Lab | | Recombinant of *Hml^Δ^-GAL4, UAS-2xEGFP* |
| Genetic reagent (*D. melanogaster*) | *UAS-Notch^ACT^* | Gift from Dr. Artavanis-Tsakonas, **Artavanis-Tsakonas et al., 1999** | | |
| Genetic reagent (*D. melanogaster*) | *lz-Gal4, UAS-mGFP* | Bloomington *Drosophila* Stock Center | BDSC:6314 RRID:BDSC_6314 | |
| Genetic reagent (*D. melanogaster*) | *UAS-pnt^RNAi^* | Bloomington *Drosophila* Stock Center | BDSC:35038 RRID:BDSC_35038 | |
| Genetic reagent (*D. melanogaster*) | *UAS-pnt^RNAi^* | Bloomington *Drosophila* Stock Center | BDSC:31936 RRID:BDSC_31936 | |
| Genetic reagent (*D. melanogaster*) | *UAS-hep^ACT^* | Bloomington *Drosophila* Stock Center | BDSC:9306 RRID:BDSC_9306 | |
| Genetic reagent (*D. melanogaster*) | *UAS-mihep* | Bloomington *Drosophila* Stock Center | BDSC:35210 RRID:BDSC_35210 | |
| Genetic reagent (*D. melanogaster*) | *UAS-numb^RNAi^* | Bloomington *Drosophila* Stock Center | BDSC:35045 RRID:BDSC_35045 | |
| Genetic reagent (*D. melanogaster*) | *LexAop2-6XmCherry* | Bloomington *Drosophila* Stock Center | BDSC:52271 RRID:BDSC_52271 | |
| Genetic reagent (*D. melanogaster*) | *UAS-Notch^RNAi^* | Bloomington *Drosophila* Stock Center | BDSC:7077 RRID:BDSC_7077 | |
| Genetic reagent (*D. melanogaster*) | *Msi-GFP* | Bloomington *Drosophila* Stock Center | BDSC:61750 RRID:BDSC_61750 | MiMIC protein trap |

*Continued on next page*

*Continued*

| Reagent type (species) or resource | Designation | Source or reference | Identifiers | Additional information |
|---|---|---|---|---|
| Genetic reagent (*D. melanogaster*) | *Hml^Δ-QF2* | Bloomington *Drosophila* Stock Center | BDSC:66468 RRID:BDSC_66468 | |
| Genetic reagent (*D. melanogaster*) | *10XQUAS-6XmCherry* | Bloomington *Drosophila* Stock Center | BDSC: 52269 RRID:BDSC_52269 | |
| Genetic reagent (*D. melanogaster*) | *bnl-LexA* | Gift from Dr. Roy, **Du et al., 2017** | | |
| Genetic reagent (*D. melanogaster*) | *UAS-msi^RNAi*, *UAS-Dcr2* | Gift from Dr. Wappner, **Bertolin et al., 2016** | | |
| Genetic reagent (*D. melanogaster*) | *Hml^Δ-DsRednls* | Gift from Dr. Brückner, **Makhijani et al., 2011** | | |
| Genetic reagent (*D. melanogaster*) | *UAS-Pvr^RNAi* | Gift from Dr. Shilo, **Rosin et al., 2004** | | |
| Antibody | Anti-Notch extracellular domain (NECD) (Mouse monoclonal) | Developmental Studies Hybidoma Bank (DSHB) | Cat# ABS571, RRID:AB_528408 | IF(1:50) |
| Antibody | Anti-Hrs 8-2 (Mouse monoclonal) | Developmental Studies Hybidoma Bank (DSHB) | Cat#, Hrs 8-2 RRID:AB_722114 | IF(1:100) |
| Antibody | Anti-MMP1 (Mouse monoclonal) | Developmental Studies Hybidoma Bank (DSHB) | Cat# 3B8D12, RRID:AB_579781; 3A6B4, RRID:AB_579780; and 5H7B1 RRID:AB_579779, | IF(1:100 of a 1:1:1 mixture of 3B8D12, 3A6B4 and 5H7B1) |
| Antibody | Anti-Hnt (Mouse monoclonal) concentrate | Developmental Studies Hybidoma Bank (DSHB) | Cat# 1G9-c, RRID:AB_528278 | IF(1:200) |
| Antibody | Anti-P1 (Mouse monoclonal) | Gift from Dr. Ando, **Kurucz et al., 2007** | | IF(1:100) |
| Antibody | Anti-PPO2 (Rabbit polyclonal) | Gift from Dr. Asano, **Asano and Takebuchi, 2009** | | IF(1:200) |
| Antibody | Anti-Numb (Guinea pig polyclonal) | Gift from Jan lab, **Roegiers et al., 2001** | | IF(1:200) |
| Antibody | Anti-Numb (Rabbit polyclonal) preabsorbed | Gift of Jan lab, **Rhyu et al., 1994** | | IF(1:200) |
| Antibody | Anti-Sima (Guinea pig polyclonal) | Banerjee Lab, **Wang et al., 2016** **Rhyu et al., 1994** | | preabsorbed; IF(1:100) |
| Sequence-based reagent | Sima-RA/RB_F | This study | PCR primers | GCAGAACTTCAAGGTGCAATAA |
| Sequence-based reagent | Sima-RA/RB_R | This study | PCR primers | CACCGTTCACCTCGATTAACT |
| Sequence-based reagent | sima-RC_F | This study | PCR primers | GAGGCGCACTAGTGACAAA |
| Sequence-based reagent | sima-RC_R | This study | PCR primers | CGAGCGAGATAGCAACGG |
| Sequence-based reagent | msi_F | FlyPrimerBank | PP29850 | ACGTCGTCTGACAAGCTCAAG |

| Reagent type (species) or resource | Designation | Source or reference | Identifiers | Additional information |
|---|---|---|---|---|
| Sequence-based reagent | msi_R | FlyPrimerBank | PP29850 | GAATGTGATGAAACCAAAGCCG |

## *Drosophila* strains

The *Drosophila* lines from our lab stock were used in this study as follows: *dome^MESO^>GFP Hml^Δ^-DsRed*, *dome^MESO^-GFP.nls Hml^Δ^-DsRed.nls*, *dome^MESO^-EBFP2, CHIZ-GAL4 UAS-mGFP* (IZ-specific GAL4; **Spratford et al., 2020**), *Tep4>GFP Hml^Δ^-DsRed*, *Tep4-QF2* (first used in this study), and *Hml^Δ^>2xEGFP*. The following lines were obtained from the Bloomington *Drosophila* Stock Center (BDSC): *lz-Gal4 UAS-mGFP* (#6314), *UAS-pnt^RNAi^* (#35038 or #31936), *UAS-hep^ACT^* (#9306), *UAS-mihep* (#35210), *UAS-numb^RNAi^* (#35045), *LexAop2-6XmCherry* (#52271), *UAS-Notch^RNAi^* (#7077), *Msi-GFP* MiMIC protein trap (#61750), *Hml^Δ^-QF2* (#66468), and *10XQUAS-6XmCherry* (#52269). The following lines were kind gifts from other labs: *bnl-LexA* from Dr. Roy (**Du et al., 2017**), *lz-GAL4* from Dr. Pollock (**Lebestky et al., 2000**), *UAS-Notch^ACT^* from Dr. Artavanis-Tsakonas (**Artavanis-Tsakonas et al., 1999**), *UAS-msi^RNAi^ UAS-Dcr2* from Dr. Wappner (**Bertolin et al., 2016**), *Hml^Δ^-DsRednls* from Dr. Brückner (**Makhijani et al., 2011**), and *UAS-Pvr^RNAi^* from Dr. Shilo (**Rosin et al., 2004**).

The potency of the RNAi lines listed above has been confirmed. Briefly, we demonstrate loss of Numb or Notch protein expression with *numb^RNAi^* (**Figure 5—figure supplement 1K-M**) or *Notch^RNAi^* (**Figure 5—figure supplement 2E-G**), respectively. Lines such as *pnt^RNAi^* are used routinely in the laboratory for diverse experiments in different tissues. For *pnt^RNAi^*, we use two different and independently generated RNAi lines for *pnt* that both give rise to the same reproducible phenotype, both qualitatively and quantitatively, with two different GAL4 drivers (*Hml^Δ^>* and *CHIZ>*, see **Figure 4Q**, **Figure 4—figure supplement 1F**). The potency of the miRNA against the JNKK hep is demonstrated by the complete loss of staining for MMP1, a known downstream target of JNK pathway (**Figure 3I–I″**). *msi^RNAi^* shows a statistically significant reduction of *msi* transcript levels by qPCR (~ 67% lower, **Figure 6—figure supplement 1G**) and phenotypes are seen with *msi^RNAi^* using multiple independent drivers (*Hm^Δ^l >* and *lz>*, **Figure 6H–M**). *Pvr^RNAi^* was previously used and validated (**Rosin et al., 2004**), also in our lab (**Mondal et al., 2011**; **Mondal et al., 2014**) and the phenotype shown (**Figure 2L**) is the same as published.

## Preparation of single-cell suspension from larval lymph glands

Larvae were collected at the third instar stage (90–93 hr AEL for scRNA-Seq; 90–96 hr AEL for bulk RNA-Seq of *domeMESO-GFP.nls Hml^Δ^-DsRed.nls*; 93–117 hr AEL for bulk RNA-Seq of *lz>GFP, Hml^Δ^-DsRednls*) and washed with DEPC water on a shaker to remove food traces before dissection. Pairs of lymph gland anterior/primary lobes were dissected (posterior/secondary and tertiary lobes were removed by mechanical separation using forceps) from 11 larvae, including 6 females and 5 males (for single-cell RNA sequencing) and approximately 100 larvae (for bulk RNA sequencing) in 1× modified dissecting saline (MDS) buffer (9.9 mM HEPES-KOH, 137 mM NaCl, 5.4 mM KCl, 0.17 mM NaH$_2$PO$_4$, 0.22 mM KH$_2$PO$_4$, 3.3 mM Glucose, and 43.8 mM Sucrose, pH 7.4) and lymph glands were then placed into a glass dish containing Schneider's medium (Gibco) kept on ice. Glass dishes were pretreated with 1% BSA in phosphate-buffered saline (PBS) and rinsed prior to use to prevent adherence of primary lobes. Three biological replicates were done in parallel. The lymph glands were dissociated as previously described with some modifications (**Harzer et al., 2013**; **Khan et al., 2016**). After being washed with MDS buffer twice, these tissues were transferred to 1.5 ml DNA LoBind tubes (Eppendorf) and incubated with 200 µl of dissociation solution containing 1 mg/ml of papain (Sigma-Aldrich, P4762) and 1 mg/ml of collagenase (Sigma-Aldrich, C2674) in Schneider's medium. They were dissociated for 15 min in a shaking incubator at 25°C, 300 rpm. Next, 500 µl of cold Schneider's medium was added and the suspension was gently pipetted up and down using a low-binding 1000 µl tip (Olympus Plastics) 20 times for mechanical dissociation. After centrifugation at 3000 rpm for 5 min, cells were resuspended and washed with 500 µl of 1× PBS (Corning, MT21040CV) containing 0.04% of UltraPure BSA (Invitrogen, AM2616) and then passed through a 35-µm cell strainer (Falcon 352235). For preparation of the single-cell RNA-Seq sample, the cell suspension was concentrated by centrifuging and resuspending in a lower volume of PBS containing 0.04% BSA (30 µl). Cell concentration and viability were assessed using the Countess II automated cell counter (Applied Biosystems). The

samples with final concentration of more than 650 cells/μl and viability of more than 85% were used for single-cell RNA-Seq.

## Flow cytometry and cell sorting

For all RNA extractions from sorted populations, dissociated live lymph gland cells (from approximately 100 pooled larvae) were sorted using the BD FACSARIA-H. Gates and compensation were based on single-color controls. Cells were sorted into 300 μl of DNA/RNA Shield (Zymo) in DNA LoBind tubes and frozen at – 80°C prior to RNA extraction.

For DNA content analysis, dissociated lymph gland cells were fixed in 1 ml of 1% formaldehyde solution in PBS after being dissociated using the above protocol. Cells were incubated in fixative in low binding tubes for 30 min at 4°C on a shaker, then were spun down and washed with PBS. Fixed cells were resuspended in a solution of PBS containing NucBlue Live ReadyProbes Reagent (Hoechst 33342) and incubated at room temperature on a shaker for 30 min. Cells were transferred to 5 ml polystyrene tubes for flow cytometry analysis on the BD LSRII. Cells were gated to exclude doublets using the FSC-H versus FSC-W and SSC-H versus SSC-W comparisons.

## RNA extraction and qRT-PCR

For the bulk RNA-Seq and qRT-PCR, total RNA was extracted using the Quick-RNA Microprep Kit (Zymo). RNA quality control was performed using the Agilent 4150 TapeStation system. For qRT-PCR analysis, cDNA was generated using SuperScriptIV VILO Master Mix (Thermo Fisher Scientific). qRT-PCR was performed on cDNA using the PowerUp SYBR Green Master Mix (Thermo Fisher Scientific) and the StepONE Real-Time PCR system (Applied Biosystems). Primers used were as follows: sima-RA/RB Forward 5′-GCAGAACTTCAAGGTGCAATAA-3′; sima-RA/RB Reverse 5′-CACCGTTCACCTCGATTAACT-3′; sima-RC Forward 5′-GAGGCGCACTAGTGACAAA-3′; sima-RC Reverse 5′-CGAGCGAGATAGCAACGG-3′; msi Forward 5′-ACGTCGTCTGACAAGCTCAAG-3′; msi Reverse 5′-GAATGTGATGAAACCAAAGCCG-3′ (*Hu et al., 2013*).

## Bulk RNA-Sequencing and Analysis

cDNA libraries were prepared using KAPA Stranded mRNA-Seq Kit (KAPA Biosystems) for the MZ/IZ/CZ bulk RNA-Seq experiment and the Universal Plus mRNA-Seq Kit (Nugen) for the CCs bulk RNA-Seq. Libraries were sequenced on two lanes of a HiSeq3000 (Illumina) or NovaSeq 6000 SP (Illumina), respectively. RNA sample and cDNA library concentration and quality control were assessed using the Agilent 4200 TapeStation system. Sequencing data was analyzed using Partek Flow, a web-based software platform. Sequences were aligned to the *Drosophila melanogaster* reference genome r6.22 (FlyBase) using STAR aligner with default parameters. Read counts were normalized by counts per million (CPM). Differential gene expression analysis was performed using ANOVA with a fold change cutoff of 2 and FDR<0.05 (see *Supplementary file 3*).

## Single-cell RNA-Sequencing

Three samples were processed using 10× Single Cell 3′ GEX version 3 (10× Genomics) and sequenced on a NovaSeq 6000 S4 PE (Illumina) at UCLA Technology Center for Genomics & Bioinformatics. More than 8000 cells per sample were put through the Chromium Controller Instrument ( 10× Genomics) to partition single cells into Gel bead-in-Emulsions (GEMs) and 10× barcoded libraries were then constructed. All cDNA libraries were sequenced on one lane of NovaSeq flow cell using a symmetric run (2×150 bp).

## Single-cell RNA-Seq data processing and analysis

Partek Flow was used to analyze single-cell sequencing data. Before alignment, each paired read was trimmed according to 10× Genomics Chromium Single Cell 3′ v3 specifications. Trimmed reads were aligned as described in the bulk RNA-Seq above. After alignment, UMIs were deduplicated and barcodes were then filtered and quantified to generate a single-cell count matrix. Genes with fewer than 30 total reads across three samples were excluded. Low-quality cells and potential doublets were filtered out by selecting out cells with high read counts (>85,000), an especially low (<1500) or high

(>4500) number of genes detected, or a high percentage of mitochondrial reads (>6%). 21,157 cells passed these quality control filters across the three samples.

Read counts were normalized with the following order: (1) CPM; (2) Add 1; and (3) Log2 transformation. Genes that were not expressed in any cells were excluded and a total of 9458 genes were detected. Data was then corrected for batch effects between samples using the General Linear Model. All rRNA and three sex-related lncRNA (*lncRNA:roX1*, *lncRNA:roX2*, and *lncRNA:CR40469*) were filtered out. The data was imputed using the MAGIC algorithm (*van Dijk et al., 2018*) with the number of nearest neighbors=50.

For graph-based clustering, the top 2000 genes with the highest dispersion were used to perform principal component (PC) analysis (PCA). Based on the Scree plot, the first 13 PCs were selected to use as the input for clustering and data visualization tasks. Graph-based clustering was first performed with 50 nearest neighbors (NNs) and resolution (res) of 0.1–0.5, giving 6–14 clusters, with 8 clusters (res=0.19) showing the most marked differences in gene expression between clusters. The data was visualized using t-SNE with perplexity of 50 and PCA initialization.

Each cluster was compared to the others using an ANOVA to identify differentially expressed genes that were enriched more than 1.5-fold (with a FDR cutoff<$10 \times e^{-6}$) (see *Supplementary file 1*). Cluster identity was assigned based on the presence of known zone-specific markers in the differentially expressed genes. Gene set enrichment analysis (GO, KEGG) was performed on the differentially expressed genes for each cluster (see *Figure 1—figure supplement 3A*).

Gene expression is shown in a couple of different ways. Heatmaps and dot plots were constructed for differentially expressed genes to show expression patterns/trends across the different subpopulations. For heatmaps, the expression values for a specific gene are converted to standardized z-scores as follows: the imputed expression value for that gene in each single cell is subtracted from the mean expression for that gene in all cells and then divided by the standard deviation for all cells using imputed expression data. All heatmaps are displayed using the z-score range of –2 to +2 (with the exception of *Figure 2—figure supplement 3A*, which uses –3 to +3). For dot plots, the mean expression (mean) and percentage of cells that show expression (non-zero percent) of a given gene is calculated for each subpopulation using non-imputed gene expression data.

To evaluate the developmental progression of the cells from a progenitor to a differentiated cell type, a subsequent trajectory and pseudotime analysis using Monocle 2 (*Qiu et al., 2017*) were performed. PSC and X clusters were both excluded from trajectory analysis. The trajectory was calculated using the top 1824 genes with the highest dispersion. For cluster X, a separate trajectory was constructed using the top 1500 highest dispersion genes.

We performed subclustering on isolated clusters (CC, X, and PL) to determine subpopulations. The subclustering analysis follows the general procedure as the initial graph-based clustering and used 2000 genes with the highest dispersion within each of the isolated clusters. Two CC subclusters were identified using 5 PC, 60 NN, and res 0.1. Three X subclusters were generated using 4 PC, 30 NN, and res 0.5. PL showed four subclusters with 7 PC, 50 NN, and res 0.2.

Bulk and scRNA-Seq data can be found at: GEO accession GSE168823.

## AUCell analysis

To analyze the activity of a gene set in our data, we used the AUCell tool (*Aibar et al., 2017*), with the top 25% of genes. The minimum gene set size was generally 5, but in a few cases where the number of genes in the pathway was between 3 and 5, a smaller minimum was used. Gene lists used for AUCell analysis are found in *Supplementary file 4*. Each of the lists was derived from a specific source including GO terms, KEGG pathways, and published literature. The individual source is given for each list in *Supplementary file 4*. The Pearson's correlation coefficient (r) shown in indicated graphs was calculated using Partek software. For AUCell analysis, z-scores ranging from –2 to +2 are displayed on heatmaps where the z-score is a measure of the standard deviations from the mean of each individual cell's AUC score. Absolute or relative z-scores of >1.65 or <−1.65 are considered biologically relevant and statistically significant, approximately equivalent to p-values<0.05 with a one-tailed t-test.

## Lymph gland dissection, immunostaining, and imaging

Lymph glands were dissected and processed as previously described (*Jung et al., 2005*). Unless indicated otherwise in the figure legend, all stainings were formed on lymph glands from wandering

third instar larvae. Briefly, for MMP1 staining, lymph glands were dissected and immediately placed into 4% formaldehyde in PBS on ice and then fixed for 15 min at room temperature. For all other stainings, lymph glands were dissected into cold PBS and then fixed in 4% formaldehyde in PBS at room temp for 20 min. After fixation, tissues were washed three times in PBS with 0.3% Triton X-100 (PBST) for 10 min each, blocked in 10% normal goat serum in PBST (blocking solution) for 30 min, followed by incubation with primary antibodies in blocking solution. Primary antibodies were incubated with tissues overnight at 4°C and then washed three times in PBST for 10 min each, followed by incubation with secondary antibodies for 3 hr at room temperature. Samples were washed four times in PBST, with DAPI (1:1000, Invitrogen) added to the third wash to stain nuclei, and then placed into VectaShield mounting medium (Vector Laboratories) and mounted on glass slides. Notch internalization assays were performed as described (*Mukherjee et al., 2011*) using mouse anti-NECD (1:50, DSHB C458.2H). Staining for endosomal proteins was performed as described (*Riedel et al., 2016*) using mouse Hrs 8-2 antibody (DSHB). Lymph glands and dissociated cells were imaged using a Zeiss LSM 880 Confocal Microscope. All microscopy data are representative images from a total of approximately 10 biological replicates (n) in most cases. For each n, full z-stacks were imaged, processed, and analyzed using ImageJ or Imaris software. For MZ, CZ, and CC indexes, Imaris software was used to reconstruct a 3D volume from a z-stack and the spots feature was used to quantify individual cell types based on their fluorescence intensity values. To assess levels of protein staining in individual cells, ImageJ (e.g., for Numb and Sima staining) or Imaris (e.g., for Numb and PPO2 staining) was used. For all quantitation graphs, mean with standard deviation are shown and significance was calculated by unpaired t-test. All statistics were performed using Prism (GraphPad) software and p-values are shown in charts or figure legends as indicated. For space considerations, in some graphs p-values are represented in GraphPad style using asterisks as follows: n.s. if p>0.05; * if p≤0.05; ** if p≤0.01; *** if p≤0.001; **** if p≤0.0001.

The primary antibodies were used as follows: mouse anti-MMP1 (1:100 of a 1:1:1 mixture of 3B8D12, 3A6B4, and 5H7B1, DSHB) (*Page-McCaw et al., 2003*), mouse anti-P1 (1:100, kind gift from Dr. Ando) (*Kurucz et al., 2007*), rabbit anti-PPO2 (1:200, kind gift of Dr. Asano) (*Asano and Take-buchi, 2009*), mouse anti-Hnt (1:200, DSHB 1G9-c) (*Yip et al., 1997*), guinea pig anti-Numb (1:200, kind gift of Jan lab) (*Roegiers et al., 2001*), preabsorbed rabbit anti-Numb (1:200, kind gift of Jan lab) (*Rhyu et al., 1994*), and guinea pig anti-Sima (preabsorbed; 1:100) (*Wang et al., 2016*). Primary antibodies were detected with secondary antibodies conjugated to Cy3 (1:100; Jackson ImmunoResearch Laboratories), Alexa 633 (1:100), Alexa 555 (1:200), or Alexa 488 (1:200) (Invitrogen).

## Acknowledgements

The authors thank Fangtao Chi for engineering the *CHIZ-GAL4* construct used in this study, and past and present members of the laboratory for their help and advice. The authors gratefully acknowledge FlyBase; the Bloomington *Drosophila* Stock Center; the Vienna *Drosophila* Resource Center; the Kyoto Stock Center, and the fly community including Sougata Roy, Pablo Wappner, Dalmiro Blanco-Obregon, Katja Brückner, Tsunaki Asano, Yuh Nung Jan, Dirk Bohmann, and Andrea Page-McCaw for reagents. The authors acknowledge the help of the Broad Stem Cell Research Center (BSCRC) and Owen Witte for help and support, the MCDB/BSCRC Core Facility in Microscopy, and the BSCRC core in Flow Cytometry, particularly Felicia Codrea, Jessica Scholes, and Jeffrey Calimlim for help with cell sorting. The authors thank the UCLA TCGB center, Xinmin Li, and Michael Mashock for help in sequencing. The authors acknowledge the Partek Flow technical support team and in particular Xiaowen Wang's help, which was crucial for data analysis. The authors thank undergraduate research scholars Peiliang Zhou, Chloe Su, and Khoi Luc for their contributions. The authors thank David Eisenberg and Vy Phan Lai for their support of DMV through the Center for Global Mentoring. UB is supported by National Institutes of Health grants R01 HL-067395 and R01 CA-217608; JRG by Ruth L Kirschstein National Research Service Award number T32HL69766 and UPLIFT (UCLA Postdocs' Longitudinal Investment in Faculty) Award number K12GM106996; LMG by Ruth L Kirschstein Institutional National Research Service Award number T32CA009056 and by National Heart, Lung, and Blood Institute of the National Institutes of Health under award number 3R01HL067395-16S1; DMV by the Center for Global Mentoring at UCLA-DOE Institute for Genomics & Proteomics; and CMS by Ruth L Kirschstein National Research Service Award number T32HL863458.

## Additional information

### Competing interests

Utpal Banerjee: Reviewing editor, *eLife*. The other authors declare that no competing interests exist.

### Funding

| Funder | Grant reference number | Author |
|---|---|---|
| National Heart, Lung, and Blood Institute | R01-HL067395 | Utpal Banerjee |
| National Cancer Institute | R01-CA217608 | Utpal Banerjee |
| National Heart, Lung, and Blood Institute | T32-HL69766 | Juliet R Girard |
| National Institute of General Medical Sciences | K12-GM106996 | Juliet R Girard |
| National Cancer Institute | T32-CA009056 | Lauren M Goins |
| National Heart, Lung, and Blood Institute | T32-HL863458 | Carrie M Spratford |

The funders had no role in study design, data collection and interpretation, or the decision to submit the work for publication.

### Author contributions

Juliet R Girard, Lauren M Goins, Conceptualization, Data curation, Formal analysis, Funding acquisition, Investigation, Methodology, Project administration, Resources, Supervision, Validation, Visualization, Writing - original draft, Writing – review and editing; Dung M Vuu, Data curation, Formal analysis, Investigation, Methodology, Resources, Software, Validation, Visualization, Writing - original draft, Writing – review and editing; Mark S Sharpley, Formal analysis, Methodology, Resources, Writing – review and editing; Carrie M Spratford, Investigation, Resources, Writing – review and editing; Shreya R Mantri, Investigation, Resources; Utpal Banerjee, Conceptualization, Formal analysis, Funding acquisition, Project administration, Resources, Supervision, Writing – review and editing

### Author ORCIDs

Juliet R Girard ⓘ http://orcid.org/0000-0001-8274-2136
Utpal Banerjee ⓘ http://orcid.org/0000-0001-6247-0284

### Decision letter and Author response

Decision letter https://doi.org/10.7554/eLife.67516.sa1
Author response https://doi.org/10.7554/eLife.67516.sa2

## Additional files

### Supplementary files

• Supplementary file 1. Differentially expressed genes in scRNA-Seq data.

• Supplementary file 2. Comparison of clusters from this study with those identified as clusters by *Cho et al., 2020*.

• Supplementary file 3. Differentially expressed genes in bulk RNA-Seq data.

• Supplementary file 4. Gene lists used in this study for AUCell analysis.

• Transparent reporting form

### Data availability

Sequencing data have been deposited in GEO under Accession Code GSE168823 Complete Source Data are provided.

The following dataset was generated:

| Author(s) | Year | Dataset title | Dataset URL | Database and Identifier |
|---|---|---|---|---|
| Girard JR, Goins LM, Vuu DM, Banerjee U | 2021 | Paths and Pathways that Generate Cell-Type Heterogeneity and Developmental Progression in Hematopoiesis | https://www.ncbi.nlm.nih.gov/geo/query/acc.cgi?acc=GSE168823 | NCBI Gene Expression Omnibus, GSE168823 |

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
