## [Editor Report]

This paper will be of interest to scientists who study hematopoiesis. The authors combine single cell RNA-seq with bulk RNA-seq of transcripts from blood cells in the *Drosophila* larval hematopoietic organ. They present extensive analysis of the datasets, and the pseudotime analyses present a model of how hematopoietic progenitors can differentiate along transitory paths. These datasets reveal cell-type specific isoform expression of Notch pathway regulators, and genetic experiments prove the importance of these factors in development of one lineage. These transcriptomic analyses and subsequent genetic experiences provide strong support for the major claims of the paper.

---

## [Decision Letter]

**Decision letter after peer review of original submision:**

Thank you for submitting your article "Paths and Pathways that Generate Cell-Type Heterogeneity and Developmental Progression in Hematopoiesis" for consideration by *eLife*. Your article has been reviewed by 3 peer reviewers, including Erika A Bach as the Reviewing Editor and Reviewer #1, and the evaluation has been overseen by Anna Akhmanova as the Senior Editor.

Essential revisions:

1) Address issues with AUC scores:

a) Define how you established the AUC list.

b) Clarify how you define AUC scores that are statistically significant.

c) Include a zero (0) on the y-axis of all graphs and P values for all graphs.

2) Provide a comparison of the data between the replicates.

3) Provide validation of some of the genes associated with clusters/sub-clusters, including Delta in MZ1, PSC, X and MZ2 clusters, and Ubx in posterior lymph gland lobes, and provide validation to support your model that there is *no* transition path for IZ to proPL.

4) Demonstrate the potency of RNAi lines, including pnt RNAi.

5) Explain the main differences and similarities between your data set and Jiwon Shim's by addressing specific issues such as Cluster "PH1" (your MZ1, PSC, X and MZ2 clusters) and Cluster X (and include a dot plot representation of Cluster X).

6) Provide additional quantification of CC in "Numb and Musashi in CC determination", additional information on how Numb intensity was quantified, and P values

7) Provide for lymph gland analyses the quantification of CC and MZ indexes and P values.

8) Consider using dot plots instead of box plots.

9) Provide in the main figures a list of the key genes defining each sub-population inside the different clusters.

10) Provide the number of cells in each cluster, including the IZ cluster.

11) Provide a schematic representation of crystal cell specification and maturation involving Notch, Numb and Sima.

12) Make extensive editorial changes, including shortening the manuscript per the reviewers' suggestions and clarifying the narrative so that the introduction covers what is actually being treated in the manuscript. For example, the introduction discusses the IZ but then the manuscript does not provide much insight into this cluster and instead focuses on CCs.

Full Reviews

*Reviewer #1/Reviewing Editor:*

Girard and colleagues combine bulk RNA-seq and scRNA-seq of FACS sorted blood cells from the larval lymph gland, the hematopoietic organ in *Drosophila*, to identify cell types and differentiation paths. The lymph gland is comprised of medullary and cortical zones. At least five cell types are found in the lymph gland: niche-like cells (termed the PSC), progenitors (termed prohemocytes) that are found in the MZ, intermediate progenitors (termed IZs) that represent a transition state between prohemocyte and terminally differentiated cells, and terminally-differentiated macrophage-like cells plasmatocytes (PLs) and crystal cells (CCs) found in the CZ. The bulked sorted populations are pure and are enriched for genes known to be expressed in these populations. Analysis of the scRNA-seq data identifies 9 clusters. Some of the clusters correspond to known populations like the PSC, IZ and PL. However, some clusters represent distinct subpopulations of MZ cells (MZ1 and MZ2) and CCs (iCCs and mCCs) and previously unknown clusters proPL and X. The pseudotime analysis identifies three "branch points" and seven "states", and t-SNEs show projected relationships between the states and clusters. Numerous graphs in multiple figures provide transcript levels of genes related to metabolism (Figure 3), pnt (Figure 4), numb (Figure 5), msi (Figure 6 ). The supplements for Figures 2 and 3 provide transcript levels for MZ markers, cluster X markers, IZ markers, CC markers, PL markers and basement membrane genes. Validation of the scRNA-seq are provided for JNK pathway (Figure 3), pnt (Figure 4), numb and sima (Figure 5), msi (Figure 6). The authors extend the transcriptomics through two "case studies". In the first one, the ETS transcription factor Pnt is required for the progression from late MZ to IZ. They also show that Pnt has a later function in PLs in preventing them from becoming CCs. In the second case study, their transcriptomics reveals a specific isoform of sima expressed in mature CCs (i.e., mCCs) that co-localizes with Notch, supporting the Banerjee lab's earlier work. The transcriptomics also reveals that numb transcripts (encoding a Notch inhibitor) are expressed in iCC but the Numb protein is highest in mCC. They author resolve this paradox by showing that the numb translational inhibitor Musashi is expressed robustly in iCCs. The authors also supply analyses comparing their scRNA-seq with one published in 2020. These transcriptomic analyses and subsequent genetic experiences provide strong support for the major claims of the paper.

There is a massive amount of data in this study. The data are of high quality, and the manuscript is well written. The results support the main conclusions. This study will be a very valuable resource to the community. I have a few suggestions for possible improvement.

1. On p. 36 (line 906), the authors write that "we surmise that Notch activation is the default pathway for early Hml-expressing cells to become CCs, and that the activation of Pnt acts antagonistically to prevent this process thus favoring instead, the plasmatocyte fate." I don't understand the logic of this. PLs represent 95% of the mature hemocytes, whereas CC represent 5%. Why would most of the differentiating hemocytes have to repress Notch signaling by expressing Pnt. Loss of Pnt from Hml+ cells would very consequential as the animal would not have PLs from the LG. What I am trying to say is that the kind of regulation proposed here is not robust and could be easily disrupted by mutation. Could the authors comment on this.

2. The manuscript is very long (the results section is thirty-five long) and reader attention spans tend to be short. Could the authors please edit the manuscript to reduce its length. For example, I don't think that the entire section about cluster X is needed. The metabolism section could be condensed. Some of the discussion is redundant with the results.

3. Please provide accession numbers the raw data at NCBI and a link to the reviewers.

*Reviewer #2:*

In this manuscript, Girard et al. analysed the transcriptome of the lymph gland of *Drosophila* using high throughput sequencing. They fist used genetic markers (Domemeso-GFP and Hml-RFP) to sort the cells from three distinct regions in feeding larvae: the medullary zone (MZ) known to be populated by progenitors, the intermediate zone (IZ) populated by intermediate precursors and the cortical zone (CZ) containing the mature hemocytes. The cells from the CZ were further subdivided into plasmatocytes, immature crystal cells and mature crystal cells with the genetic markers Hml-RFP and Lz-GFP. This subdivision was carried out at later stage to maximise the number of mature hemocytes. A comprehensive molecular signature of each region and cell type was determined with bulk RNAseq. Then, the authors analysed the transcriptome of 21200 cells of the lymph gland using 10x genomics technology. They found 9 clusters of cells displaying distinctive signatures and metabolic properties, including the cells of the Posterior Signaling center (PSC), two types of progenitors in the MZ, the intermediate precursors, the plasmatocytes as well as the immature and mature crystal cells. They predicted the filiation between the clusters using Monocle, indicating a developmental trajectory starting in MZ producing IZ and then plasmatocytes. At last, the authors validated two mechanisms of cell differentiation highlighted by the transcriptome analysis. They showed the context-specific role of the transcription factor Pnt in the differentiation of plasmatocytes. Loss of Pnt in the early progenitors has no effect whereas its loss in the late progenitors prevents the differentiation of intermediate precursors and thus mature plasmatocytes without affecting the differentiation of crystal cells. As a second study case, they showed that the interplay between Notch, Numb, Sima and Musashi is involved in the maturation of crystal cells.

Overall, this manuscript presents a tremendous amount of RNAseq and in vivo data with highly detailed interpretations. It provides very valuable and substantiated information on the molecular mechanisms involved in the development of the hemocyte lineages in the *Drosophila* lymph gland. The main caveats are (1) the definition of two clusters IZ and proPL, which seem to belong both to the IZ, (2) the fact that most boxplots do not include 0 in the y-axis, which strongly biases the interpretation of the graphs, (3) the definition of mitotic precursors. Finally, shortening would have made the manuscript more easily readable and would have conveyed the message more directly.

1) Most charts presenting expression levels or AUC score across clusters do not include 0 on the y-axis and disclose highly heterogeneous ranges. For example, the ranges of the AUC displayed in Figure 3 are from 37 points in A (y-axis range 0.43 to 0.8) to 3 points in I (y-axis range 0.13 to 0.16). This is highly misleading, and biases the interpretation of the data since the authors describe minor differences the same way than large differences. Few examples are:

– In the legend of Figure 3B-C, the authors write "(B) TCA cycle enzymes are expressed at exceptionally low levels in the PSC compared with the cells of other clusters. (C) Expression of oxidative phosphorylation pathway enzymes is low in the PSC, high in MZ1 and MZ2 and moderate in IZ, proPL, and PL clusters.". In B, exceptionally low level correspond to 0.42 compared to 0.46 in the other clusters. In C, low in the PSC means 0.74 and high in the MZ corresponds to 0.78. Since AUC represent frequencies, I doubt that few point can translate into such high ranges.

– In figure 3D and E, the authors mention the levels of Zw and Idh in the clusters. They explain that both are highly enriched in PSC compared to MZ. Zw presents an expression level of 1.1 in PSC, around 1.2 fold higher than in MZ and Idh presents an expression level of ~ 7 in PSC around 2 fold higher than in MZ. The difference in level of expression and fold enrichment cannot be appreciated properly due to the heterogeneous y-axes. In addition, the authors described the two enzymes in the same terms while Idh is expressed 7 time higher than Zw.

– For Pnt (Figure 4AB), the authors describe "a very low level of Pnt" in the PSC and MZ and a significant increase in the MZ2-3. The level in PSC is around 5.2, which is much higher than most genes described in this study and the significant increase is going from 5.2 in MZ2-2 to 5.5 in the MZ2-3. The significance of this increase need to be documented with a p-value and the biological relevance of this difference seem far-fetched.

2) A better explanation should be provided for the AUC scores. The authors rightly say that AUC are reflective of co-regulation however the keys to interpret the score are not described. In addition, while a difference from 0.80 to 0.60 (Figure 3A) seems plausible and sufficient to call for an enrichment of glycolytic genes in the PSC, the biological relevance of differences from 0.74 to 0.78 (Figure 3C) or from 0.08 to 0.09 (Figure 2—figure supplement 5) seems far-fetched without further explanations/justifications.

3) The distinction between the cluster IZ and proPL needs to be clarified. The enrichments of the IZ AUC and individual IZ markers are not striking (Figure 2—figure supplement 3A-H). In addition, can the authors explain why only 6 of the 9 IZ specific genes were taken to estimate the AUC score? The strong similarities between the two clusters, in terms of markers and developmental trajectories, seem to indicate that the two clusters represent transient conditions that each cell goes through on its way towards full differentiation. Indeed, the single cell analysis has been performed in feeding larvae, when cells are actively differentiating, not in a steady state condition. Longitudinal/spatial analyses in the developing lymph gland might help in the interpretation, but this is not the scope of the present manuscript.

4) The authors assess the impact of Pnt in the MZ cells using the drivers domemeso-Gal4 and Tep4-Gal4. The authors showed that domemeso>pntRNAi prevent the progression of MZ cells toward IZ cells. Some quantification would be welcome to appreciate the penetrance of the phenotype. In addition, the authors say that Tep4>pntRNAi has no observable phenotype, while the comparison between Figure 4E and Figure 4F seem to indicate that the lobe is smaller and with less Tep4 positive cells. Such difference could arise from slight stage differences. Could the authors indicate how they staged the animals, the number of samples...? At last, the potency of the pntRNAi construct should be documented.

5) In the section "Numb and Musashi in CC determination", the authors mention that Notch-ACT raises Numb levels and Notch-RNAi decreases Numb levels in the crystal cell. This interpretation should be clarified. Since N activity is modulated using a CC specific driver and the number of lz>GFP also changes upon N modulation, the observed results may arise from regulation of Numb expression and/or from regulation of CC number. CC quantification will help sustaining their statement on the role of N. In the third paragraph linked to Figure 5—figure supplement 3H-N and Figure 5O-R', the authors describe a clear reduction of Sima puncta, PPO2 expression and number of mCC. P-values should document these observations. Also, does the expression of the type II targets decrease upon Numb RNAi? At last, Figure 6H-M, the authors indicate that msi-RNAi enhance the level of Numb in crystal cells without providing information on the procedure followed to measure Numb intensity. What does Numb intensity represent in Figures 6J and 6M? Is it the average level per cell or the level in the whole lobe?

6) The authors carried out the single cell sequencing in triplicate. It would strengthen considerably the data to provide a comparison between the replicates.

7) The paper would gain from shortening. The introduction is broad and exhaustive, the results section describes the different the clusters and states as well as two study cases, the discussion elaborates on the mode of differentiation and put forward interesting models such as the gradual rather than stepwise transitions between groups of cells. Since this is a resource paper, the validation of all the single cell data is out of the scope, hence a thorough discussion of all those data could be shorten and used in subsequent studies. Furthermore, many data are already discussed in the results section, diluting the important and novel messages that the paper conveys.

8) According to the authors, Cluster X represents mitotic states of several distinct cell types, including the CZ that carries differentiated cells. This intriguing finding indicating the presence of dividing cells throughout the lymph gland deserves some clarifications. Does it imply that none of the other clusters identified by the RNAseq analysis contains cells in mitosis? Does it mean that plasmatocytes and cells of the medullary zones have similar mitotic potential? Is there any difference in the type/levels of genes associated to cell division between the cells of the cluster that express MZ markers vs. those that express the Pl markers? I understand that the spatial analysis of the cluster X cells in the lymph gland, which would help clarifying these issues, goes beyond the scope of the manuscript. It would be nevertheless useful to compare the RNAseq data with those from the laboratory of Jiwon Shim, who also identified the clusters of mitotic cells in the lymph gland. Also, a dot plot representation of the genes associated with cell division in the different cells of the cluster X (MZ, Tr, Pl) might help identifying features specific to the different subclusters.

*Reviewer #3:*

Using a combination of bulk RNA-Seq of FACS-sorted cells and single cell RNA-seq, the authors identify various blood cell subpopulations that compose the *Drosophila* hematopoietic organ called the lymph gland. This study has been performed at one developmental time point, mid third instar larvae. The authors perform a pseudo time analysis and propose a developmental trajectory with multiple paths to mature blood cells types. RNAseq data suggest that different blood cell types express genes involved in various metabolisc processes. They establish that Pointed has different roles during lymph gland hematopoiesis. Finally, they identify that Numb and Musahi are involved in a Sima dependent Notch non canonical pathway in mature crystal cells.

This analysis is of interest, however in the current version, it is too preliminary.

1. The list of the main genes defining each sub-population inside the different clusters, as well as their expression in all the other sub clusters, has to be provided in the main figures.

2. No validation of RNA seq data is provided: A spatial reconstruction in vivo by profiling the expression of a subset of genes identified by RNA seq is necessary.

3. To support the developmental progression of lymph gland cells proposed in Figure 7,

lineage tracing experiments are required.

4. Comparison between RNA seq data obtained by (Cho et al 2020) has to be given in the main text. Furthermore, discrepancies between these 2 studies have to be clarified. How the AUC list has been established? This is a key point. For example, the PH1 cluster identified by Cho et al is spread out in MZ1, PSC, X and MZ2 cluster in this study. Why are the results so different? Delta is a marker of PH1, which is validated by analyzing its in vivo expression profile. What about delta expression in the scRNA seq performed here?

5. There is a discrepancy between the introduction and the main results of this paper. In the introduction the authors focus our attention on the IZ, but in fine we don't learn much about IZ cells from this analysis. Instead of deciphering IZ identity and fate by in vivo profiling, most functional analyses performed concern crystal cell maturation. This part is developed via 2 main figures among 7, plus 5 sup figures. From my point of view, this study represents an ideal opportunity to better characterise IZ cell identity, lineage and function. Unfortunately these data are missing in the current version of the manuscript but could be added instead of the data concerning crystal cell maturation, which is somewhat out of the scope of this manuscript and could be published in a separate paper.

6. This manuscript has to be focused on the novelty given by the RNAseq data. The data concerning crystal cell maturation, which is somewhat out of the scope of this manuscript, could be published in a separate paper.

7. There are 7 main figures and 16 sup figures + 1 additional file. All these Sup figures give information and make suggestions that unfortunately are not validated by additional experiments. Overall the reader is left with a lot of observations that are not further validated and in fine one cannot rely on. Data presented in this manuscript have to be focused to avoid overloading the reader with too many side observations, which in turn lead to losing the thread of the message of this study.

8. I have concerns about the single cell RNA seq data, since essential information is missing.

– What about the cell numbers in each cluster? The IZ cluster (Dome-GFP+ and Hml+) represents a small subset of lymph gland cells based on the CHIZ expression profile (see Figure 3K); however, it corresponds surprisingly to a quite large lymph gland cell subset, as illustrated in Figure 1J. How can one explain this?

– To identify subpopulations in clusters, the authors performed sub clustering on isolated clusters for PL and CC. Why was this not done in the same way for the other 3 main clusters (MZ2, IZ and proPL)?

– For the IZ sub-cluster: The plot in Figure 2 sup 3I is very misleading, since it suggests that genes expressed in the IZ are specific to this cluster. For example, "state 3" is present both in the MZ and proPL (Figure 2), but in Figure 2 sup 3l it is only represented in the IZ and not in the MZ and proPL clusters. The same remark holds for states 4, 5 and 7. Furthermore, as I mentioned above, the list of the main genes defining each sub-population inside clusters, as well as their expression in the other sub-clusters, has to be provided in the main figures. Furthermore, a spatial reconstruction in vivo by profiling the expression of a subset of genes identified in sub populations is mandatory to validate the RNAseq data.

– Why is the plot shown in Figure 1J different from the plot shown in Figure 2 sup 5 I? The t-SNE graphic representation does not give any indications concerning whether clusters are related or not.

– Why are there discrepancies between gene expression levels and their representation on the corresponding plot? Please see for example the case of CG30090 in Figure 2 sup 3B and the corresponding plot in C. CG30090 is expressed at a similar level in proPL and PL, but its expression in proPL is lacking in the plot. Why ?

– Concerning IZ markers, among the 6 identified by bulk RNA seq, only 4 of them have been analysed in single cell experiments. What about the 2 others?

– In Figure 7, the model of developmental progression of blood cells proposes that there is no transition path between IZ and proPL. This proposition does not fit with the data. Indeed, in the pseudo time analysis there is a clear overlap between IZ and proPL, indicating that they are connected (see states 4 and 5 in Figure 2 O-P). Genes highly expressed in IZ are also highly expressed in proPL. This is observed for all the IZ markers analysed in this manuscript (please see Figure 2 sup 3B, D, G, H). Determining whether there is a transition path between IZ and proPL has to be validated by in vivo experiments.

– Figure 7: Regarding the translational link between PL7 and iCC7 (Figure 2 sup4), again this proposition has to be validated in vivo. Furthermore, how can iCC7 (more engaged in maturation) give rise to iCC6 (less engaged in differentiation). This also needs to be validated.

– Concerning the MZ1 cluster, Ubx is expressed in these cells. Ubx is specifically expressed in lymph gland posterior lobes that are composed of hematopoietic progenitors expressing markers of MZ cells (Rodrigues et al., 2021). Altogether these data strongly suggest that MZ1 cells correspond to posterior lobe hemocytes. This has to be clarified.

– For cluster X, since DNA damage markers are expressed, this strongly suggests that this cluster might correspond to unhealthy cells damaged during the experiment. What about their ribosomal content (a criteria commonly used to check for cell health)? Is the molecular signature of cluster X found in bulk RNA seq and in the Cho et al., 2020 paper?

– What is the unit given for the Y axis in Figure 3? Why is the scale different from one graph to another and does not start always at zero? For all graphs in this manuscript the p value is missing, so the reader cannot not figure out whether the differences are statistically significant or not. Concerning the AUC analysis, how is the list of genes taken into account for a signalling pathway or a function that has been established? What kind of conclusion can be drawn from analyses regarding metabolism? In other words, considering the PSC as an example of a group of cells where glycolysis genes are highly expressed, what is the impact of this on PSC function, and how does potential glycolysis in the PSC help us to understand PSC function?

– Figure 3K: the authors need to define what CHIZ is. Since there is no staining overlap between MMP1 and CHIZ-GFP, the authors cannot conclude that MMP1 is expressed in the IZ. Furthermore, MMP1 transcripts are detected at high levels not only in the IZ but also in the MZ2, proPL and PL (Figure 3J). How can these results be reconciled with the expression profile of MMP1 shown in Figure 3K?

–Figure 4: Quantifications are missing. Crystal cell and MZ indexes have to be given.

– Figure 4G and H: That tep4 is expressed in a subset of MZ progenitors defined by dome>GFP is not new, this has been established previously. Please see Benmimoun et al. 2015, and Oyallon et al 2016.

– What about the Pnt expression profile in the LG? Figure 4C-D: DomeMESO>pnt RNAi , in addition to a defect in blood cell differentiation, this leads to a smaller LG compared to the control. This defect in size has to be mentioned and quantified. Since the role of pnt in the MZ has been previously reported by Dragojlovic-Munther M et al , 2013, this paper has to be cited. Furthermore, the Dragojlovic-Munther M et al. study indicates that in addition to preventing hemocyte differentiation, pnt RNAi in the MZ leads to lamellocyte differentiation. Do lamellocytes differentiate when pnt is knocked down using domemeso, tep4 , CHIZ and hml gal4 drivers? In Figure 4F :Tep4>GFP>pntRNAi , GFP levels are decreased compared to the control. Does Pnt control the expression of the Tep4-gal4 driver? In the text, p35 line 886 "Pnt loss in MZ2.3" is an over interpretation, since no gal4 driver specific for this group of cells has been used to perform the lof experiment. p35 lines 903 "plasmatocytes to be converted into crystal cells" is erroneous, since hml-Gal4 expression is not restricted to mature plasmatocytes but is expressed both in plasmatocyte and crystal cell precursors. p35, lines 905-910 should be in the discussion, not in the result 'section.

– Figure 5: Numb and crystal cells

The paper Cho et al 2020 has to be mentioned in the text, since it established previously by immunostaining that Numb is expressed in crystal cells.

– A previous study done in Banerjee's lab, reports on the role of Sima and Notch in crystal cell survival by preventing their dissociation (Mukherjee et al., 2011). In the present manuscript, no data and comments refer to crystal cell survival depending on Numb. Thus in the current version of the manuscript, it remains unclear as to what is the role of Numb in crystal cells. Does it control iCC to mCC, or is it required for survival of mature crystal cells? The confusion is sustained by sentences such as: "depletion of Numb prevents the maturation of iCC to the mCC state", please see p 39, lines 1000. Since Numb is detected at high levels in mCC and not in iCC, the function of Numb should be in mCC once they have matured. Furthermore, it has been previously published that Sima is required for crystal cell survival. A decrease in Sima levels is observed in Lz>numbRNAi conditions, supporting the proposition that CC depleted from Numb should disrupt since they lack Sima . In conclusion, the role of Numb in either crystal cell maturation (i. e., going from iCC to mcCC) or mCC survival has to be clarified.

– Figure 5 sup2 : Quantification is missing. p 38 lines 979-981. The authors need to clarify whether there are talking about an increase in Numb levels per crystal cell, or an increase in Numb levels per LG, which in this case reflects an increase in the number of cell expressing Numb.

– Numb subcellular localisation is different from one picture to another. In Figure 5M, Figure 5 sup2 , Figure 6A-c and 6 H-L', Numb is mainly localised at the periphery of the cell at the plasma membrane, whereas in Figure 5 sup 3A-B and sup 3F-G, Numb is detected as cytoplasmic punctate dots without staining at the plasma membrane. This is confusing and clarification is necessary.

– Figure 5 sup 3: PPO2 staining shown in Figure 5 sup 3 K-L is not in agreement with the quantification given in M. p values are missing in M and N. There are also discrepancies between these figures and the text p 39: "numb RNAi expressed in crystal cells causes ...with a concomitant increase in the iCC population". Figure 5 sup 3N: Quantification of crystal cell numbers is not convincing. Since there is a lot of variation among LGs, quantification of PPO+ cells has to be given as an index (i.e., a ratio between the total number of PPO+ cells/ per total number of LG cells). In N, 6 LGs maximum per genotype have been analysed, which is far too few. Defining whether the number of crystal cells is affected in lz>numbRNAi is essential to determine whether Numb is required to allow iCC to mature into mCC, or whether it controls mCC disruption. The MM section has to be completed; it should indicate how quantifications of cell numbers and fluorescent intensity were performed.

– For Hnt staining in lz>numbRNAi, there is a discrepancy between Figure 5 sup 3 l-l" and Figure 5 sup 3 L-L". In I-I" no difference in Hnt levels compared to control (H-H) is observed, whereas in L-L" a strong decrease is observed (K-K").

– The enlargements shown in Figure 5O-R have been taken from pictures shown in Figure 5 sup 3K-L. It would be better to show independent immunostainings. This remark is even more relevant in this case because the staining in Figure 5 sup 3 L is not convincing.

– Figure 5 sup 3 H-I: lz >numb RNAi there is a decrease in Sima staining. Is it due to a decrease in sima transcription and/or Sima protein stability and/or Sima subcellular localisation?

– Figure 6 J and M ; is "the Numb intensity" referring to the intensity of Numb per cell or the total amount of Numb intensity measured per LG? This has to be clarified in the figure, in the text (p 40) and mentioned in the MM. What about the crystal cell index in lz>msi RNAi and Hml>msRNAi?

– A schematic representation of crystal cell maturation involving N (both the canonical and non-canonical signalling), Numb and Sima would be very helpful.

– In MM p 74 lines 1888 "data was corrected for batch effects between samples". The information concerning the method used has to be provided.

9. Modifications in the text are required

– Lines 1093 "the equilibrium signal from proPl". Functional data supporting this conclusion are lacking.

– Lines 1088 "JNK signalling ... is a specific property if IZ cells". Neither JNK expression nor its function has been analysed in this study.

Reviewer comments after revision

Decision letter 2

[Editors' note: further revisions were suggested prior to acceptance, as described below.]

Thank you for resubmitting your work entitled "Paths and Pathways that Generate Cell-Type Heterogeneity and Developmental Progression in Hematopoiesis" for further consideration by *eLife*. Your revised article has been reviewed by 3 peer reviewers, one of whom is a member of our Board of Reviewing Editors, and the evaluation has been overseen by Anna Akhmanova as the Senior Editor.

The manuscript has been improved but there are some remaining issues that need to be addressed, as outlined below:

Essential revisions:

1. The authors should temper some conclusions because the conversion of AUC scores to z-scores can obscure the actual level of enrichment. Specifically, the authors should:

a. Soften the conclusion that proPL generates the equilibrium signal and that the IZ alone activates the JNK pathway as this is based on a low number of genes (lines 959-960). Please rephrase this statement.

b. Point out in the text that JNK activation is enriched in but not specific to the IZ.

c. Clarify the in vivo definition of the IZ and the ProPL cells with respect to the Hml delta QF driver. The authors use it to specifically label the IZ but the driver's expression domain is broader than IZ cells.

2. The authors should indicate in Figure 3—figure supplement 3, 1D which AUC terms encompass "glycerolipid remodelling genes" (line 592)

3. The authors should rephrase "pnt transcript is expressed at low levels in the PSC (Figure 4A)". (Line 664). One interpretation of Figure 4A is that pnt is expressed at the same levels across the different clusters but in a smaller number of cells in the PSC (hence the smaller circle).

4. The authors should provide publications for the enhancers in Figure 5—figure supplement 1 B,C,G or explain how they defined the enhancers.

5. The authors should remove the comma in line 795 ("we find, is").

6. The authors should avoid the term de-enriched/de-enrichment (lines 525, 549, 562, 836, 135, 1136, 1159, 1435, 1437, 1438) to indicate a change in the levels of expression.

7. The authors should modify their conclusion (line 652) "establishing the IPs as MMP1 producing cells". The diffused MMP1 staining in Figure 3D-D' is not convincing.

8. The authors should address how they conclude (line 696) that "Pnt is required for exit from the IZ" since there is no defect in CC differentiation in CHIZ >pnt RNAi (Figure 4l-M). They should also comment on Pnt's role in plasmatocyte differentiation.

9. The authors should modify the introduction so that it includes a statement about the cardiac tube acting as a niche to control lymph gland hematopoiesis and supporting references.

10. The authors should add a sentence/short paragraph in the discussion saying that the proposed model/definition of states awaits functional confirmation, at least in some cases.

11. In the dot plots, the authors should indicate what is meant by 'mean' and 'non-zero percent'.

12. In Figure 2H,I, the authors should show the whole lobe and indicate the IZ and the proPL with arrowheads.

Suggested revisions:

1. It is suggested but not required that the authors provide a volcano plot to illustrate the differences between proPL and IZ.

2. It is suggested but not required that the authors provide the single table that must have been generated to produce the figures of the first submission and used to calculate the z-scores.

Detailed Reviews

*Reviewer #1/Reviewing Editor:*

I have read the response to authors and the revised manuscript. The authors have addressed all of the essential revisions and all of the reviewers' comments satisfactorily. They have overhauled the manuscript, making extensive changes to the figures and the text, thereby improving their study and its conclusions. The manuscript is now easy to read, follow the logic and the data support the conclusions in the text. I recommend its publication in *eLife*.

*Reviewer #2:*

The revised manuscript from Utpal Banerjee and collaborators involves substantial editing of the text, additional experiments and significant changes in the presentation of the data. The RNAseq data provide a useful resource to the community and the validations already help understanding the mechanisms controlling lymph gland development. Altogether, the revision allows a much smoother reading and answers many questions asked by the reviewers. I have few comments that do not call for additional experiments but need to be addressed.

The authors converted the AUC-scores in normalised z-scores to enhance the contrast on the heatmaps. This representation does indeed ease the interpretation of the AUC score but hides completely their actual levels of enrichment, which leads to strong conclusions based on minor differences. Here below indicative examples.

In the discussion (line 959-960), the authors state "proPL (but not IZ) generates the equilibrium signal, whereas the IZ alone activates the JNK pathway". While this statement could be deduced from the heatmaps (Figure 2—figure supplement 2A,B for the AUC Equilibrium signal and Figure3B and Figure 3—figure supplement 2B for the JNK pathway), it is far from being conclusive (compared to the initial representation of the 1st submission, old Figure 2—figure supplement 5 for the AUC Equilibrium signal and old Figure 3I,J for the JNK pathway).

In the previous representation, Mmp1 expression levels and AP-1 targets are enriched in IZ but overlap between the IZ and the proPL, suggesting that JNK activation is enriched but not specific to IZ.

Concerning the equilibrium signal, Figure 2—figure supplement 2B indicates a strong enrichment in the proPL but the IZ also present a mild enrichment. In addition, the AUC equilibrium signal displays low variability across the different clusters (old Figure 2—figure supplement 5: highest ~0.475 for pPL-4 and lowest 0.445 for PL-7). The AUC score represents the number of genes associated with the term. Thus, the score of AUC comprising low gene number should provide stronger contrast than the AUC with high gene number since the genes are less likely to be among the top 25% of genes (threshold set up by the authors). The AUC Equilibrium signal comprise 6 genes with a single one (Pvr) enriched in proPL. With such low gene number, the scores 0.475 and 0.445 both represent less than 3 genes and no striking differences across the two clusters. Therefore, it seems overstating to say that "proPL (but not IZ) generates the equilibrium signal".

To avoid any mis-interpretation and provide the reader with all the data to interpret the heatmap, I would recommend the authors to join the matrix of the AUC scores used in the manuscript across all clusters. I would also strongly recommend to rephrase the sentence line 959-960.

A clarification is appreciated on the in vivo definition of the IZ and the ProPL cells. The authors use the Hml delta QF line and CHIZ Gal4, a split Ga4 line relying on the domemeso and Hml delta enhancers that allows the identification of the IZ cells (page 16-18). The Hml delta Ga4 driver is expressed in the cortical zone and in the intermediate zone (Spratford 2020). The Hml delta QF is derived from this line and has an identical profile of expression (line 428). Yet, with no further explanation, this very line is used to label ProPL cells and distinguish them from the IZ, in combination with CHIZ-Gal4 (Figure 2H,I).

Altogether, the use of AUC and cell states adds a new dimension to the single cell analysis and can identify transient populations that appear during development, however, the manuscript largely over-emphasizes this concept and subdivides the developing lymph gland into numerous cell subpopulations in some cases without strong evidence. Beyond the fact that the analysis concerns a single time point, does not provide spatial resolution and does not sufficiently validate the described states (three issues that are anyway beyond the scope of the manuscript), an unweighted analysis can lead to over interpretations. After all, AUC are relative definitions, as detailed in the rebuttal letter. To distinguish cell groups through AUC, qualitative and quantitative information should to be somehow taken into account: the number of cells, the number of genes in each AUC, that of the members of the AUC showing differential expression, the absolute levels of gene expression, the levels of differential expression, as well as the relative 'significance' of genes in the AUC. Typically, transcription factors that control the expression of many genes or key enzymes in a biochemical pathway may have a different weight compared to other genes. Some of this information is not available in the figures, some is present in supplemental materials/can be extrapolated from other figures (whereas each figure should be self-explanatory). The lack of granularity necessary to firmly define a cell group (state, cluster...) by no mean affects the quality of the manuscript as the validation of all the cell groups in the lymph gland can await further studies. The authors should nevertheless be more cautious in the interpretations and avoid strong statements. This will overall strengthen and further simplify the message provided by the manuscript.

*Reviewer #3:*

The revised version of the manuscript answers my main requests. The extensive editorial changes in the text and modification of figures make the manuscript much easier to read and allow one to understand the novelty of this study.

I still have two concerns:

1. Figure 3D-D': That MMP1 staining corresponds to diffused MMP1 is not convincing. If IZ cells synthetized MMP1, why is there no MMP1 staining in these cells? Furthermore, MMP1 is expressed in plasmatocytes; thus in Figure 3E-F'(Chiz>mihep and CHIZ>hepACT) what about plamatocyte differentiation in these two conditions? The modification in MMP1 expression in these 2 genetic contexts might reflect a change in plasmatocyte differentiation and thus might explain the difference in MMP1 expression. In conclusion, data supporting that MMP1 is produced by IZ cells are missing, and the sentence on line 652 "establish that IP as MMP1 producing cells" has to be modified.

2. Line 696 :"Pnt is required for exit from the IZ " How do the authors arrive at such a conclusion since CHIZ >pnt RNAi (Figure 4l-M) shows no defect in crystal cell differentiation? What about plasmatocyte differentiation?

Additional corrections:

1. The introduction should be updated, including that the cardiac tube acts as a niche to control lymph gland hematopoiesis. This has to be added.

2. Figure 2 Sup 3 : Zscore is -3 to +3. Is it OK?

Decision letter 3; after second revision:

Dear Dr Banerjee,

Congratulations, we are pleased to inform you that your article, "Paths and Pathways that Generate Cell-Type Heterogeneity and Developmental Progression in Hematopoiesis", has been accepted for publication in eLife.

Editor's evaluation:

This paper will be of interest to scientists who study hematopoiesis. The authors combine single cell RNA-seq with bulk RNA-seq of transcripts from blood cells in the *Drosophila* larval hematopoietic organ. They present extensive analysis of the datasets, and the pseudotime analyses present a model of how hematopoietic progenitors can differentiate along transitory paths. These datasets reveal cell-type specific isoform expression of Notch pathway regulators, and genetic experiments prove the importance of these factors in development of one lineage. These transcriptomic analyses and subsequent genetic experiences provide strong support for the major claims of the paper.

Please take note of the points below and we hope you will continue to support eLife.

Best wishes,

Erika Bach

Reviewing Editor

Anna Akhmanova

Senior Editor

---

## [Author Response]

Following first review:Response to essential revisions:1) Address issues with AUC scores:a) Define how you established the AUC list.

We have now clarified how we established the AUC lists in the Methods section (page 68). In brief, chosen gene lists are compiled in ways that are relevant to the individual situation. If a set of genes is anticipated to be co-regulated based on some independent criterion, then we ask if that set is also enriched in any cell of the lymph gland, and if multiple cells belong to this enrichment profile, do they form a cluster or a combination of related clusters?

Such a strategy also helps us, for example, in mapping a list of genes identified as subzone-specific by Cho et al., onto our distribution of clusters, allowing for comparisons between the two approaches.

For more predictive purposes, gene sets could comprise, for example, published data sets of coregulated genes, those curated under well defined GO terms, KEGG pathway lists, or known and time-tested components of metabolic pathways that we have curated as stable, non-redundant participants across species. The individual genes used to construct the AUC lists as well as the source data for those lists can be found in Supplementary File 4.

b) Clarify how you define AUC scores that are statistically significant.

p-values directly calculated from the AUCell plots would be inflated in their significance because the number of data points (i.e. the number of cells used in the scRNA-Seq) is very large (please see Lin et al. 2013: “Too Big to Fail: Large Samples and the p-value Problem” for details). Very large sample size (# of cells) causes very small changes to gain significance, artificially lowering p-values. The more appropriate statistical criterion for such analyses is the number of standard deviations that separates two means of the sets of data as a way to assess the size of the effect. We have now addressed this issue by calculating z-scores (a representation that includes standard deviations from the mean). For consistency, we now use a single, standardized spread of z-scores that ranges from -2 to +2 for all such statistical representations throughout the paper.

Corresponding p-values can be calculated from the z-scores and are now reported in the figure legends.

For GO terms or KEGG pathway components, we first conduct Gene Set Enrichment Analysis (GSEA) to determine, in an unbiased manner, which terms or pathways are significantly enriched in each cluster. The resulting False Discovery Rate (FDR) values are meaningful, and we have now included them for the specific GO or KEGG terms in a new supplementary figure (Figure 1 figure supplement 3A).

c) Include a zero (0) on the y-axis of all graphs and P values for all graphs.

With the above in mind, and following some excellent comments and suggestions from the reviewers, we have revamped the way we display and use AUC scores; we agree that box plots are not the most convenient representations of such scores. To address concerns about comparing “highly heterogeneous ranges'' of AUC values across clusters, we now use heatmaps with standardized z-scores ranging -2 to +2 as explained above, to represent all AUC values and their variations across one standard scale (See new Figures 3A, C; Figure 2—figure supplement 2AB; Figure 2—figure supplement 3H; Figure 3—figure supplement 1D; Figure 3—figure supplement 2B). Related AUC lists are presented together on a heatmap with the z-score representing the standard deviations from the mean. Such comparative representations have been successfully utilized in previous studies of mammalian hematopoiesis (Zhu et al. 2020; Blood 136 (7): 845–856). The AUC scores in this standardized heatmap format displays individual values and thus 0’s on the y-axis are not necessary for the comparisons. Please also see a more detailed explanation with examples below (Reviewer #2, comment 1).

2) Provide a comparison of the data between the replicates.

This was an excellent suggestion with a pleasing outcome. We now include a comparison of the overall predicted cluster size and structure between the three replicates (Figure 1—figure supplement 2AB). This includes the percentage of cells belonging to each cluster and we find that each cluster contributes similarly in structure and in percentage in individual biological replicate samples (Figure 1 figure supplement 2A-B). These are true biological replicates and not simply technical ones, with high viability and high recovery rate maintained across the isolates. This adds to the robustness and reproducibility of the methods. We thank the reviewers for this valuable suggestion.

3) Provide validation of some of the genes associated with clusters/sub-clusters, including Delta in MZ1, PSC, X and MZ2 clusters, and Ubx in posterior lymph gland lobes, and provide validation to support your model that there is no transition path for IZ to proPL.

First, we thank reviewers 1/editor and 2 for their opinion that validation of the expression pattern of all identified genes in the single cell data is out of the scope of this manuscript (Reviewer 2) and yet that several individual genes have been validated (Reviewer 1) through multiple means. We whole-heartedly agree with this. We wish to briefly elaborate on this point.

In the Discussion section (pages 33-35), we have explained that the methodology employed here is quite different from those in Cho et al., and so is the nature of what we consider is the best way to validate our data, given that doing so for the entire list is beyond the scope. For bulk RNA-Seq experiments, we first separate cells based on single or double markers for a cell type (*dome*^meso^ and *Hml^Δ^* single positives, double positives, and double negatives, etc). And we do this with a different genotype to separate CCs. Thus, we know which population belongs to a particular zone. Since we start with separated MZ, CZ, IZ, CC cells based on how they have been historically defined, there can be no question that what we sequence in the separated “MZ” cells are indeed MZ. So when two clusters in the scRNA-Seq show those characteristics and yet are separable, we can be confident in calling them MZ1 and MZ2. Additionally, we use lists of published hallmark genes (that have been validated through genetic analysis and/or immunohistochemistry through the last two decades of lymph gland studies) to compare to the lists of differentially expressed genes found for each cluster. Our algorithm for generating data (unsupervised graph-based clustering) is agnostic to these lists, but when mapped, they belong to the appropriate zone and cell type (such as PSC, Crystal Cells, MZ, and Plasmatocytes). This first step of isolating populations of cells by FACS, defining markers through bulk RNA-Seq, and mapping them onto scRNA-Seq is not often used in other transcriptomic analyses, but it provides a benchmark and validation for the cluster structure. The bulk RNA-Seq and scRNA-Seq are orthogonal methods conducted on different biological samples on independent days using very different techniques. Thus their remarkable agreement is also a measure of the robustness of the strategy (examples of comparisons shown in Figure 1K, Figure 5K-L, Figure 6D-E, Figure 2 figure supplement 1A, K, Figure 2—figure supplement 3A). We hope we made the distinction clear between isolating a physically and genetically defined zone of cells and seeing which of the clusters identified within it differentially express known markers and the approach in which all cells of the lymph gland are mixed together even from different time points in development, and calling a set of cells a subcluster of prohemocytes (PH1, 2, 3, etc as in Cho et al.), because they express some markers. Both are valid approaches, but the latter approach cannot guarantee a single physical localization of the subcluster within a zone (e.g. MZ, CZ, etc).

Most importantly, we include multiple individual examples of cross-checks of revealed expression patterns with loss of function genetic assays and we also include two complete case studies (which, as a reviewer says, could have made their own manuscripts); we included them here precisely because they provide functional validation for the expression patterns, and in the process provide elegant genetic solutions to problems that have eluded us in the past. The fact that an alternate isoform of a protein (Sima in case study 2), or a previously unexplored step-wise role of a pathway involving a very well studied protein (Pointed in case study 1) could be linked to novel functional cascades by following up their transcriptomic profiles, to us, is the best validation of the data, and it then successfully loops back to predict the expression of additional members of the pathway (eg., Numb and Musashi). Since we are limited to choosing a small and representative number of genes to validate, we prefer ones that also help us understand principles of blood development better, over choosing a handful of randomly selected genes to validate, for which *Gal4* drivers are available. The fact that entirely new genetic pathways can be constructed relying on the RNA-Seq data predicted for their components, in our mind, is a very powerful validation of the system and is the sort of validation that only *Drosophila* can currently provide.

Finally, the ultimate goal of this manuscript is not to identify more “hallmark genes” and new cell-type specific markers, although such markers fall out of some of the analysis. Generally speaking, developmental genes are rarely cell-specific (genes such as *Lz*, *NimC1*, and *Antp* are exceptions since they represent proteins expressed at terminal stages of development). We focus more on the trends of coexpression of pathway-specific genes that are revealed by RNA-Seq, and therefore validating such pathways by loss of function is our primary approach and we believe if counted with the above criteria in mind, and combined with protein expression patterns, we have validated a fairly significant number of the identified genes in this manuscript.

With regards to *Delta* expression, *Delta* was identified as a PH1 marker by Cho et al. We also detect *Delta* in our analysis, but did not include it because it is expressed at very low levels. We now do so in the heatmap shown (Figure 1K). In terms of differential expression, PSC cells show the highest expression of *Delta*, which is lower in MZ1 and becomes insignificant in MZ2. These results are comparable with Cho et al. who do not comment much on the PSC, but, when GTraced for *Delta-Gal4* expression, their data also show the highest expression in PSC. This is consistent with our data and also with the overall expression level of *Delta* being low. We have also included a comparison between the two studies in the discussion (page 35-36) but also retained the more detailed comparison (Supplementary file 2).

The primary MZ1 cluster in our study is small (1.3% of total cells), but likely includes the much smaller subclusters defined by Cho et al. 2020: PH1 (0.4% of total cells), PH2 (0.4% of total cells), and a subset of the PH3 cells (2.1% of total cells). In this scenario, PH1 would comprise such a small proportion of MZ1 that it is not a surprise that *Delta* is not identified as a differentially represented hallmark of MZ1. This illustration also justifies why we did not further subcluster the primary MZ1 cluster. Doing so, we end up with subclusters that are not very robust.

Concerning the comment, “and *Ubx* in posterior lymph gland lobes...”. In the primary lobe, *Ubx* is identified as a low expression marker with its highest representation in MZ1 as in Cho et al. 2020, who found it to be a marker of PH2. *Ubx* expression in MZ1 is surprising, worthy of a future genetic analysis. But for now, all we can say is that the expression data seem consistent between the two studies.

The reference to posterior lobes in the context of this manuscript is unclear. All posterior lobes are removed during dissection of the tissue and so this analysis is strictly for the anterior lobe. The reason for the expression of Ubx in the posterior lobe region was explored by our laboratory almost two decades ago (Mandal et al. 2004). The excellent Rodrigues et al. 2021 paper that Reviewer 3 refers to cites our past work and makes the following statement that we have known for a long time as well: “Ubx is specifically expressed in the tertiary lobes of third instar larval lymph glands.” Posterior, yes, but not even in the secondary, instead the tertiary lobe expresses Ubx. So there is no chance of any contamination during dissection given the intervening secondary lobes and pericardial cells. This is not to say that some unknown signaling process or tertiary lobe cells might not have contributed to turning on *Ubx* in early larval stages, but that would be purely speculative at this point.

Regarding “No transition path from IZ to proPL”, we apologize if we overstated our case in implying that even in principle, there can be no cross-talk between IZ and proPL. This is more properly addressed in the current revision (pages 16-18 in results; pages 37-38 in Discussion). However, it is not true, as Reviewer 3 suggests, that an overlap in trajectory/pseudotime implies a lack of independence between the two paths. To make this point clearer, we now include a new figure (Figure 2G), illustrating what overlap in pseudotime means (a different but equivalent representation can be seen in Figure 4C; Supplementary Figure 3C of the Cho et al. 2020 manuscript). We have also added new in vivo data that clearly demonstrate the distinctinction between IZ and proPL cells (Figure 2H-L, Figure 2—figure supplement 2C). Inasmuch as pseudotime represents temporal progress in development and not differences in cell types, two independent pathways can indeed overlap in pseudotime. Both IZ and proPL are found in transitional states that link MZ2 with PL, but we know that IZ and proPL are distinct clusters (Figure 1J; Figure 2F) with distinct gene expression patterns (Figure 1K; Figure 3B), and have distinct functions (Figure 2J-L; Figure 3D-F’’; Figure 2—figure supplement 2C). On a 3D tSNE (Figure 2F; Video 1), these two large clusters are not adjacent along their boundaries as they are with both MZ2 and PL. We also know that these two paths are different since IZ and proPL can be visualized with distinct genetic drivers and have differences in signaling properties in vivo (pages 16-18, Figure 2H-L; Figure 3D-F’’). Therefore IZ and proPL provide two different paths. However, since absence of evidence is not evidence of absence, we should not have implied that there is no possibility of exchange between them. We thank the reviewer for pointing this out and have made the appropriate changes listed above.

4) Demonstrate the potency of RNAi lines, including pnt RNAi.

We have now included data that demonstrate the potency of the RNAi lines in the Methods section and in individual figures and figure legends. Not all lines could be retested by RT-PCR due to our lab closures, but each line is vetted for the robustness and specificity of its activity. For example, and in the context of the reviewer’s specific query, the *pnt^RNAi^* line shows a very strong and consistent phenotype as now documented with quantitation and p-values (Figure 4B-D). Additionally, we used 2 different and independently generated pre-validated RNAi lines for *pnt* that both give rise to the same reproducible phenotype when each is used with 2 independent drivers (Figure 4P; Figure 4—figure supplement 1F). Finally, these *pnt^RNAi^* lines have been validated as well, in several projects in the past since they are used routinely in the laboratory for diverse experiments in different tissues.

5) Explain the main differences and similarities between your data set and Jiwon Shim's by addressing specific issues such as Cluster "PH1" (your MZ1, PSC, X and MZ2 clusters) and Cluster X (and include a dot plot representation of Cluster X).

A discussion of the main differences and similarities between our data set and that by Jiwon Shim’s lab (Cho et al. 2020) is now included in the main text (pages 9-12; 19; 35-36).

Briefly, from this comparative analysis we conclude that:

– The broadly defined clusters identified by Cho et al. agree fairly well with the primary clusters we identified in our experiments as seen in Supplementary File 2A-B.

– Due to inherent differences and biases of the different experimental methods employed by the two studies (e.g., DropSeq vs 10X Genomics) (page 35-36), we do not expect an exact 1:1 match of our clusters with the smaller subclusters from Cho et al. nor a complete overlap of their exact genetic profiles.

– Based on transcriptomic and genetic data as well as comparison of sizes, MZ1 cluster is most similar to the PH1 and PH2 subclusters of Cho et al. (pages 9-12, 35-36). The MZ1 cluster (1.3% of total cells) likely contains cells of the subclusters PH1 (0.4%), PH2 (0.4%), and also a small number of the PH3 cells (2.1%) (Supplementary File 2E).

– MZ2 cluster (26% of total cells) is most similar in gene expression and size to PH4 (27.2%) (Supplementary File 2C-E).

– It is important to keep in mind that PH1/2/3 etc are obtained by a “subclustering” algorithm (after the authors “aggregated cell clusters according to the expression of previously published marker genes by manual curation”), which is very different from the clusters such as MZ1, MZ2, etc obtained through unsupervised graph-based clustering methods. So we are not surprised by several subclusters mapping to our clusters.

The primary X cluster (1.1% of total cells) is most similar in gene expression and size to the primary “GST-rich” cluster (1.2% of total cells) in Cho et al. (Supplementary File 2A-B, E). The X and GST-rich clusters share many genes in common, and we find that the genes shared between Cluster X and the GST-rich cluster show strong enrichment of DNA damage response pathways (p-value = 0.00004). X and PH1 also show similarities but that can be largely attributed to common cell-cycle genes (p-value = 5.031×10^-19^), and Cho et al., also mention high mitotic activity for PH1. In terms of a broader comparison however, the number of shared genes is highest between X and GST-rich.

A dot plot representation of genes highly enriched in Cluster X has now been included (Figure 2 figure supplement 1J). We thank the reviewer, it does look much nicer than what we had before.

6) Provide additional quantification of CC in "Numb and Musashi in CC determination", additional information on how Numb intensity was quantified, and P values

Thanks for this important suggestion. We now provide additional quantification and p-values for the various sorts of data concerning CCs in the “Numb and Musashi in CC determination” section (Figure 5K, O; Figure 5—figure supplement 2D; Figure 5—figure supplement 3H-I; and Figure 6D, J, M). The intensity values of Numb (Figure 5—figure supplement 1M; Figure 5—figure supplement 2D; Figures 6J, M) represent “per cell” expression and not of the entire gland. However closely placed cells were quantified together to prevent skewing of the data. In general, we have now included p-values for other data outside of this CC section as well, and can be found in all relevant figures in the revised manuscript (new Figures 1, 4, 5, 6, and related supplementary figures).

7) Provide for lymph gland analyses the quantification of CC and MZ indexes and P values.

We have now provided quantification of CC and MZ indexes and p-values for lymph gland analyses (Figure 4P; Figure 5O; Figure 4—figure supplement 1A, C, E-F; Figure 5—figure supplement 2D, G; Figure 5—figure supplement 3H-I, Figure 6J, M).

8) Consider using dot plots instead of box plots.

We thank Reviewer 2 for this suggestion. We have included several dot plots (Figure 2N; Figure 4A; Figure 5L; Figure 6E; Figure 2—figure supplement 1J-K; Figure 2—figure supplement 2G). Box plots have been entirely eliminated throughout the study, including for AUC scores and replaced with heat-maps and dot plots as appropriate.

9) Provide in the main figures a list of the key genes defining each sub-population inside the different clusters.

We now provide a list of the key gene combinations that define the subpopulations (Figure 7A). As mentioned in Essential revisions #3, please note that trajectory states represent progress in pseudotime and not different cell types. Also important is our basic premise that developmental genes are rarely cell type specific. Therefore, clusters are defined by expression trends. While the entire transcriptome contributes to defining this trend, we have included (Figure 7A) some of the combinations of genes that we have investigated and validated as part of this study.

10) Provide the number of cells in each cluster, including the IZ cluster.

We have included the number of cells in each cluster including the IZ cluster (Figure 1J) and also included this number for each replicate (Figure 1—figure supplement 2B).

11) Provide a schematic representation of crystal cell specification and maturation involving Notch, Numb and Sima.

Thanks, this is an excellent suggestion. We have done so in a new Figure panel and this does improve the readability of the manuscript (Figure 7D).

12) Make extensive editorial changes, including shortening the manuscript per the reviewers' suggestions and clarifying the narrative so that the introduction covers what is actually being treated in the manuscript. For example, the introduction discusses the IZ but then the manuscript does not provide much insight into this cluster and instead focuses on CCs.

We thank the reviewers and RE for this critique. The introduction is shortened and re-focused on the main themes of the manuscript. In general, we agree that the descriptive parts were confusing and repetitive. In the annotated (changes tracked) version of the revised manuscript, we hope it is clear that we have made extensive changes to every section, shortening where appropriate, and avoiding duplications throughout the manuscript. There are too many such changes to list them here individually.

The changes do not affect the scientific conclusions but, we hope, make them more comprehensible.

Thank you for this suggestion on which all reviewers agreed and we appreciate this constructive feedback a lot. Overall the manuscript is shorter by 7.5 pages. The net decrease in figure panels is 53. This is in spite of the many new additions asked for by the reviewers.

As for the comment on IZ, this manuscript includes more details on IZ than any other published work to date. To supplement the material further, we now include more functional data on the IZ (Figure 2H-L; Figure 2—figure supplement 2C; Figure 4—figure supplement 1F). CC development is a more mature field that introduced our laboratory into hematopoiesis and naturally it is more amenable to complex genetic analyses (Rizki and Ritzki 1981, Genetics 97(Suppl. 1): s90; Lebestky et al. 2000, Science 288, 146–149).

Response to detailed comments (essential revisions and other reviewer suggestions)
*Reviewer #1/Reviewing editor:*
There is a massive amount of data in this study. The data are of high quality, and the manuscript is well written. The results support the main conclusions. This study will be a very valuable resource to the community. I have a few suggestions for possible improvement.1. On p. 36 (line 906), the authors write that "we surmise that Notch activation is the default pathway for early Hml-expressing cells to become CCs, and that the activation of Pnt acts antagonistically to prevent this process thus favoring instead, the plasmatocyte fate." I don't understand the logic of this. PLs represent 95% of the mature hemocytes, whereas CC represent 5%. Why would most of the differentiating hemocytes have to repress Notch signaling by expressing Pnt. Loss of Pnt from Hml+ cells would very consequential as the animal would not have PLs from the LG. What I am trying to say is that the kind of regulation proposed here is not robust and could be easily disrupted by mutation. Could the authors comment on this.

Thank you. Calling this a “default” pathway was an oversimplification of the data presented and we have now rephrased our statements in the text (page 27-28). We hope this resolves the main issue.

But the reviewer asks a very insightful question about CC number. The answer is long and not a central focus of this manuscript. The basis is in the dynamic regulation of the ligand (as it is for all cases of Notch activation in flies), Serrate expression is dynamic, temporally and spatially, and controls how many cells receive the Notch signal. This regulation depends on multiple factors, including, for example, systemic signals triggered by the O_2_/CO_2_ balance of the local environment.

2. The manuscript is very long (the results section is thirty-five long) and reader attention spans tend to be short. Could the authors please edit the manuscript to reduce its length. For example, I don't think that the entire section about cluster X is needed. The metabolism section could be condensed. Some of the discussion is redundant with the results.

Thank you for the valuable suggestion, we have addressed this as above in the essential revisions (12). The introduction and discussion are shortened and repetitions removed. X and metabolism are telling us something novel about development that few laboratories study, but that future investigators might find valuable and so while we shorten them as you ask, we retain their prominent mention in the paper. The editing, asked for by all reviewers, was a very valuable suggestion.

3. Please provide accession numbers the raw data at NCBI and a link to the reviewers.

To review GEO accession GSE168823 (now included in Methods page 68).

Reviewer #2:In this manuscript, Girard et al. analysed the transcriptome of the lymph gland of *Drosophila* using high throughput sequencing. They fist used genetic markers (Domemeso-GFP and Hml-RFP) to sort the cells from three distinct regions in feeding larvae: the medullary zone (MZ) known to be populated by progenitors, the intermediate zone (IZ) populated by intermediate precursors and the cortical zone (CZ) containing the mature hemocytes. The cells from the CZ were further subdivided into plasmatocytes, immature crystal cells and mature crystal cells with the genetic markers Hml-RFP and Lz-GFP. This subdivision was carried out at later stage to maximise the number of mature hemocytes. A comprehensive molecular signature of each region and cell type was determined with bulk RNAseq. Then, the authors analysed the transcriptome of 21200 cells of the lymph gland using 10x genomics technology. They found 9 clusters of cells displaying distinctive signatures and metabolic properties, including the cells of the Posterior Signaling center (PSC), two types of progenitors in the MZ, the intermediate precursors, the plasmatocytes as well as the immature and mature crystal cells. They predicted the filiation between the clusters using Monocle, indicating a developmental trajectory starting in MZ producing IZ and then plasmatocytes. At last, the authors validated two mechanisms of cell differentiation highlighted by the transcriptome analysis. They showed the context-specific role of the transcription factor Pnt in the differentiation of plasmatocytes. Loss of Pnt in the early progenitors has no effect whereas its loss in the late progenitors prevents the differentiation of intermediate precursors and thus mature plasmatocytes without affecting the differentiation of crystal cells. As a second study case, they showed that the interplay between Notch, Numb, Sima and Musashi is involved in the maturation of crystal cells.Overall, this manuscript presents a tremendous amount of RNAseq and in vivo data with highly detailed interpretations. It provides very valuable and substantiated information on the molecular mechanisms involved in the development of the hemocyte lineages in the *Drosophila lymph* gland. The main caveats are (1) the definition of two clusters IZ and proPL, which seem to belong both to the IZ, (2) the fact that most boxplots do not include 0 in the y-axis, which strongly biases the interpretation of the graphs, (3) the definition of mitotic precursors. Finally, shortening would have made the manuscript more easily readable and would have conveyed the message more directly.1) Most charts presenting expression levels or AUC score across clusters do not include 0 on the y-axis and disclose highly heterogeneous ranges. For example, the ranges of the AUC displayed in Figure 3 are from 37 points in A (y-axis range 0.43 to 0.8) to 3 points in I (y-axis range 0.13 to 0.16). This is highly misleading, and biases the interpretation of the data since the authors describe minor differences the same way than large differences. Few examples are:– In the legend of Figure 3B-C, the authors write "(B) TCA cycle enzymes are expressed at exceptionally low levels in the PSC compared with the cells of other clusters. (C) Expression of oxidative phosphorylation pathway enzymes is low in the PSC, high in MZ1 and MZ2 and moderate in IZ, proPL, and PL clusters.". In B, exceptionally low level correspond to 0.42 compared to 0.46 in the other clusters. In C, low in the PSC means 0.74 and high in the MZ corresponds to 0.78. Since AUC represent frequencies, I doubt that few point can translate into such high ranges.

Many of the concerns raised by the reviewers about AUC and how to measure and interpret its statistical and biological relevance are addressed above in essential revision (1). There we provide an explanation of the importance of z-scores that are easier to see in heatmaps, and so all AUCell analyses are now converted to heat maps with a single scale.

The reviewers’ concerns with the small differences in AUC scores are genuine and we thank them for their very careful analysis of our data. As stated in essential revisions (1), we no longer have AUC data shown as box plots, but they are now heatmaps with standardized z-scores.

But we do not wish to give the impression that the data in the box plots were incorrect and somehow disappeared in a new representation. For a full explanation of AUC data, we strongly recommend (2). Nature Methods, 14, 1083-1086; who first used AUC for transcriptomic analysis, but also the important details in https://www.bioconductor.org/packages/release/bioc/vignettes/AUCell/inst/doc/AUCell.html. Both of these resources helped us a lot. A quote from the bioconductor article is:

“The AUC is not an absolute value, but it depends on the cell type (i.e. cell size, amount of transcripts), the specific dataset (i.e. sensitivity of the measures) and the gene-set. […] Since the AUC represents the proportion of expressed genes in the signature, we can use the relative AUCs across the cells to explore the population of cells that are present in the dataset according to the expression of the gene-set.”

The above quote asserts that many factors affect AUC scores and also that its value is derived on a cell by cell basis. Most importantly, in our case, the actual AUC score will be different depending on the input data-set. As a hypothetical example, a data set-A, “signal transduction” vs set B called “EGFR signaling” (components of which are included in the “signal transduction” category as well), will give very different AUC scores even for a single cell involved in EGFR signaling. However, each set can be compared between cells (but the same cell cannot be compared between sets A and B). Thus, what matters is not AUC scores themselves, but how many standard deviations apart one group of cells is from another, when compared over all cells.

We now see and agree that box plots are a messy and confusing way to show this comparison. They are particularly troublesome when used to display comparisons of sets of data represented in arbitrary units (for all the reasons the reviewers point out). The z-score includes both difference from the mean and the standard deviation in its calculations and therefore small AUC scores with very tight distributions can give meaningful z-score values, while large mean values with very big dispersions might not.

For some of the specific cases cited by Reviewer 2, we have converted the z-scores into adjusted p-values (based on percentile deviations of the z-score distributions) and below are some numbers for zscores and p-values for each cluster for the metabolic processes referred to by Reviewer 2, read each as *Cluster (z/p)*:

For Glycolytic genes: PSC (+5.7/ < 0.00001); MZ1 (+2.5/0.006); MZ2/IZ/proPL/ PL z ranges from +0.98 to -0.79 and p-values from 0.2-0.5. CONCLUSION: PSC and MZ1 are significantly enriched (HIGH) for glycolytic genes with PSC significantly higher than MZ1. No significant enrichment of the glycolytic gene set is seen in MZ2/IZ/proPL/PL

For TCA cycle genes: PSC (-4.2/ 0.000013); MZ1/MZ2/IZ/proPL/ PL, -score ranges from -0.46 to +0.17 and p-values from 0.3-0.4. CONCLUSION: PSC is significantly de-enriched (LOW) for TCA cycle components; No significant enrichment or de-enrichment of the TCA gene-set in MZ1/MZ2/IZ/proPL/PL.

For nuclearly encoded Ox-Phos genes: PSC (-2.2/ 0.015); MZ1 (+1.95/0.026); MZ2/IZ/proPL/ PL, zscore ranges from +1.09 to -0.47 and p-values from 0.1-0.3. Conclusion: Ox-Phos related genes are de-enriched (LOW) and are somewhat higher in MZ1 (meets 5% but short of 1% criterion for significance). There is no significant enrichment or de-enrichment of the Ox-Phos gene-set in MZ2/IZ/proPL/PL.

We now use formal statistical descriptors (such as on pages 21-22 for the above pathways).

– In figure 3D and E, the authors mention the levels of Zw and Idh in the clusters. They explain that both are highly enriched in PSC compared to MZ. Zw presents an expression level of 1.1 in PSC, around 1.2 fold higher than in MZ and Idh presents an expression level of ~ 7 in PSC around 2 fold higher than in MZ. The difference in level of expression and fold enrichment cannot be appreciated properly due to the heterogeneous y-axes. In addition, the authors described the two enzymes in the same terms while Idh is expressed 7 time higher than Zw.

We state in the paper that Idh is the most abundant dehydrogenase in the cell that maintains NADPH levels important for detoxification of free radicals, Zw (G6PDH) is another dehydrogenase that also increases NADPH but is expressed at a low level. G6PDH is a PPP pathway enzyme that is enriched in the PSC cells, which must maintain very low ROS as explained in the text.

Even at low expression, the enzyme G6PDH is critical for ROS removal because it generates reduced glutathione (GSH) in a process that consumes NADPH. Normally this is not an issue because other sources, principally Idh can make NADPH (but it makes alpha-ketoglutarate in the process, and not glutathione) that can be utilized in the G6PDH coupled reaction to make GSH. But Idh activity requires isocitrate, made by the TCA cycle and if TCA is low (as in the PSC), G6PDH gets upregulated by a small degree to raise NADPH (400 million kids get anemic from low G6PDH since the RBCs lack citrate). All this to say that we lumped Idh and Zw together not because of expression levels but because of their coupled functions. We feel this will be too much detail to include into this manuscript. We hope you agree.

– For Pnt (Figure 4AB), the authors describe "a very low level of Pnt" in the PSC and MZ and a significant increase in the MZ2-3. The level in PSC is around 5.2, which is much higher than most genes described in this study and the significant increase is going from 5.2 in MZ2-2 to 5.5 in the MZ2-3. The significance of this increase need to be documented with a p-value and the biological relevance of this difference seem far-fetched.

The pnt levels are now expressed as a dot plot as per the reviewer’s previous suggestion (Figure 4A). As explained in the text (page 28), the change in *pnt* expression (as, likely with many developmental genes) is gradual and needs to be considered for its trend and not stepwise changes as for hallmark genes and therefore requires functional (genetic) validation in order for the trend to mean something. The dot plot shows a trend rising from its low expression in PSC to higher in MZ1/MZ2-1/2. Its rise in MZ2-3 is apparent with further increase in IZ/proPL, and in PL and then a very clear de-enhancement in CC. This gradual modulation of expression becomes meaningful only when combined with genetics, and is the reason for the “Case Study”. This is described in the text (pages 26-28) and figures (Figure 4A-P; Figure 4—figure supplement 1A-F), which establish independent functions of *pnt* in late MZ, IZ, and PL.

2) A better explanation should be provided for the AUC scores. The authors rightly say that AUC are reflective of co-regulation however the keys to interpret the score are not described. In addition, while a difference from 0.80 to 0.60 (Figure 3A) seems plausible and sufficient to call for an enrichment of glycolytic genes in the PSC, the biological relevance of differences from 0.74 to 0.78 (Figure 3C) or from 0.08 to 0.09 (Figure 2—figure supplement 5) seems far-fetched without further explanations/justifications.

Please see our response to comment 1 above.

3) The distinction between the cluster IZ and proPL needs to be clarified. The enrichments of the IZ AUC and individual IZ markers are not striking (Figure 2—figure supplement 3A-H). In addition, can the authors explain why only 6 of the 9 IZ specific genes were taken to estimate the AUC score? The strong similarities between the two clusters, in terms of markers and developmental trajectories, seem to indicate that the two clusters represent transient conditions that each cell goes through on its way towards full differentiation. Indeed, the single cell analysis has been performed in feeding larvae, when cells are actively differentiating, not in a steady state condition. Longitudinal/spatial analyses in the developing lymph gland might help in the interpretation, but this is not the scope of the present manuscript.

There are several parts to this question.

– The distinction between IZ and proPL has been better clarified in the main text (pages 16-18, 3738) and in the essential revision (3).

– AUC scores are no longer used in describing IZ genes. We present them individually in the revised version (Figure 2—figure supplement 1K). 6 of the genes are picked since the others are expressed at very low levels in the scRNA-Seq (page 15).

– Importantly, 5 out of those 6 genes (all but *CG31821*) are present in the differentially expressed genes list for the IZ cluster (Supplementary File 1).

– The expression patterns of the individual IZ-enriched genes are now shown in a dot plot representation (Figure 2—figure supplement 1K).

– A temporal mapping of gene activity in true-time, as the reviewer points out, is beyond the scope of this manuscript, but is a critical follow-up project.

4) The authors assess the impact of Pnt in the MZ cells using the drivers domemeso-Gal4 and Tep4-Gal4. The authors showed that domemeso>pntRNAi prevent the progression of MZ cells toward IZ cells. Some quantification would be welcome to appreciate the penetrance of the phenotype. In addition, the authors say that Tep4>pntRNAi has no observable phenotype, while the comparison between Figure 4E and Figure 4F seem to indicate that the lobe is smaller and with less Tep4 positive cells. Such difference could arise from slight stage differences. Could the authors indicate how they staged the animals, the number of samples...? At last, the potency of the pntRNAi construct should be documented.

We have now repeated our experiments with carefully staged larvae and included more definitive quantification for all of the *pnt^RNAi^* related data (Figures 4D, I, P; Figure 4—figure supplement 1A-F). Our analysis of all the images (n=10) confirms that *Tep4*>*pnt*^RNAi^ does not change the number of *Hml^Δ^*^+^ cells (Figure 4G-I), the number of *Tep4*^+^ cells (Figure 4—figure supplement 1C), or the overall lobe size (Figure 4—figure supplement 1D) to any significant level. We have replaced the previous images with those that better represent the statistical data. The potency of the *pnt*^RNAi^ line is discussed in essential revision (4) above, and is also clear from the quantification and statistical significance (Figure 4D; p<0.0001) of the observed phenotype. We also use two independent *pnt^RNAi^* lines and find similar results with both (Figure 4P; Figure 4—figure supplement 1F).

5) In the section "Numb and Musashi in CC determination", the authors mention that Notch-ACT raises Numb levels and Notch-RNAi decreases Numb levels in the crystal cell. This interpretation should be clarified. Since N activity is modulated using a CC specific driver and the number of lz>GFP also changes upon N modulation, the observed results may arise from regulation of Numb expression and/or from regulation of CC number. CC quantification will help sustaining their statement on the role of N. In the third paragraph linked to Figure 5—figure supplement 3H-N and Figure 5O-R', the authors describe a clear reduction of Sima puncta, PPO2 expression and number of mCC. P-values should document these observations. Also, does the expression of the type II targets decrease upon Numb RNAi? At last, Figure 6H-M, the authors indicate that msi-RNAi enhance the level of Numb in crystal cells without providing information on the procedure followed to measure Numb intensity. What does Numb intensity represent in Figures 6J and 6M? Is it the average level per cell or the level in the whole lobe?

We have quantified Numb levels in *Notch^RNAi^* and *Notch^ACT^* backgrounds (Figure 5—figure supplement 2D) to support our statements on the role of Notch. We have added p-values to document the reduction of Sima puncta (Figure 5O) and reduction in PPO2 expression (Figure 5—figure supplement 3H). The trend in crystal cell numbers in *numb*^RNAi^ vs wild type is best appreciated in the FACS results (Figure 5T), which show that the total CC number remains constant but the distribution shows less mCC and higher proportion of iCC.

We do not know if the expression of type II targets decreases upon *Numb^RNAi^* and these experiments would require extensive FACS sorting with many different populations and qPCRs that are not possible in the current circumstances. The intensity values in Figures 6J and 6M represent “per cell” expression and not of the entire gland. However closely placed cells were quantified together to prevent skewing of the data.

6) The authors carried out the single cell sequencing in triplicate. It would strengthen considerably the data to provide a comparison between the replicates.

We are very thankful for this suggestion. We have addressed this point above in essential revision (2).

7) The paper would gain from shortening. The introduction is broad and exhaustive, the results section describes the different the clusters and states as well as two study cases, the discussion elaborates on the mode of differentiation and put forward interesting models such as the gradual rather than stepwise transitions between groups of cells. Since this is a resource paper, the validation of all the single cell data is out of the scope, hence a thorough discussion of all those data could be shorten and used in subsequent studies. Furthermore, many data are already discussed in the results section, diluting the important and novel messages that the paper conveys.

We have addressed this point above in essential revision (12).

8) According to the authors, Cluster X represents mitotic states of several distinct cell types, including the CZ that carries differentiated cells. This intriguing finding indicating the presence of dividing cells throughout the lymph gland deserves some clarifications. Does it imply that none of the other clusters identified by the RNAseq analysis contains cells in mitosis? Does it mean that plasmatocytes and cells of the medullary zones have similar mitotic potential? Is there any difference in the type/levels of genes associated to cell division between the cells of the cluster that express MZ markers vs. those that express the Pl markers? I understand that the spatial analysis of the cluster X cells in the lymph gland, which would help clarifying these issues, goes beyond the scope of the manuscript. It would be nevertheless useful to compare the RNAseq data with those from the laboratory of Jiwon Shim, who also identified the clusters of mitotic cells in the lymph gland. Also, a dot plot representation of the genes associated with cell division in the different cells of the cluster X (MZ, Tr, Pl) might help identifying features specific to the different subclusters.

Although Cluster X shows especially high levels of mitotic activity, this does not imply that the cells in the other clusters are devoid of any proliferation. When purely mitosis-related genes are shown on the t-SNE some regions that border between clusters e.g. proPL-7 cells bordering PL and IZ have z-scores for G2-M related genes that range between 1.5 and 4.3 (p-values between 0.07 and <0.00001) (Figure 2 figure supplement 1D). Where these cells are spatially located in the lymph gland will be important for follow up studies. However the stress-related DNA damage activity is exclusive to Cluster X with z-score of 8.6 (p < 0.00001) (Figure 2—figure supplement 1E), for the mitotic DNA damage category. All other clusters have z-scores for this latter category that averages to approximately 0 (p = 0.5) and so it is possible that the nature of mitotic stress related activity seen in Cluster X is somehow different from that of other subsets of mitotic cells.

With regards to mitotic potential, MZ cells show very low mitotic activity while PL cells show higher mitotic activity at this developmental time point (as judged by phospho-Histone H3 staining in previous studies and mitotic G2/M transition enrichment here). As mentioned in the main text, the majority of MZ cells are found in the G2 phase. We believe it is this lengthening of G2 that may trigger the signs of replication stress in Cluster X but as the reviewer mentions a spatial analysis of the physical location of cluster X cells within the lymph gland will be necessary and is beyond the scope of this manuscript.

Finally, as described in essential revisions (5), signatures of Cluster X are detected in the Cho et al., “GST-rich cluster”. Through data mining, we have also detected clusters with X signatures in scRNA-Seq analyses of the mammalian hematopoietic system (Velten et al. 2017; pages 13-14). Future studies will very likely find that clusters such as X play an important and specific role in blood development. But for now, we have minimized speculations in the text regarding X.

Reviewer #3:Using a combination of bulk RNA-Seq of FACS-sorted cells and single cell RNA-seq, the authors identify various blood cell subpopulations that compose the *Drosophila* hematopoietic organ called the lymph gland. This study has been performed at one developmental time point, mid third instar larvae. The authors perform a pseudo time analysis and propose a developmental trajectory with multiple paths to mature blood cells types. RNAseq data suggest that different blood cell types express genes involved in various metabolic processes. They establish that Pointed has different roles during lymph gland hematopoiesis. Finally, they identify that Numb and Musahi are involved in a Sima dependent Notch non canonical pathway in mature crystal cells.This analysis is of interest, however in the current version, it is too preliminary.1. The list of the main genes defining each sub-population inside the different clusters, as well as their expression in all the other sub clusters, has to be provided in the main figures.

This point is addressed above in essential revision 9.

2. No validation of RNA seq data is provided: A spatial reconstruction in vivo by profiling the expression of a subset of genes identified by RNA seq is necessary.

Please see essential changes point 3.

3. To support the developmental progression of lymph gland cells proposed in Figure 7,lineage tracing experiments are required.

Figure 7 is a model figure to help make the discussion section more comprehensible. It is the summary of the transcriptomic and genetic data resulting from the analysis. It shows a schematic representation of a snap-shot in time and the diversity of cell types. Lineage tracing involves studies through real development using specific cell drivers for each zone, which is not the focus of this paper.

4. Comparison between RNA seq data obtained by (Cho et al 2020) has to be given in the main text. Furthermore, discrepancies between these 2 studies have to be clarified. How the AUC list has been established? This is a key point. For example, the PH1 cluster identified by Cho et al is spread out in MZ1, PSC, X and MZ2 cluster in this study. Why are the results so different? Delta is a marker of PH1, which is validated by analyzing its in vivo expression profile. What about delta expression in the scRNA seq performed here?

These concerns have been addressed above in essential revisions points (1, 3, and 5).

On a practical note, Professor Shim is a close colleague and past member of our laboratory. She made her data available to us long before its publication. We have concentrated on how this transcriptomic approach in marker identification (new for lymph gland studies, but quite commonplace as a tool otherwise), can be extended to contribute to functional analysis and explain new developmental mechanisms in *Drosophila*. It is not our goal to repeat all their findings since the entire gene list is included.

It is reasonable to ask us to provide similarities and explain sources of differences, in the overall context of the two studies. We have now placed this in the main text as the reviewer suggests (page 3536) while not removing it from the very detailed comparisons that we had already provided in the original manuscript (Supplementary file 2). In Figure 1K, and in the essential revisions section (3), we now include *Delta* expression and also explain why the results are consistent with Cho et al. The Shim lab is working on Delta and there are many other genes for us to concentrate on. We also point out (essential revisions 3), that *Delta* is not a “marker for PH1”; it is expressed in PH1 as well as in the PSC, just as we find (page 11 and Figure 1K). The differences between subclusters in Cho et al., and clusters in this manuscript are now made clear in multiple places in the text and in these responses to the Reviewers’ comments. And we thank them for bringing this up.

5. There is a discrepancy between the introduction and the main results of this paper. In the introduction the authors focus our attention on the IZ, but in fine we don't learn much about IZ cells from this analysis. Instead of deciphering IZ identity and fate by in vivo profiling, most functional analyses performed concern crystal cell maturation. This part is developed via 2 main figures among 7, plus 5 sup figures. From my point of view, this study represents an ideal opportunity to better characterise IZ cell identity, lineage and function. Unfortunately these data are missing in the current version of the manuscript but could be added instead of the data concerning crystal cell maturation, which is somewhat out of the scope of this manuscript and could be published in a separate paper.

The manuscript about IZ that the reviewer is likely referring to is repeatedly cited in the manuscript and available in bioRxiv. It is under minor revisions for publication in the journal Development (Spratford et al. 2020). We cite the manuscript as well as the *CHIZ-GAL4* line, which is carefully and thoroughly characterized there. That said, we have added some more functional data on IZ to this revision (Figure 2J-L; Figure 2—figure supplement 2C; Figure 4—figure supplement 1F), in addition to the material on IZ that was presented in the original manuscript (Figures 1, 2, 3, 4, 7 and their corresponding supplementary figures); and as it stands now, this manuscript has more data on IZ than any other published material. Author’s note: this paper is now in press in Development.

We appreciate the Reviewer’s opinion about publishing another paper with the CC data. Based on the argument presented in essential revisions (3), doing so will be contrary to how we define proper validation of transcriptomic data in *Drosophila*. Validation by functional analysis was one of the primary motivations for initiating this extensive study.

6. This manuscript has to be focused on the novelty given by the RNAseq data. The data concerning crystal cell maturation, which is somewhat out of the scope of this manuscript, could be published in a separate paper.

See Reviewer #3 comment 5. We are hoping that this paper will help move *Drosophila* analyses of transcriptomic data to a step beyond identification of markers. The novelty in this work is the coupling of transcriptomic and functional data, not so easy to do in mammalian hematopoiesis.

7. There are 7 main figures and 16 sup figures + 1 additional file. All these Sup figures give information and make suggestions that unfortunately are not validated by additional experiments. Overall the reader is left with a lot of observations that are not further validated and in fine one cannot rely on. Data presented in this manuscript have to be focused to avoid overloading the reader with too many side observations, which in turn lead to losing the thread of the message of this study.

Please see essential revisions (12). The supplementary and main figure data are treated with equal rigor. Many supplementary data panels have been consolidated.

8. I have concerns about the single cell RNA seq data, since essential information is missing.– What about the cell numbers in each cluster? The IZ cluster (Dome-GFP+ and Hml+) represents a small subset of lymph gland cells based on the CHIZ expression profile (see Figure 3K); however, it corresponds surprisingly to a quite large lymph gland cell subset, as illustrated in Figure 1J. How can one explain this?

The cell numbers/percentages in each cluster (and each biological replicate) are now included in (Figure 1J; Figure 1—figure supplement 2B). These are in agreement with other independent studies.

Figure 3K in the original version is Figure 3D in this revision. It shows a part of the middle third section of a lymph gland, and makes a specific and quite a different point. We are not sure how the reviewer can surmise from this panel that less than 15% of cells comprise the IZ cluster. Particularly since this manuscript is the first to actually quantify the number of IZ cells and present the data in a statistically meaningful way. The IZ cluster in the scRNA-Seq represents ~15% (14 -16% between the 3 biological replicates). This is consistent with the data in Spratford et al. 2020, where *CHIZ-GAL4*+ cells, representing the first bona fide positive marker of IZ, account for ~17% of the total number of cells at 96h AEL. So there is no discrepancy. In fact, these data support our argument that *CHIZ-GAL4+* cells (17%) are akin to cells of the IZ cluster (14-16%) and distinct from the cells of the other transitional cluster (proPL; 25-27%).

– To identify subpopulations in clusters, the authors performed sub clustering on isolated clusters for PL and CC. Why was this not done in the same way for the other 3 main clusters (MZ2, IZ and proPL)?

This is a great question. PL and CC cells are both in stages of terminal differentiation. Nearly all CCs are in pseudotime State 6 and PL cells are in State 7. These are reasonably sized clusters, which when sub clustered (using graph based unsupervised methods), can still be represented by single gene expressions, providing validation of them as being subclusters (eg., high or low *lz-GFP* for CC and high or low NimC1 for PL). In contrast, more dynamic clusters eg., MZ2, IZ etc are in many different states (meaning stages of development at any particular time). Subclustering non-terminal, progenitor and transitional populations that are in mixed developmental stages (States) or are particularly small, such as with MZ1, does not generally yield robust subclusters.

– For the IZ sub-cluster: The plot in Figure 2 sup 3I is very misleading, since it suggests that genes expressed in the IZ are specific to this cluster. For example, "state 3" is present both in the MZ and proPL (Figure 2), but in Figure 2 sup 3l it is only represented in the IZ and not in the MZ and proPL clusters. The same remark holds for states 4, 5 and 7. Furthermore, as I mentioned above, the list of the main genes defining each sub-population inside clusters, as well as their expression in the other sub-clusters, has to be provided in the main figures. Furthermore, a spatial reconstruction in vivo by profiling the expression of a subset of genes identified in sub populations is mandatory to validate the RNAseq data.

As for the list of genes, this has been addressed above in essential revision 9 and in response to Reviewer #3 comment 1. We have removed Figure 2 Sup 3I since the reviewer finds it confusing/misleading. The essence of this information is available in Figure 2D-E. We have also added a dot plot (Figure 2 figure supplement 1K) to make the point about IZ-enriched genes more clear. We further note that 5 out of 6 of these genes are differentially expressed (enriched) in the IZ cluster (Supplementary file 1). Many of the other questions raised have been addressed in essential revisions, and should be resolved with the three important points that we have already made above, but are perhaps worth restating:

1. It is not our expectation that we will find single gene markers for every population. In the new Figure 7A, we describe combinations that reasonably represent a cluster. What is measured is highest relative enrichment compared with other populations, not uniquely expressed markers. We have been careful to say “enriched” rather than “specific” in the context of the IZ, and even for individual genes.

2. States and subclusters are not the same. The state represents advancement in pseudotime. A number of cell types could belong to the same developmental time (state). Clusters are made on the basis of global differential gene expression and they may therefore usually represent different cell types (e.g. MZ1 or PL). The clustering process can break up heterogeneity within what was previously thought of as a single cell type (such as MZ broken into MZ1 and MZ2). The state designations such as MZ2-3 and IZ-3 are not “subclusters” of MZ or IZ, rather they are parts of two different clusters that are in developmental state 3 in pseudotime. MZ2-3 is more similar to MZ2-1 than to IZ-3 in spite of their both being in state 3. In contrast, PH1/2/3 etc of Cho et al., are subclusters not merely states in pseudotime and could, in principle, represent cell types if proven so with markers.

3. For the purposes of gene expression differences, in our study, it is best to look at clusters (e.g. MZ1 and MZ2) or true subclusters such as iCC and mCC. But while individual gene expression can vary as a cell matures through pseudotime states, these differences do not represent enough of a change in transcriptomic (and therefore in markers) to represent unique cell types. Our apologies, if this was not made clear in the text, we have tried to make these distinctions clear in the revision and in this response to reviewer’s comments that we have taken very seriously.

– Why is the plot shown in Figure 1J different from the plot shown in Figure 2 sup 5 I? The t-SNE graphic representation does not give any indications concerning whether clusters are related or not.

The t-SNE in Figure 1J is a 2D (2 dimensional) t-SNE representation of the data (i.e. the Ndimensional data reduced to 2) while the plot in the previous Figure 2—figure supplement 5I (which is now moved to a main Figure 2F) is a 3D (3 dimensional) t-SNE representation. The principal components are marked as axes of a cartesian plot on the figure (t-SNE 1, t-SNE 2, and t-SNE 3). We have included the words “3D t-SNE” at the top and color-coded arrows and arrowheads in (Figure 2F) to indicate the locations of the adjoining clusters. Finally, a true 3D rendition of the 3D t-SNE was also provided (Video 1) where the relative positions of all clusters is more obvious.

– Why are there discrepancies between gene expression levels and their representation on the corresponding plot? Please see for example the case of CG30090 in Figure 2 sup 3B and the corresponding plot in C. CG30090 is expressed at a similar level in proPL and PL, but its expression in proPL is lacking in the plot. Why?

These plots were thresholded to show highest expressing cells. We have now removed them as the dot plot (Figure 2—figure supplement 1K) replacing them makes the same point, only better.

– Concerning IZ markers, among the 6 identified by bulk RNA seq, only 4 of them have been analysed in single cell experiments. What about the 2 others?

We have now added a dot plot representation of all 6 of the IZ enriched markers identified by bulk RNA-Seq (Figure 2—figure supplement 1K), and noted that 5 out of the 6 genes are differentially expressed in the IZ cluster (page 15; Supplementary file 1).

– In Figure 7, the model of developmental progression of blood cells proposes that there is no transition path between IZ and proPL. This proposition does not fit with the data. Indeed, in the pseudo time analysis there is a clear overlap between IZ and proPL, indicating that they are connected (see states 4 and 5 in Figure 2 O-P). Genes highly expressed in IZ are also highly expressed in proPL. This is observed for all the IZ markers analysed in this manuscript (please see Figure 2 sup 3B, D, G, H). Determining whether there is a transition path between IZ and proPL has to be validated by in vivo experiments.

This point is fully addressed in essential revisions (3). The overlap in pseudotime between IZ and proPL, which we believe is now better illustrated in new Figure 2G, indicates that both distinct cell types (distinct by unsupervised graph based clustering, size distribution, and also supported by in vivo labeling and functional experiments) represent two separate transitional populations that bridge MZ2 and PL. The overlap in pseudotime implies they each follow similar states of maturation.

As for no transition between IZ and proPL, we agree with the reviewer that we overstated the case; we have revised our explanations and details are in essential revisions point 3.

The reviewer is correct that IZ and proPL have several genes in common. We had commented on this in the text (pages 20-21). Both are transitioning populations and have expressed genes in common. But their individual functions set them apart. And their minimal contact boundaries makes them separate, without precluding the possibility of some cross talk pointed out by the reviewer. Our new representation of pseudotime (Figure 2G) shows more clearly how the different zonal clusters can be overlapping in pseudotime and yet maintain their cluster identity.

– Figure 7: Regarding the translational link between PL7 and iCC7 (Figure 2 sup4), again this proposition has to be validated in vivo. Furthermore, how can iCC7 (more engaged in maturation) give rise to iCC6 (less engaged in differentiation). This also needs to be validated.

There have been several past studies that have presented convincing data for dedifferentiation and transdifferentiation of plasmatocytes to CCs (Leitão and Sucena 2015; Terriente-Felix et al. 2013) so there is genetic precedence for such reversion and in that respect, this is not an entirely new phenomenon. We cited both manuscripts. To add to that, iCC-7 cells show similarities to PL-7 that the rest of CCs do not share (Figure 2N; Figure 2—figure supplement 3G). Since virtually all PL cells are placed in state 7, and CCs mature in state 6, any conversion from PL to CC would have to represent a state 7 to state 6 transition. Note that the route to mCC-6 is from iCC-6 which does not involve such reversals.

– Concerning the MZ1 cluster, Ubx is expressed in these cells. Ubx is specifically expressed in lymph gland posterior lobes that are composed of hematopoietic progenitors expressing markers of MZ cells (Rodrigues et al., 2021). Altogether these data strongly suggest that MZ1 cells correspond to posterior lobe hemocytes. This has to be clarified.

This is addressed above in essential revision (3). The tertiary lobes were not included in this study.

– For cluster X, since DNA damage markers are expressed, this strongly suggests that this cluster might correspond to unhealthy cells damaged during the experiment. What about their ribosomal content (a criteria commonly used to check for cell health)? Is the molecular signature of cluster X found in bulk RNA seq and in the Cho et al., 2020 paper?

Part of our initial quality control in scRNA-Seq is excluding cells with especially high or low mitochondrial gene content (as a check for cell health and to exclude any damaged cells), and Cluster X cells show similar mitochondrial reads as other cells (Figure 1—figure supplement 2C). Cluster X cells also show similar levels of ribosomal content as other cell clusters (Figure 1—figure supplement 2D).

The genes we see expressed here were first identified by UV irradiating mammalian cells that are blocked in G2 until their DNA is repaired. Wieschaus amongst others has shown this also happens in normal cells in S-G2 transitions that are also transcriptionally active (Blythe and Wieschaus 2015) and display so called “replicative stress”.

As for comparison with the Cho et al. 2020 data, this signature is seen in what they refer to as the “GST-rich” cluster. X and “GST-rich” are both primary clusters that share a large number of differentially expressed genes (Supplementary file 2B) and are similar in size (1.14% vs 1.21%, respectively) (Supplementary file 2E). And when their shared genes are analyzed in GSEA, we find that genes involved in “cellular response to DNA damage stimulus” are strongly enriched in both (p-value=0.00004).

– What is the unit given for the Y axis in Figure 3? Why is the scale different from one graph to another and does not start always at zero? For all graphs in this manuscript the p value is missing, so the reader cannot not figure out whether the differences are statistically significant or not. Concerning the AUC analysis, how is the list of genes taken into account for a signalling pathway or a function that has been established? What kind of conclusion can be drawn from analyses regarding metabolism? In other words, considering the PSC as an example of a group of cells where glycolysis genes are highly expressed, what is the impact of this on PSC function, and how does potential glycolysis in the PSC help us to understand PSC function?

Please see above essential revision 1 for a discussion of AUC values, statistical significance, and p-values. Please see pages 21-22 for discussion of the impact of high glycolysis and low OxPhos on PSC function and pages 38-39 for discussion of what conclusions can be drawn from analyses regarding metabolism.

– Figure 3K: the authors need to define what CHIZ is. Since there is no staining overlap between MMP1 and CHIZ-GFP, the authors cannot conclude that MMP1 is expressed in the IZ. Furthermore, MMP1 transcripts are detected at high levels not only in the IZ but also in the MZ2, proPL and PL (Figure 3J). How can these results be reconciled with the expression profile of MMP1 shown in Figure 3K?

We have defined *CHIZ-GAL4* (page 17) and included a reference to the paper that describes how this split *GAL4* was constructed and validated (Spratford et al. 2020). We mention in the text (page 2526) that MMP1 is a secreted protein and so we expect to see MMP1 protein in neighboring cells. But all of this protein is altered by manipulation of JNK in *CHIZ+* cells (Figure 3D-F’’). MMP1 expression is lost with loss of function and dramatically increased with gain of function of JNK in the *CHIZ*+ cells alone.

–Figure 4: Quantifications are missing. Crystal cell and MZ indexes have to be given.

Thanks for pointing this out. This has been addressed in essential revision (7).

– Figure 4G and H: That tep4 is expressed in a subset of MZ progenitors defined by dome>GFP is not new, this has been established previously. Please see Benmimoun et al. 2015, and Oyallon et al 2016.

We cite them for their work (page 26). Figure 4G-H (now Figure 4E-F) has a different purpose. Since IZ cells are positive for both *Hml^Δ^* and *dome* (authors’s note: in the context of CHIZ-GAL4), we needed to design *QF* and *GAL4* constructs to demonstrate that some of these *dome*-positive, *Tep4*-negative cells are not IZ, i.e. there is a pre-IZ population beyond *Tep4*. This is a new finding and key to our interpretation of the experiments in Figure 4.

– What about the Pnt expression profile in the LG? Figure 4C-D: DomeMESO>pnt RNAi , in addition to a defect in blood cell differentiation, this leads to a smaller LG compared to the control. This defect in size has to be mentioned and quantified. Since the role of pnt in the MZ has been previously reported by Dragojlovic-Munther M et al , 2013, this paper has to be cited. Furthermore, the Dragojlovic-Munther M et al. study indicates that in addition to preventing hemocyte differentiation, pnt RNAi in the MZ leads to lamellocyte differentiation. Do lamellocytes differentiate when pnt is knocked down using domemeso, tep4 , CHIZ and hml gal4 drivers? In Figure 4F :Tep4>GFP>pntRNAi , GFP levels are decreased compared to the control. Does Pnt control the expression of the Tep4-gal4 driver? In the text, p35 line 886 "Pnt loss in MZ2.3" is an over interpretation, since no gal4 driver specific for this group of cells has been used to perform the lof experiment. p35 lines 903 "plasmatocytes to be converted into crystal cells" is erroneous, since hml-Gal4 expression is not restricted to mature plasmatocytes but is expressed both in plasmatocyte and crystal cell precursors. p35, lines 905-910 should be in the discussion, not in the result 'section.

The Dragojlovic-Munther and Martinez-Agosto 2013 is cited on page 26 (and was cited in the original submission). Unfortunately, that study did not provide the specific *pnt^RNAi^* line used and there are aspects of that study that for us, needed revisiting as they do not reproduce in our hands. So we left it at that, citing them. We did not look for lamellocyte formation. The issue about MZ2-3 is now explained better (page 27). Loss of *pnt* has no effect on *Tep4>GFP* expression (Reviewer #2 comment 4). In regards to suggestion about line 903, we have replaced “plasmatocytes” with “*Hml^Δ^-GAL4* positive cells” (page 28). The discussion item about RTK and Notch has been modified to reflect the *pnt* data (also in Reviewer #1 comment 1).

– Figure 5: Numb and crystal cellsThe paper Cho et al 2020 has to be mentioned in the text, since it established previously by immunostaining that Numb is expressed in crystal cells.

Cho et al. 2020 showed numb::GFP expression in CCs, but did not note differences between protein and RNA expression in CC subsets. The difference in protein expression between iCC and mCC is the main point of this analysis. Nevertheless, we have now cited this observation (page 31).

– A previous study done in Banerjee's lab, reports on the role of Sima and Notch in crystal cell survival by preventing their dissociation (Mukherjee et al., 2011). In the present manuscript, no data and comments refer to crystal cell survival depending on Numb. Thus in the current version of the manuscript, it remains unclear as to what is the role of Numb in crystal cells. Does it control iCC to mCC, or is it required for survival of mature crystal cells? The confusion is sustained by sentences such as: "depletion of Numb prevents the maturation of iCC to the mCC state", please see p 39, lines 1000. Since Numb is detected at high levels in mCC and not in iCC, the function of Numb should be in mCC once they have matured. Furthermore, it has been previously published that Sima is required for crystal cell survival. A decrease in Sima levels is observed in Lz>numbRNAi conditions, supporting the proposition that CC depleted from Numb should disrupt since they lack Sima . In conclusion, the role of Numb in either crystal cell maturation (i. e., going from iCC to mcCC) or mCC survival has to be clarified.

We do not see a major contradiction between what we state and the reviewer’s comments. While the title of the Mukherjee et al. 2011 paper does not mention it, Figure S4 in that paper clearly shows in the model that Notch/Sima signaling is required for “late crystal cell maturation and survival”. The maturation data was not the focus of that paper but was clearly demonstrated and is now replicated and validated by both RNA-Seq and flow cytometry data. We chose to focus our study on the maturation aspect and not survival. This does not, in any way contradict the notion that if mCCs happen to form, sustained lack of the Notch/Sima signal will cause them to burst.

The role of Numb is schematized in the model (new Figure 7D). The observed loss of *numb* phenotype using the RNAi reagents (Figure 5O-T; Figure 5—figure supplement 3D-I) is fewer mCC and more iCC than in wild type (without a change in the overall number of CCs). Sima/Notch signaling in mCC cells turns on genes that then give those cells the mCC identity. So with severely attenuated Numb/Sima/Notch signaling are you an mCC that looks like an iCC or an iCC that never became an mCC? Beyond semantics, the data clearly show a decrease in the number of mCCs.

– Figure 5 sup2 : Quantification is missing. p 38 lines 979-981. The authors need to clarify whether there are talking about an increase in Numb levels per crystal cell, or an increase in Numb levels per LG, which in this case reflects an increase in the number of cell expressing Numb.

Numb quantitation has been clarified in the Methods (page 69; see also essential revisions 6).

– Numb subcellular localisation is different from one picture to another. In Figure 5M, Figure 5 sup2 , Figure 6A-c and 6 H-L', Numb is mainly localised at the periphery of the cell at the plasma membrane, whereas in Figure 5 sup 3A-B and sup 3F-G, Numb is detected as cytoplasmic punctate dots without staining at the plasma membrane. This is confusing and clarification is necessary.

These are different assays. The live endocytosis assay experiments with active internalization (Figure 5—figure supplement 3A-B) are quite different from conventional staining of fixed tissue (Figure 6A-C, H-L’; Figure 5—figure supplement 2A-C’’’), this likely accounts for the differences in Numb staining. The endocytic punctae are less prevalent but can still be seen (Figures 5M, N’) with conventional staining but the membrane staining is more prominent.

– Figure 5 sup 3: PPO2 staining shown in Figure 5 sup 3 K-L is not in agreement with the quantification given in M. p values are missing in M and N. There are also discrepancies between these figures and the text p 39: "numb RNAi expressed in crystal cells causes ...with a concomitant increase in the iCC population". Figure 5 sup 3N: Quantification of crystal cell numbers is not convincing. Since there is a lot of variation among LGs, quantification of PPO+ cells has to be given as an index (i.e., a ratio between the total number of PPO+ cells/ per total number of LG cells). In N, 6 LGs maximum per genotype have been analysed, which is far too few. Defining whether the number of crystal cells is affected in lz>numbRNAi is essential to determine whether Numb is required to allow iCC to mature into mCC, or whether it controls mCC disruption. The MM section has to be completed; it should indicate how quantifications of cell numbers and fluorescent intensity were performed.

Both the PPO2 staining in Figure 5—figure supplement 3F’’, G’’ and the quantification in H and I show that in a *numb^RNAi^* background, CCs with the highest levels of PPO2 (i.e. mCCs) are depleted (H) but the overall number of CCs is not changed (I). p-values are now included in H (p=0.0001) and I (not significant) to support the data. The effect of *numb^RNAi^* on CC formation is not based solely on staining but on three independent experiments all showing a decrease in mCCs (Sima punctae in Figure 5O-Q’ and Figure 5—figure supplement 3D-E’’; PPO2 levels in Figure 5R-S’ and Figure 5—figure supplement 3F-I; and *lz>GFP*^HI^ cells in Figure 5T) with no change in total CC numbers (Figure 5T; Figure 5—figure supplement 3I). The flow cytometry analysis (Figure 5T) also shows a decrease in mCCs (GFP^HI^) and a concomitant increase in iCCs (GFP^LO^) with no decrease in total CC. All together these results fully support our model (Figure 7D). The methods section includes how quantifications of cell numbers and fluorescent intensities were performed (page 69).

– For Hnt staining in lz>numbRNAi, there is a discrepancy between Figure 5 sup 3 l-l" and Figure 5 sup 3 L-L". In I-I" no difference in Hnt levels compared to control (H-H) is observed, whereas in L-L" a strong decrease is observed (K-K").

As we note above (Reviewer #2, comment #5) and in the manuscript, the number of CCs does not change but the maturity does. Both Figure 5—figure supplement 3I-I” (now Figure 5—figure supplement 3E-E’’) and Figure 5 sup 3 L-L" (now Figure 5—figure supplement 3G-G’’’) show the same concept: that signs of CC maturity (Sima punctae in E-E’’ and PPO2 levels in G-G’”) decrease while the total number of Hnt+ CCs does not change. The Hnt staining in E’ and G’ were done with different fluorescent secondary antibodies (with different dynamic ranges) and so cannot be directly compared. The visual comparison between intensities of the fluorophores is trumped by the statistical data (Figure 5O; Figure 5—figure supplement 3H-I).

– The enlargements shown in Figure 5O-R have been taken from pictures shown in Figure 5 sup 3K-L. It would be better to show independent immunostainings. This remark is even more relevant in this case because the staining in Figure 5 sup 3 L is not convincing.

These data are quantified and the statistics shown (Figure 5O; Figure 5—figure supplement 3HI; p-values between 0.0001 and 0.0063), which should ease the reviewer’s concerns. We have indicated that images in Figure 5 sup 3K-L (now Figure 5—figure supplement 3D-G’’’) correspond to lower magnifications of Figure 5O-R (now Figure 5P-S’) in the figure legends. Showing an inset with high magnification has a different purpose and is a standard practice to help the reader hone into an area of interest in that particular picture and does not imply that we have only stained one lymph gland since the statistics are provided.

– Figure 5 sup 3 H-I: lz >numb RNAi there is a decrease in Sima staining. Is it due to a decrease in sima transcription and/or Sima protein stability and/or Sima subcellular localisation?

This question is beyond the scope of this manuscript but is an important suggestion to explore in future studies.

– Figure 6 J and M ; is "the Numb intensity" referring to the intensity of Numb per cell or the total amount of Numb intensity measured per LG? This has to be clarified in the figure, in the text (p 40) and mentioned in the MM. What about the crystal cell index in lz>msi RNAi and Hml>msRNAi?

Thanks for pointing this out. Some of the technical details of Numb intensity measurement are included in the results (page 33), the Methods (page 69), and Figure 6J, M legend, and also addressed in essential revisions (6). But to respond to the reviewer’s more important real concern, all intensities are measured on a “per cell” and not an overall LG basis.

The *msi^RNAi^* experiments focused on whether Numb protein levels changed. The expectation is that the phenotype on CC cells will reflect Numb levels, but a formal analysis of Msi’s effects on CC number will be explored in future studies.

– A schematic representation of crystal cell maturation involving N (both the canonical and non-canonical signalling), Numb and Sima would be very helpful.

Excellent idea! We have provided a schematic representation in Figure 7D.

– In MM p 74 lines 1888 "data was corrected for batch effects between samples". The information concerning the method used has to be provided.

Information concerning the method used for batch effects corrections has been provided in the Methods (page 66).

9. Modifications in the text are required– Lines 1093 "the equilibrium signal from proPl". Functional data supporting this conclusion are lacking.

Functional data showing loss of equilibrium signaling (*Pvr^RNAi^*) in *Hml^Δ^-GAL4*+ cells results in a strong phenotype (at an early time point) but no phenotype is seen when driven in *CHIZ-GAL4*+ cells (at any time point) are provided in new Figures 2J-L and Figure 2—figure supplement 2C (pages 17-18). Because the equilibrium signal initiates at a developmental timepoint (2nd instar) when PL cells are not present, we conclude that *Hml^Δ^*+ proPL cells participate in equilibrium signaling.

– Lines 1088 "JNK signalling ... is a specific property if IZ cells". Neither JNK expression nor its function has been analysed in this study.

We performed loss and gain of function analysis using mutant forms of JNK pathway members. MMP1 is a known target of JNK and we show that levels of MMP1 change upon manipulation of JNK signaling with loss of *JNKK* (*hep^RNAi^*) and activation of *JNKK* (*hep^ACT^*) in *CHIZ-GAL+* IZ cells (Figures 3D-F’’; pages 25-26). We also show that AP-1 complex (Fos/Jun) targets (which functions downstream of JNK) (including *Mmp1* and *GlcT*, noted in the text on page 25) are also enriched in the IZ (Figure 1 figure supplement 3A; FDR=0.00243).

(Authors’ note: Following the detailed response and corresponding changes to the reviewers’ comments, the resubitted manuscript needed more clarifcation for the reviewers.)

[Editors' note: further revisions were suggested prior to acceptance, as described below.]

Essential revisions:1. The authors should temper some conclusions because the conversion of AUC scores to z-scores can obscure the actual level of enrichment. Specifically, the authors should:a. Soften the conclusion that proPL generates the equilibrium signal and that the IZ alone activates the JNK pathway as this is based on a low number of genes (lines 959-960). Please rephrase this statement.

On page 38, in the Discussion section, we have replaced the statement “For example, proPL (but not IZ) generates the equilibrium signal, whereas the IZ alone activates the JNK pathway” with the phrase “For example, together the genetic and RNA-Seq data suggest that proPL is likely a major source of the equilibrium signal, whereas IZ largely contributes to the JNK signal.”

b. Point out in the text that JNK activation is enriched in but not specific to the IZ.

The statement addressed in 1a. was from the Discussion Section, which we have now tempered. In the Results Section, we have not ever used the word “specific” in reporting on any transcriptomic data (as we explained in the previous rebuttal). We have consistently used the term “enriched'' since we agree that genes are not on/off in any one cell type.

This includes the results on JNK (page 25-26).

Also in results, we interpreted the genetics of the MMP1 data on the basis of its staining being completely eliminated in a *CHIZ-GAL4, JNKK miRNA* background and not based solely on the staining pattern of a secreted protein. Nevertheless, we have changed the phrasing.

Previously stated: *"*establishing the IPs as MMP1 producing cells*"*

Now changed to the phrase “suggesting the IPs are a source of MMP1” (page 26).

c. Clarify the in vivo definition of the IZ and the ProPL cells with respect to the Hml delta QF driver. The authors use it to specifically label the IZ but the driver's expression domain is broader than IZ cells.

We have added extra details to make this clearer (page 17).

Briefly, *Hml^Δ^-GAL4* and *Hml^Δ^-QF* are identical in their expression and they fully overlap. What is different is *CHIZ-GAL4* (in Spratford et al.), which is a split *GAL4* that contains a combination of *Hml^Δ^* and *dome^MESO^* enhancers and includes the strong p65 activation domain. *CHIZ-GAL4* expression colocalizes with the overlap between the directly driven *Hml-DsRed* and *dome^MESO^-GFP* reporters*,* and these *Hml-DsRed, dome^MESO^-GFP* double positive cells define the IZ in our bulk RNA-Seq. In the current manuscript, we demonstrate that *CHIZ-GAL4,* which marks IZ cells, and *Hml^Δ^-QF*, which marks proPL and PL cells, are non-overlapping (Figure 2H-I; Figure 2—figure supplement 2C-D). The reason is not fully clear, but is likely due to differences in timing or level of expression. We apologize if this was not stated clearly and we hope the new description is more useful (page 17).

2. The authors should indicate in Figure 3—figure supplement 3, 1D which AUC terms encompass "glycerolipid remodelling genes" (line 592)

We have included which AUC term encompasses “glycerolipid remodelling” in the AUC table (Supplementary file 4, row 23) as well as in the text (page 23-24) and the legends for Figure 3—figure supplement 1D and 1F (page 58-59).

3. The authors should rephrase "pnt transcript is expressed at low levels in the PSC (Figure 4A)". (Line 664). One interpretation of Figure 4A is that pnt is expressed at the same levels across the different clusters but in a smaller number of cells in the PSC (hence the smaller circle).

Yes, that is correct, and we have changed the sentence to “*pnt* transcript is expressed in very few cells in the PSC” (page 26).

4. The authors should provide publications for the enhancers in Figure 5—figure supplement 1 B,C,G or explain how they defined the enhancers.

These references are now included in the figure legends for Figure 5—figure supplement 1B-C and G (page 60)*.*

5. The authors should remove the comma in line 795 ("we find, is").

We have removed this comma (page 31). (Author's Comment: Although it was grammatically correct with the comma).

6. The authors should avoid the term de-enriched/de-enrichment (lines 525, 549, 562, 836, 135, 1136, 1159, 1435, 1437, 1438) to indicate a change in the levels of expression.

Thank you. We have replaced these terms with “not enriched” (pages 21-22, 33, 45, and 58) or “absence of enrichment” (pages 22 and 46).

7. The authors should modify their conclusion (line 652) "establishing the IPs as MMP1 producing cells". The diffused MMP1 staining in Figure 3D-D' is not convincing.

As described above in point 1b, the MMP1 conclusion is based on the genetic data that all MMP1 expression is lost when the JNK pathway is knocked down in IZ cells in a *CHIZ-GAL4*; *UAS-JNKK miRNA* background.

Nevertheless, we have changed the previous phrase: "establishing the IPs as MMP1 producing cells" to the phrase “suggesting the IPs are a source of MMP1” (page 26).

8. The authors should address how they conclude (line 696) that "Pnt is required for exit from the IZ" since there is no defect in CC differentiation in CHIZ >pnt RNAi (Figure 4l-M). They should also comment on Pnt's role in plasmatocyte differentiation.

As explained (page 27), there are routes to CC that bypass IZ, which we demonstrated by showing normal CC formation despite the complete absence of IZ cells (or any other *Hml+* cells) in a genetic background with *pnt^RNAi^* driven in MZ using *dome^MESO^-GAL4* (Figure 4B-D)*.* We believe that these minor routes may become more dominant under mutant conditions (further elaborated on page 38-39)*.* We have also added an additional detail to the text related to the effects of *pnt* loss of function on plasmatocyte differentiation. We now point out that Spratford et al. demonstrates that Pnt is required for exit from the IZ and plasmatocyte differentiation (page 27-28). Our conclusion is supported by the cited data (Spratford et al.) that shows the absence of plasmatocytes when *pnt^RNAi^* is driven in IZ cells, but the data in this manuscript (Figure 4L-M) clearly shows CC differentiation still occurs in this background.

9. The authors should modify the introduction so that it includes a statement about the cardiac tube acting as a niche to control lymph gland hematopoiesis and supporting references.

OK, we have done so (page 3) and cited two Crozatier papers for this work.

10. The authors should add a sentence/short paragraph in the discussion saying that the proposed model/definition of states awaits functional confirmation, at least in some cases.

We have added this statement (page 37).

11. In the dot plots, the authors should indicate what is meant by 'mean' and 'non-zero percent'.

This has been included in the legends for Figure 2N (page 45), Figure 4A (page 47), Figure 5L (page 49), Figure 6E (page 51), Figure 2—figure supplement 1J-K (page 56), and Figure 2—figure supplement 2H (page 57). In these figure legends we say that for each of the populations shown in the dot plot, the color of the dot represents the mean expression level of a given gene (mean) and the dot size indicates the percent of cells that express that gene (non-zero percent).

12. In Figure 2H,I, the authors should show the whole lobe and indicate the IZ and the proPL with arrowheads.

This is now included in Figure 2—figure supplement 2C-D.

Suggested revisions:1. It is suggested but not required that the authors provide a volcano plot to illustrate the differences between proPL and IZ.

Now included in Figure 2E and described in the text (page 17). We thank the reviewers for this suggestion, that has helped us visualize and better understand the differences between the two populations.

2. It is suggested but not required that the authors provide the single table that must have been generated to produce the figures of the first submission and used to calculate the z-scores.

The figures in the first submission were not based on a single table. Each figure is based on a separate table that includes the individual AUCell values for every individual cell identified in the scRNASeq along with each cell’s identity and that cell’s cluster, state, and cluster-state assignment. We therefore do not believe such a collection of tables will be helpful for the reader. But as we have included the full scRNA-Seq dataset, list of individual genes for each AUCell list, and the parameters we used for the AUC analysis, any interested researcher can reconstruct those graphs and any other analysis included in the original or re-submission. We believe that the way we now display the AUC data in heatmap format, which is a common practice for such analysis (e.g., Zhu et al. 2020; Blood 136 (7): 845–856), accurately portrays the most important aspects of the analysis which is the difference between groups and not the absolute values for the AUC lists (which we and the reviewer noted is highly dependent on the number of genes in the list, the number of genes identified in the scRNA-Seq, and the parameters set for the analysis). For example, taking mitotic G2/M genes as AUC, we find that the mean AUC score over all cells is 0.08 and the value for cluster X is 0.12 (other clusters in the 0.09 range). Looking at these data alone makes all clusters seem similar in value, and the absolute numbers are small. But the value for X is greater than 3 standard deviations (SD) away from the mean calculated for all cells taken together, while the other clusters are less than 1 SD away. The exact value of the AUC for any cluster varies from 0 to 1 and is not meaningful and cannot be compared with other AUC’s in the same table. Therefore, we think that the table would confuse the average reader, who will likely think that our conclusions are drawn from the absolute AUC scores or absolute differences between them.